Resource

# A transcriptomic microglia taxonomy across mouse and human pathologies

Single-cell studies have revealed substantial microglial diversity in development, homeostasis and disease. However, a framework enabling comparison and stratification of microglial states across contexts is needed. Here we generated an atlas of myeloid cell states by single-cell RNA sequencing more than one million central nervous system cells from more than 30 physiological and pathological conditions. This atlas enables us to establish a comprehensive taxonomy of myeloid cell states across brain disorders and related mouse models, comprising 27 superclusters and 192 clusters that are prevalent across diseases and largely conserved. We augment this taxonomic framework with spatial transcriptomics to map how immune cell states are organized within tissue and interact with their local cellular environment. Using in vivo perturbations, we also show that activation-associated microglial states are dependent on interferon and colony-stimulating factor 1 receptor signaling. Together, these findings provide a spatially aware taxonomic framework for central nervous system immune cells in health and disease.

Innate immune responses in the central nervous system (CNS) are largely executed by microglia and CNS-associated macrophages (CAMs), the principal intrinsic immune cells of the CNS parenchyma and brain interfaces. Changes in these cell types are highly sensitive hallmarks of subtle pathological alterations[1,2], making them potential therapeutic targets across virtually all CNS disorders[3–13]. Recent single-cell studies suggest that different diseases are associated with distinct microglial transcriptional signatures reflecting specific functional programs.

However, it remains unclear whether these transcriptional changes are truly context specific or instead represent a shared spectrum of microglial activation. Although prior work has described heterogeneous ontogenies and activation states in aging and neurodegeneration, whether such states represent broadly conserved programs or disease-specific phenomena remains unresolved, particularly given the limited pathological scope and cross-study heterogeneity of existing reports[14]. This uncertainty is further compounded by potential species differences between human and mouse microglia and by substantial methodological heterogeneity between studies. These include differences in microglial isolation procedures, whole-cell versus nuclear sequencing, genetic mouse strains[3,15], technology platforms (10x

Genomics[16], Smart-Seq2 (ref. 17), mCEL-Seq2 (refs. 4,15), analytical pipelines (RaceID[4], Monocle[18,19], Seurat[20]) and intrinsic confounders such as laboratory-specific gut microbiota[21,22]. In human studies, additional technical variability arises from the use of autopsy versus biopsy material and from fresh versus formalin-fixed tissues. As a result, it remains unresolved whether disease-specific microglial signatures exist in situ. While recent single-cell and single-nucleus studies have provided molecular atlases of the nervous system[23–26], a comprehensive human-mouse spatial atlas of CNS myeloid cells across diverse pathologies has been lacking.

Here, we define the breadth of microglial and other CNS myeloid transcriptional states and compare them across autoimmune, inflammatory, neoplastic, degenerative and dysplastic disease states in humans and mice. We used consistent isolation protocols, identical experimental and computational pipelines and (where applicable) mouse models with matched genetic background and sex. Using targeted and genome-wide subcellular spatial transcriptomics, we characterized local cellular interactomes of transcriptionally and proteomically defined microglial superclusters and clusters, and used cell-specific mutants to assess pathway relevance in microglial states. By applying standardized high-dimensional transcriptomic

✉e-mail: marco.prinz@uniklinik-freiburg.de

and spatial approaches, we systematically surveyed myeloid cells across disease-associated perturbations and derived a transcriptional taxonomy based on hierarchical clustering. Spatial characterization of microglial superclusters revealed distinct neighborhood relationships during perturbations. We propose that microglial responses to homeostatic disruption are governed by a shared transcriptional repertoire organized into a limited number of functional superclusters spanning a wide range of cellular reactions.

## Results

### Superclusters reflect myeloid functional diversity in a human CNS atlas

To establish a pathology-wide census of CNS myeloid cells, we performed single-nucleus RNA sequencing (snRNA-seq) of ~576,000 cells from 212 high-quality human CNS samples derived from 177 individuals spanning 12 disease conditions, as well as non-diseased and fetal brains across up to 10 brain regions (Fig. 1a and Supplementary Fig. 1a). To derive a transcriptome-based hierarchical taxonomy reflecting functional diversity, we applied a graph-based Paris clustering approach, previously shown to generate informative[24,26] dendrogram-based cell taxonomies for CNS cell types.

Following standard Seurat-based integration and graph-based clustering, immune cell populations were identified using canonical marker genes. Microglia expressing *P2RY12*, *TMEM119* and *SALL1* constituted the dominant immune population across conditions, while CAMs expressing *SELENOP* and *FOLR2* were frequently observed. Glioblastoma (GBM), oligodendroglioma, astrocytoma, Aicardi–Goutières syndrome (AGS) and multiple sclerosis (MS) showed enrichment of monocyte-derived macrophages (MDMs) expressing *F13A1* and *TGFBI* (Extended Data Fig. 1a–d). In total, we resolved 192 transcriptionally distinct myeloid clusters, comprising 149 microglia (Mg), 10 CAM and 33 MDM clusters. To structure this diversity, we performed module-based scoring using literature-curated gene sets, grouping clusters into functional superclusters. Because several superclusters were detected under both homeostatic and non-homeostatic conditions, we introduced a condition-level annotation distinguishing 'Homeostasis' from 'Non-homeostasis'. This yielded 14 microglial superclusters (6 homeostatic/non-disease and 8 non-homeostatic), 5 CAM superclusters and 8 MDM superclusters (Fig. 1b and Extended Data Fig. 2). The resulting hierarchical dendrogram depicts relationships between clusters and superclusters and their distribution across conditions.

Microglial superclusters corresponded to eight major functional programs: 'surveillance', 'inflammation', 'proliferation', 'antigen presentation', 'phagocytosis', 'cytokine production', 'neuroprotection' and 'interferon (IFN) signature', consistent with previously described microglial modules[27]. Pairwise Spearman correlation analysis confirmed that these superclusters represent transcriptionally distinct programs akin to higher-order modules that might also be captured by unsupervised approaches such as nonnegative matrix factorization or topic modeling (Fig. 1c,d).

Across all pathologies, multiple microglial superclusters were present simultaneously, indicating a spectrum of activation states rather than discrete disease-specific phenotypes. 'Surveillance' microglia dominated histologically normal tissue but were also present in amyotrophic lateral sclerosis (ALS), normal-appearing white matter (NAWM) in MS, hippocampal sclerosis and focal cortical dysplasia (FCD). In contrast, 'phagocytosis' supercluster microglia were significantly increased in Alzheimer's disease (AD), Parkinson's disease (PD), Huntington's disease (HD) and brain tumors. The 'IFN signature' supercluster was enriched in AGS and fetal brains, while 'Inflammation' supercluster microglia were prominent in hippocampal sclerosis, FCD, oligodendroglioma, astrocytoma, GBM and brain metastasis. Stage-wise analyses revealed progressive diversification of microglial states with disease progression, including early-versus-late AD stages and active-versus-inactive MS lesions (Extended Data Fig. 1e,f).

Non-homeostatic conditions consistently exhibited greater microglial cluster diversity across all brain regions compared to homeostasis (Fig. 1e). This suggests that the ability of microglia to diversify in response to disease is not restricted to a particular CNS region but is a general feature of their activation. *FKBP5* was broadly expressed across activated microglial states (Extended Data Fig. 3a–e). Mg50, Mg52, Mg55 and Mg60 ('surveillance'; *CSF1R*+*P2RY12*+) were detected in multiple sclerosis normal-appearing white matter (MS NAWM), FCD and hippocampal sclerosis. The 'phagocytosis' supercluster comprised 29 clusters, including disease-enriched states such as Mg65 (*CD83*+*HSPH1*+) in GBM, Mg67 (*PDK4*+*LPL*+) and Mg70 (*CD163*+*INPP4B*+) in astrocytoma grade 4 and Mg72 (*ACSL1*+*HMOX1*+) in oligodendroglioma grade 3. Several phagocytosis-associated clusters (Mg66, Mg76, Mg77, Mg82, Mg111, Mg136, Mg140) were shared across neurodegenerative diseases, MS, AGS and GBM. Mg76 showed elevated *APOE* and *SPP1* expression, previously linked to human disease-associated microglia ('DAM')[28–30] and 'MgnD'[31] states, while *APOE*, *TREM2* and *SPP1* were also expressed during human CNS development[6] (Extended Data Fig. 3f). Additional disease-enriched clusters included Mg95 (*GPNMB*+*LGALS3*+) and Mg96 (*CLEC7A*+*ADAMTS2*+) in oligodendroglioma grade 3, Mg113 and

---

**Fig. 1 | Transcriptomic taxonomy of human CNS myeloid cells. a**, Overview of the study design, including data collection and analysis workflow for profiling CNS myeloid cells (microglia, monocytes, CAMs) across multiple conditions and CNS regions. **b**, Paris clustering-based transcriptomic taxonomy tree of CNS myeloid cells organized in a dendrogram (*n* = 128,732 cells) with cell types, conditions, superclusters and clusters. Bar plots show cell distribution across CNS regions and contexts. The color bar represents the Gini coefficient for each cluster. The dot plots show average expression of selected marker genes enriched in the described cell types. AD early, early-onset Alzheimer's disease; AD late, late-onset Alzheimer's disease. **c**, Uniform manifold approximation and projection (UMAP) visualization of 118,762 human microglia colored by superclusters. Each supercluster reflects a distinct gene expression program: 'Surveillance': *OLFML3*, *P2RY12*, *TGFBR1*, *TMEM119*; 'Neuroprotection': *BDNF*, *GDNF*, *IGF1*, *LIF*, *TGFB1*; 'Phagocytosis': *APOE*, *AXL*, *C3AR1*, *CD63*, *CD68*, *CD9*, *CLEC7A*, *LGALS3*, *MYO1E*, *SPP1*, *TREM2*, *TYROBP*; 'Inflammation': *AOAH*, *APOD*, *C5AR1*, *CCL8*, *FGR*, *MSR1*, *RNF169*; 'Cytokine production': *CCL2*, *CXCL10*, *IL10*, *IL1B*, *NFKBIA*, *TNF*; 'Antigen presentation': *B2M*, *HLA-A*, *HLA-DMA*, *HLA-DQA1*, *HLA-DQB1*, *HLA-DRB5*, *NLRC5*; 'IFN signature': *CD69*, *CXCL10*, *IFI16*, *IFIT2*, *IFIT3*, *IFITM3*, *IRF7*, *ISG15*, *MNDA*, *MX1*, *USP18*; 'Proliferation': *BRIP1*, *RAD51B*, *TOP2A*. **d**, Heat map showing similarity between microglia superclusters based on average gene expression profiles. Colors represent *z*-scores of average gene expression. The color bars correspond to the superclusters shown in **b**. Pairwise similarity was quantified using Spearman rank correlation coefficients. Statistical significance was assessed using two-sided Spearman correlation tests. *P* values are unadjusted; ***$P < 0.001$. **e**, Microglial cluster diversity across CNS regions and conditions. Dot plot showing the number of microglial clusters (dot size) and nuclei analyzed (dot color scale) per brain region in homeostatic versus non-homeostatic contexts. Differences in cluster counts were modeled using generalized linear models (GLMs; Poisson or negative binomial, log link) with nuclei count included as a covariate to adjust for sampling depth and region or patient included as random effects where indicated. Statistical significance was assessed using two-sided Wald tests of the condition coefficient. As a single global contrast was tested, no multiple-comparison correction was applied. Non-homeostasis was associated with increased cluster diversity (rate ratio of 1.46, $P = 1.37 \times 10^{-7}$). **f**, Spatial proteomics of selected microglial clusters by high-dimensional IMC. Upper: representative IMC image showing IBA1 (red), HLA-DR (turquoise), HLA-DPA1 (yellow), HLA-DRA (greenish cyan), CD68 (light blue), CD11c (mint), P2RY12 (lemon) and TMEM119 (greenish-yellow) expression in cells representing microglial cluster 33 in an MS lesion. Collagen IV (green) highlights blood vessels. Arrowheads highlight microglia. Asterisks denote other myeloid cells. DNA is shown in blue. Upper panel: representative IMC image showing IBA1 (red), P2RY12 (lemon) and TMEM119 (greenish-yellow) expression in cells representing homeostatic microglia in control cortex. Scale bars, 20 µm. Panel **a** created in BioRender; Dumas, A. https://biorender.com/jmh4pys (2026).

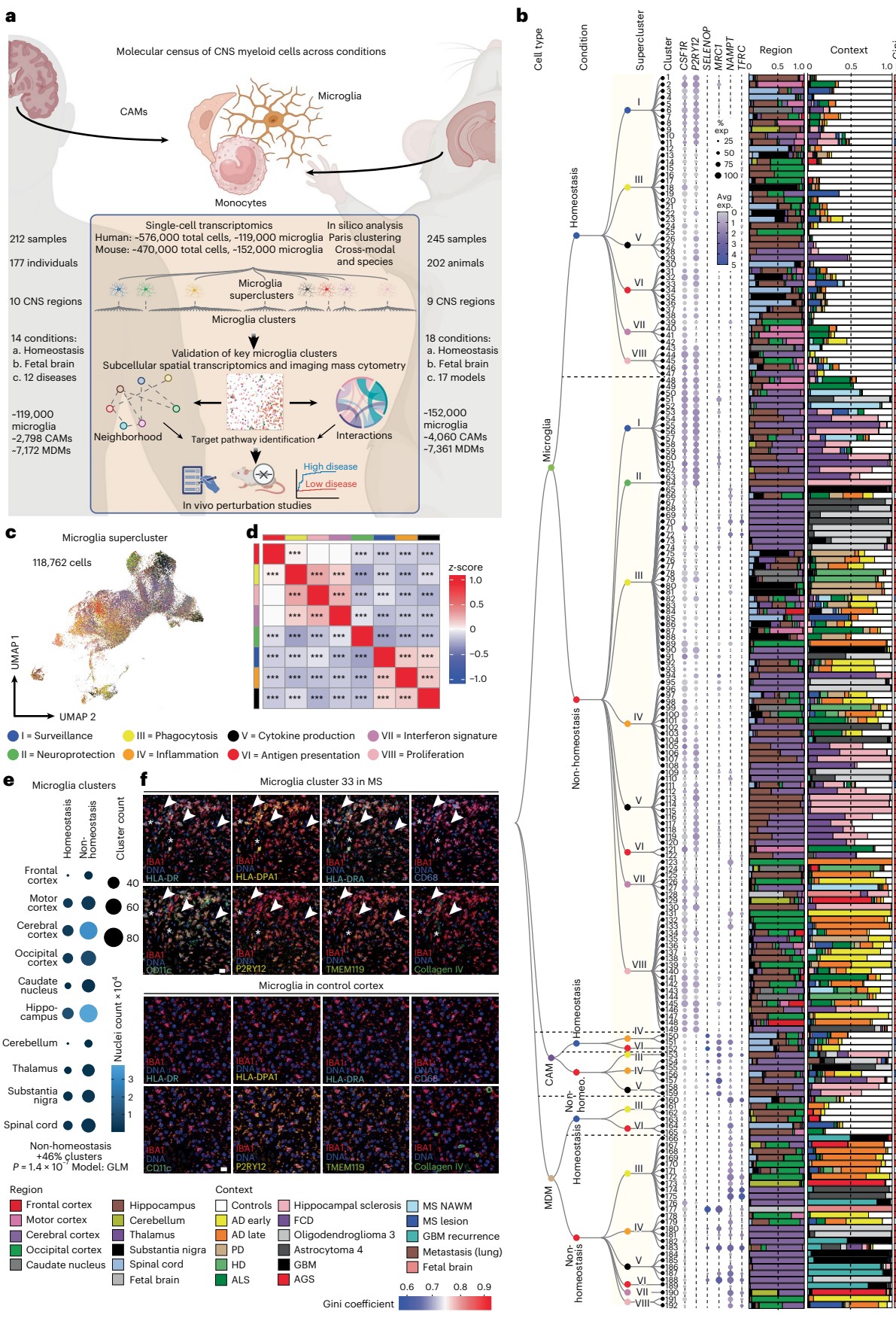

Mg118–Mg120 ($CCL2^+CCL3^+TNF^+IL1B^+$) in FCD and Mg129 ($MX1^+IFI44^+$) in AGS.

Protein-level validation using imaging mass cytometry (IMC) confirmed the presence of transcriptionally defined clusters in situ. Mg33 ($HLA\text{-}DPA1^+HLA\text{-}DRA^+P2RY12^+TMEM119^+$) was detected in an active medullary section from an individual with MS but absent in controls (Fig. 1f), while Mg129 ($MX1^+Iba1^+SLC2A5^+P2RY12^+TMEM119^+$) microglia were selectively enriched in AGS cerebellum (Extended Data Fig. 3g).

In summary, we used graph-based clustering of snRNA-seq data to define a new, hierarchical myeloid cell taxonomy across diverse contexts in the human CNS, allowing the definition of several overarching microglia/CAM/MDM superclusters with annotated functions. We further identified a broad transcriptional spectrum of additional microglial clusters that formed the next level of this taxonomical tree across diverse human CNS pathologies.

## Spatial single-cell microglial clusters in human CNS pathologies

To map the spatial organization of transcriptionally defined microglial superclusters and clusters relative to disease-associated histopathology, we performed high-resolution confocal microscopy combined with subcellular spatial transcriptomics using a myeloid-focused 1,000-plex Nanostring CosMx panel (Fig. 2a and Supplementary Table 4).

We analyzed five representative human pathologies (AD, AGS, FCD, active MS and GBM) enriched for distinct microglial clusters. Spatial transcriptomic cell-type annotation and UMAP visualization resolved neural, vascular and immune populations, including resident and infiltrating immune cells, across disease contexts (Fig. 2b and Extended Data Fig. 4a,b). Disease-specific alterations in immune cell composition were readily apparent (Fig. 2c).

Mapping transcriptionally defined microglial clusters in situ revealed distinct spatial distributions. In control tissue, $P2RY12^+$ Mg4, Mg18 and Mg30 were evenly distributed (Fig. 2d). In AD, Mg82 ($LGALS9^+C1QC^+$) and Mg89 ($ITGAX^+$), both belonging to the 'phagocytosis' supercluster, as well as Mg127 ($ITGAX^+IFIT^+$; 'IFN signature'), localized in close proximity to amyloid deposits. In FCD, Mg113, Mg118 and Mg119 ($CCL2^+CCL3^+CCL4^+IL1B^+$; 'cytokine production') clustered near pathognomonic dysmorphic neurons. In AGS, interferon supercluster microglia (Mg121, Mg123; $IFITM1^+OAS2^+$) were more diffusely distributed. In MS, tissues adjacent to demyelinating lesion rims were enriched for $SPP1^+HLA\text{-}DPA1^+$ Mg30, Mg33 and Mg34, indicative of antigen-presenting microglia. Comparative analysis across pathologies revealed enrichment of the 'antigen presentation' supercluster in MS lesions, rims and NAWM, robust induction of 'phagocytosis' and 'IFN signature' superclusters in AGS and preferential enrichment of 'cytokine production' and 'antigen presentation' superclusters in FCD (Fig. 2e).

We next examined regional variation in microglial superclusters across neurodegenerative and inflammatory conditions. The 'surveillance' supercluster predominated across control regions but was consistently reduced in disease. Homeostatic cortex exhibited partial representation of the 'cytokine production' supercluster, while homeostatic spinal cord was enriched in 'antigen presentation' and 'phagocytosis' superclusters. Disease-specific regional remodeling was evident: AD showed expansion of 'phagocytosis' in hippocampus and 'proliferation' superclusters in occipital cortex; PD showed enrichment of 'proliferation' in thalamus and 'phagocytosis' superclusters in substantia nigra; ALS exhibited expansion of 'antigen presentation' in motor cortex and marked 'proliferation' in hippocampus. In AGS, the cerebellum was dominated by the 'IFN signature' supercluster, whereas cortex showed increased 'phagocytosis'. MS lesions displayed broad induction of multiple superclusters alongside depletion of 'surveillance' microglia relative to NAWM and controls (Fig. 2f).

Together, targeted subcellular spatial transcriptomics revealed that transcriptionally defined microglial superclusters and clusters occupy distinct pathological niches in situ, exhibiting disease-specific and region-specific spatial organization within affected CNS tissues.

## Dynamic spatial interactions of microglia in human CNS disease

To characterize how microglial superclusters integrate into local cellular networks, we analyzed spatial distributions of all cell types within CosMx spatial transcriptomics data and performed neighborhood interaction analyses (Fig. 3a and Extended Data Fig. 4c). In control human tissue, neurons and astrocytes occupied central hub positions within the cellular interaction network, consistent with their roles in tissue homeostasis, while microglia localized predominantly to peripheral positions, indicative of a surveillant, weakly interactive state (Fig. 3b and Extended Data Fig. 4d). In AD, the overall network structure remained largely preserved, although clusters belonging to the 'phagocytosis' supercluster (Mg19, Mg82, Mg89) became more prominent.

In contrast, pronounced network reorganization was observed in AGS, MS and FCD. In MS, microglia associated with the 'antigen presentation' supercluster shifted toward the network center and exhibited strong interactions with peripheral immune populations, including $CD8^+$ tissue-resident memory T cells and plasmablasts. In FCD, microglia of the 'cytokine production' supercluster preferentially interacted with nonimmune cells such as neurons and astrocytes, reflecting disease-specific modes of neuroimmune cross-talk. These findings indicate that microglial interaction networks and partners vary according to pathological context.

To visualize spatial redistribution of functional microglial states, we mapped supercluster-specific transcriptional signatures across tissues (Fig. 3c). In MS, 'antigen presentation' and 'phagocytosis' superclusters accumulated preferentially in myelinated regions at lesion rims and within NAWM, whereas lesion cores showed comparatively reduced supercluster diversity. Together, these analyses demonstrate that microglial superclusters and their constituent

---

**Fig. 2 | Targeted subcellular spatial transcriptomics of microglial states.**
**a**, Schematic diagram depicting single-cell spatial transcriptomic analysis (CosMx technology), strategy for microglial cluster imputation based on gene expression and cellular neighborhood analysis of identified clusters from diseased and control samples. DEGs, differentially expressed genes. **b**, UMAP visualization of 68,327 single-cell transcriptional states based on spatial transcriptomic data. Colors in the UMAP represent different cell types. Oligo, oligodendrocyte; OPC, oligodendrocyte precursor cell; $T_{reg}$, regulatory T cell; $T_{RM}$, resident memory T cell; DC, dendritic cell; Fib., fibroblast; SMC, smooth muscle cell. **c**, Bar plots represent immune cell proportion across diverse contexts. Ctrl, control; Les., lesion. Colors represent individual immune cell types. Dashed line represents the proportion of 0.5. One-sided hypergeometric test with Benjamini–Hochberg correction; ***$P < 0.001$, **$P < 0.01$. **d**, Left: representative immunohistochemistry showing topical disease pathology of the investigated samples ($n = 6$ for control, $n = 3$ for AD, MS, GBM, $n = 2$ for AGS,

FCD; $n = 2$). Scale bars, 25 µm. Middle: Illustrative samples of spatial distribution of microglia (red) across specified contexts, with zoom-in. Mg, microglial cluster. Circle with dashed line marks amyloid plaque-containing area in AD. White dashed line indicates the boundary between demyelinated (left, labeled as DE) and myelinated (right) white matter. Right: imputed microglial states, with colored dots inside individual cells indicating transcripts of selected genes. Heat maps in each panel depict $z$-scored average gene expression of selected microglial cluster-enriched marker genes. **e**, Bar plots indicating the proportion Mg superclusters across contexts. Dashed line represents the proportion of 0.5. One-sided hypergeometric test with Benjamini–Hochberg correction; ***$P < 0.001$, **$P < 0.01$, *$P < 0.05$. **f**, Bar plots representing regional distribution of Mg superclusters in different contexts. One-sided hypergeometric test with Benjamini–Hochberg correction; ***$P < 0.001$, *$P < 0.05$. Panel **a** created in BioRender; Dumas, A. https://biorender.com/jmh4pys (2026).

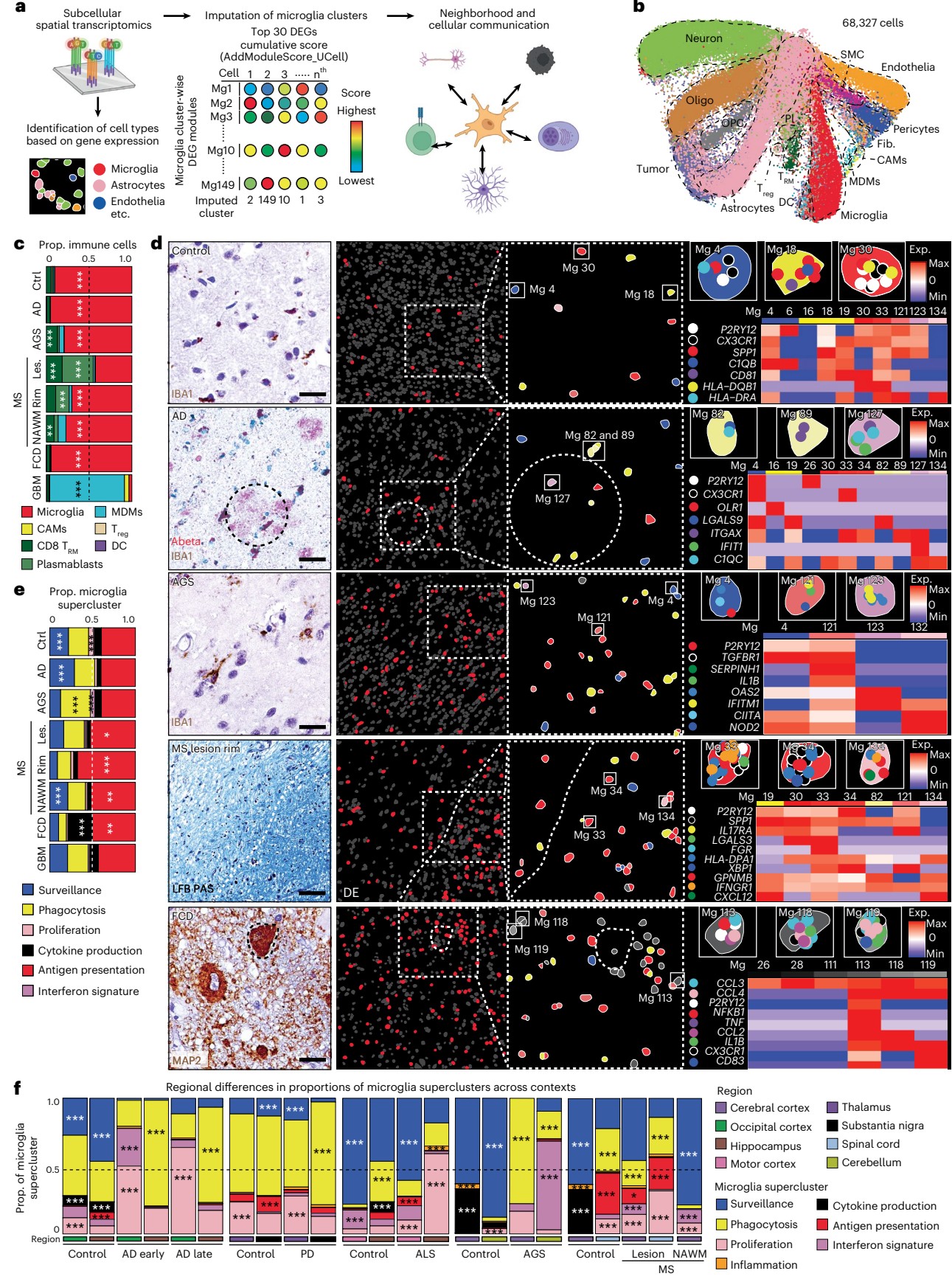

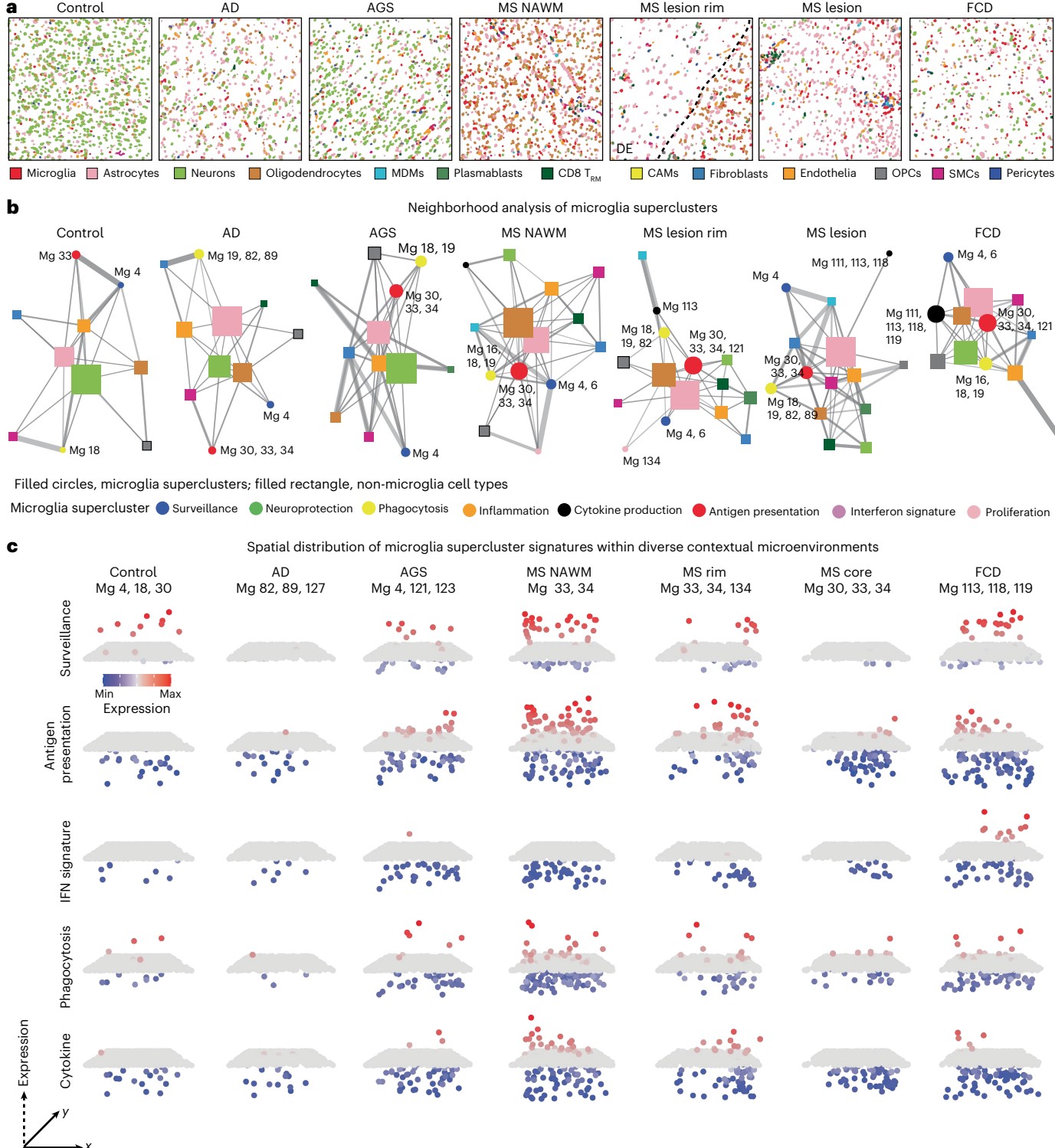

**Fig. 3 | Disease-specific cellular neighborhoods of human microglia.**
**a**, Representative spatial maps of CosMx spatial transcriptomics data displaying individual cells colored as cell types as defined in Fig. 2b. **b**, Network diagrams illustrating the interaction network of microglial superclusters (filled circles) with other cell types (filled rectangles, colored as in **a**). Microglial clusters belonging to the supercluster are indicated. The node size corresponds to the total number of cells of that type in the representative images, while the edge width reflects the proportion of neighboring cells between connected nodes. **c**, Three-dimensional dot plots depicting z-scored expression of selected microglial superclusters (row wise) based on individual microglia belonging to the shown cluster (Mg) from **a**. Arrows indicate the axis direction in the plot, with the dashed arrow representing the axis for z-scored expression of genes building the displayed supercluster. Height and color are redundant encodings of the same z-score statistic.

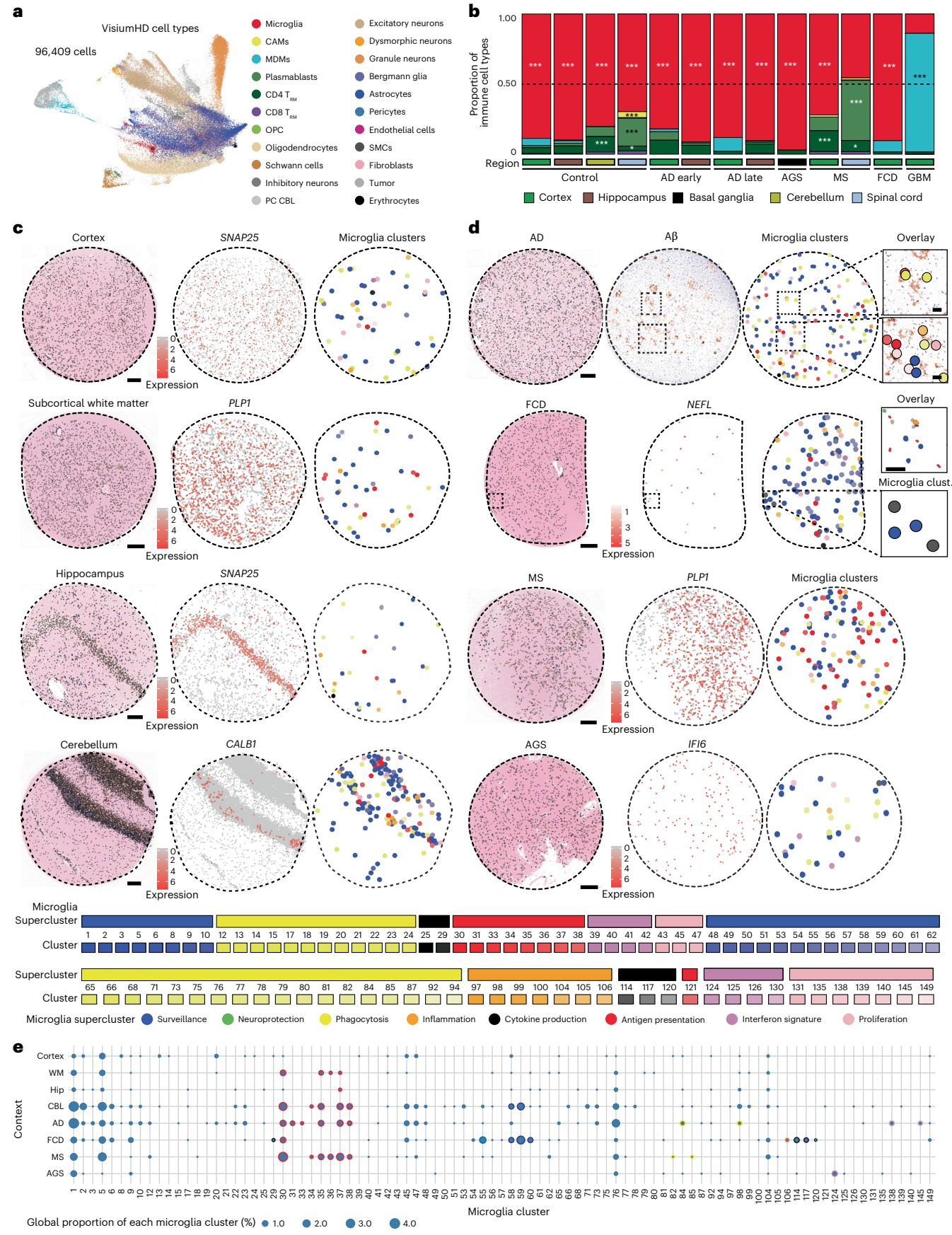

**Fig. 4 | Genome-wide subcellular spatial transcriptomics of microglia in CNS pathology. a**, UMAP visualization of 96,409 single-cell transcriptional states based on Visium HD spatial transcriptomic data. Each color represents a distinct cell type. **b**, Bar plots showing proportions of immune cell types across different CNS regions and disease contexts. Colors in the plot represent individual immune cell types as indicated in **a**. Dashed line represents the proportion of 0.5. One-sided hypergeometric test with Benjamini–Hochberg correction; ***$P < 0.001$, *$P < 0.05$. **c**, Left: high-resolution H&E images overlaid with spatially resolved single-cell identities. Middle: spatial expression maps of key marker genes. Right: spatial distribution of microglial clusters. Shades of colors represent the respective superclusters. Scale bars, 200 μm **d**, Left: high-resolution H&E images overlaid with spatially resolved single-cell identities.

Middle: AD panels show amyloid beta immunohistochemistry, while other diseases display spatial expression of pathology-associated genes. Right: spatial plots of microglial clusters. Shades of colors represent the respective superclusters. In AD, microglial superclusters localize near amyloid plaques. Zoom-in shows overlay of amyloid beta and respective microglial clusters. In FCD, microglial cluster adjacent to dysmorphic neurons. Zoom-in shows overlay of H&E and linked microglial clusters amyloid beta. Scale bars, 200 μm (overviews) and 50 μm (zoom-in). **e**, Dot plot representing global proportions of microglial clusters from representative samples. Dots with highlighted borders represent the microglial clusters enriched in the specific contexts in snRNA-seq data (Fig. 1a). The color of the border represents the supercluster as indicated in **c**.

clusters are not uniformly distributed but instead exhibit marked disease-specific and region-specific spatial and interaction dynamics across CNS pathologies.

## Genome-wide subcellular spatial mapping of microglial states

To unbiasedly resolve microglial superclusters and clusters at subcellular resolution, we applied whole-transcriptome spatial profiling using Visium HD. Binned transcriptomic data (2 μm) were overlaid onto high-resolution hematoxylin and eosin (H&E)-stained images, enabling reconstruction of gene expression profiles for individual nuclei. Single-cell expression data from six CNS regions and five diseases (AD, AGS, MS, FCD, GBM) were jointly analyzed, comprising ~96,000 cells and ~18,000 genes (Fig. 4a and Extended Data Fig. 5a).

This approach resolved diverse CNS cell populations, including microglia, CAMs, MDMs, lymphocytes, vascular cells, neurons, glia and region-specific populations such as Bergmann glia (*GRIA1⁺DAO⁺*) and Purkinje neurons (*PVALB⁺BHLHE22⁺HOMER3⁺*; Fig. 4a and Extended Data Fig. 5b–d). Consistent with CosMx data, immune cell composition varied across diseases, with microglia representing the dominant immune population and MS displaying the highest proportions of tissue-resident memory T cells and plasmablasts (Fig. 4b). In control tissue, microglia were sparsely and evenly distributed across CNS regions, whereas disease states showed increased microglial abundance, particularly in AD, MS and FCD (Fig. 4c,d and Extended Data Fig. 5c). As observed in snRNA-seq analyses and other studies, control spinal cord was enriched for the 'antigen presentation' supercluster relative to cortex[32] (Fig. 4c and Extended Data Fig. 5c,e,f). Disease-specific remodeling of microglial superclusters was evident, including increased 'phagocytosis' and 'proliferation' superclusters in AD hippocampus, 'IFN signature' supercluster in AGS basal ganglia,

'antigen presentation' supercluster in MS cortex and spinal cord, and combined 'cytokine production' and 'surveillance' superclusters in FCD cortex (Fig. 4d and Extended Data Fig. 5e,f).

Spatial mapping revealed close association of discrete microglial clusters with local pathological features. In AD, multiple Mg clusters localized near amyloid-β plaques (Fig. 4d,e). In FCD, Mg58–Mg60 ('surveillance' supercluster) were preferentially found in regions lacking dysmorphic neurons, whereas Mg29, Mg114 and Mg117 ('cytokine production' supercluster) accumulated near *NEFL*-expressing dysmorphic neurons (Fig. 4d,e and Extended Data Fig. 5e,g). Together, unbiased whole-genome subcellular spatial transcriptomics enabled high-resolution mapping of disease-associated microglial clusters in situ, revealing a rich diversity of local microglial states organized into functional superclusters.

## Mouse myeloid taxonomy reveals superclusters conserved in humans

To extend the analysis of microglial heterogeneity to mouse models, we applied the same experimental and analytical framework used for human tissue to mouse CNS samples. In total, ~470,000 cells from 245 high-quality samples spanning 18 disease models, healthy adult brain and development were profiled, yielding ~172,000 immune cells, including >151,000 microglia (Figs. 1a and 5a and Supplementary Fig. 1b).

Murine microglia were defined based on expression of canonical markers including *Hexb* and *P2ry12*, while CAMs, MDMs, lymphocytes and dendritic cells were detected in context-dependent proportions (Extended Data Fig. 6a–c). Within the murine myeloid taxonomy, we resolved 78 transcriptionally distinct clusters, comprising 54 microglia, 6 CAM and 18 MDM clusters. Using the same module-scoring strategy applied to human microglia, we grouped clusters into functional

**Fig. 5 | Transcriptomic taxonomy of mouse CNS myeloid cells across disease models. a**, Transcriptome-based taxonomy tree of mouse CNS myeloid cells organized in a dendrogram ($n = 163,152$ cells) with cell types, conditions, superclusters and clusters labeled. Bars show cell distribution across CNS regions and contexts. The color bar represents the Gini coefficient for each cluster. The dot plots display average expression of selected marker genes enriched in the described cell types. 5xFAD, familial Alzheimer's disease model; Aging, 24-month-old mice; APP23, amyloid precursor protein 23 model; CC, corpus callosum; Cup, cuprizone model; E16, embryonic day 16; FNX, facial nerve axotomy; GF, germ free; GL261, glioma model; OB, olfactory bulb; R6/2, Huntington's disease model; Rnase-2, *Rnase-2⁻ᐟ⁻* model; μMT, *μMT⁻ᐟ⁻*. **b**, UMAP visualization of 151,731 microglia colored by supercluster. Each supercluster reflects its own gene expression program: 'Surveillance': *Hexb, Olfml3, P2ry12, Siglech, Sparc, Tgfbr1, Tmem119*; 'Neuroprotection': *Bdnf, Gdnf, Igf1, Lif, Tgfb1*; 'Phagocytosis': *Apoe, Axl, C3ar1, Cd9, Cd63, Cd68, Clec7a, Lgals3, Myo1e, Spp1, Trem2, Tyrobp*; 'Inflammation': *Apod, Aoah, C5ar1, Ccl12, Fgr, Msr1, Rnf169*; 'Cytokine production': *Ccl2, Cxcl10, Il10, Il1b, Nfkbia, Tnf*; 'Antigen presentation': *B2m, H2-D1, H2-Aa, H2-Ab1, H2-DMa, H2-Eb1, H2-K1, Nlrc5*; 'IFN': *Cd69, Cxcl10, Ifi204, Ifi213, Ifit2, Ifit3, Ifitm3, Irf7, Isg15, Mx1, Oasl2, Usp18*; 'Proliferation': *Birc5, Brip1, Mcm5, Mki67, Rad51b, Top2a*. **c**, Heat map showing similarity between Mg superclusters based on average gene expression profiles. Colors

represent $z$-scores of average gene expression. The color bars correspond to the superclusters shown in **a**. Pairwise similarity was quantified using Spearman rank correlation coefficients. Statistical significance was assessed using two-sided Spearman correlation tests. $P$ values are unadjusted; **$P < 0.01$, ***$P < 0.001$. **d**, Microglial cluster diversity across mouse CNS regions and conditions. Dot plot showing the number of microglial clusters (dot size) and nuclei analyzed (dot color scale) per brain region in homeostatic versus non-homeostatic contexts. Differences in cluster counts were modeled using a Poisson GLM, with nuclei count included as a covariate to adjust for sampling depth and brain region modeled as a random effect in a generalized linear mixed model. Statistical significance was assessed using two-sided Wald tests. As a single global contrast was tested, no multiple-comparison correction was applied. Non-homeostasis was associated with increased cluster diversity (rate ratio of 8.11, $P = 3.29 \times 10^{-20}$). **e**, Upper: representative immunofluorescence of CD74 (red) enriched in TMEM119⁺ microglial cluster 40 (green) and DAPI (blue) and quantification thereof below. Kruskal–Wallis test, $P < 0.0001$; Dunn's multiple-comparisons test for individual comparisons, Scale bars, 20 μm. Lower: RNAscope image of *Msr1* (red) enriched in *Hexb⁺* microglial clusters 37 and 38 (green), and DAPI (blue) and quantification thereof (below). Kruskal–Wallis test, $P < 0.0001$; Dunn's multiple-comparisons test for individual comparisons. *$P < 0.05$, ***$P < 0.001$. Scale bars, 20 μm.

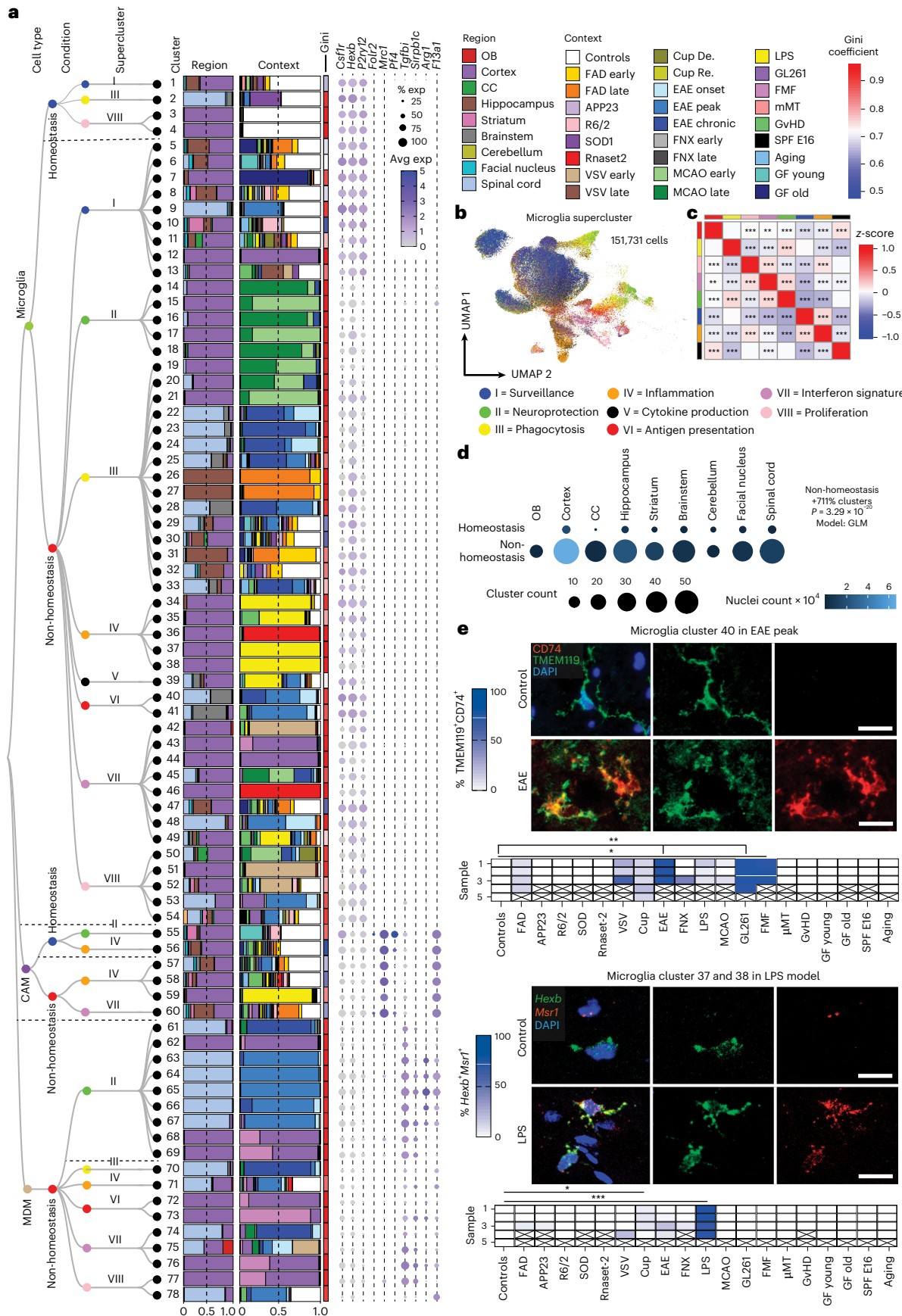

superclusters, revealing striking parallels between mouse and human microglial organization across disease contexts. In contrast to humans, murine homeostatic microglia formed fewer superclusters, indicating limited transcriptional diversity in the healthy mouse brain. Overall, we identified 11 microglial superclusters (3 homeostatic/non-disease and 8 non-homeostatic/disease), 4 CAM superclusters (2 homeostatic and 2 non-homeostatic) and 6 MDM superclusters, predominantly observed under non-homeostatic conditions. Hierarchical dendrograms depict relationships between mouse myeloid clusters, superclusters and their distribution across conditions (Fig. 5a and Extended Data Fig. 7).

Consistent with human data, murine microglial superclusters were categorized as 'surveillance', 'neuroprotection', 'phagocytosis', 'inflammation', 'cytokine production', 'antigen presentation', 'interferon signature' and 'proliferation'. While these transcriptional co-modules were conserved across species, their relative balance differed, indicating partially distinct structuring of protective and inflammatory programs in murine microglia (Fig. 5b,c). As in humans, murine microglia diversified broadly in non-homeostatic conditions across all tested brain regions, with a substantially greater increase in cluster number than observed in the human CNS (Fig. 5d).

The 'surveillance' supercluster predominated in homeostasis but was also detected at lower levels in several disease models. In contrast, the 'phagocytosis' supercluster was enriched in familial Alzheimer's disease (FAD), as well as superoxide dismutase-1 (SOD1), cuprizone-induced demyelination, middle cerebral artery occlusion (MCAO) and experimental autoimmune encephalomyelitis (EAE) at all clinical stages. The 'interferon signature' supercluster was prominent in $Rnaset2^{-/-}$, vesicular stomatitis virus (VSV) encephalitis, EAE onset, GL261 glioma and familial Mediterranean fever (FMF). Elevated proportions of the 'inflammation' supercluster were observed in $Rnaset2^{-/-}$, lipopolysaccharide (LPS) and graft-versus-host disease (GvHD) models. Stage-wise analyses revealed dynamic remodeling of microglial states during disease progression, including early-to-late transitions in FAD, VSV, MCAO and cuprizone demyelination/remyelination, as well as progressive increases in phagocytosis-associated states from EAE onset to chronic phases (Extended Data Fig. 6d,e).

Quantitative analysis of cluster distributions revealed limited disease specificity but frequent enrichment across conditions. Mg29 ($P2ry12^+Trem2^+Clec7a^+$) and Mg30 ($P2ry12^+Trem2^+Ctsb^+$), both linked to the 'phagocytosis' supercluster, as well as Mg47 ($Mx1^+Ifit3^+$) from the 'interferon signature' supercluster, were shared across multiple disease models (Fig. 5a). Mg29 and Mg30 additionally expressed $Tyrobp$, $Apoe$ and $Csf1$, consistent with a DAM-like transcriptional program present in FAD, cuprizone demyelination, EAE, GvHD, aging and germ-free housing. Additional context-enriched clusters included Mg40 ($Cd74^+H2\text{-}K1^+H2\text{-}D1^+$) in EAE, GL261, and FMF Mg50 ($Top2a^+Mki67^+$) representing the 'proliferation' supercluster in cuprizone, VSV, GL261, early MCAO and EAE; Mg17 ($Myo1e^+Xdh^+Sash1^+$) selectively enriched in MCAO and linked to 'neuroprotection'; and Mg37/Mg38 ($Tnf^+Aoah^+Msr1^+$) almost exclusively detected in the LPS model (Fig. 5a and Extended Data Fig. 8a,b).

Protein-level and RNA-level validation confirmed transcriptionally defined clusters in situ. Double immunolabeling for CD74 and TMEM119 identified $CD74^+TMEM119^+DAPI^+$ microglia corresponding to Mg40 in EAE, GL261 and FMF, with statistical enrichment restricted to these models. RNAscope analysis validated enrichment

of $Msr1^+Hexb^+$ microglia corresponding to Mg37/Mg38 in the LPS model (Fig. 5e). Collectively, these data define a murine CNS myeloid taxonomy that mirrors the human hierarchy, revealing conserved superclusters and a restricted set of microglial states redistributed across pathological contexts.

## Cross-species integration shows conserved microglial superclusters

To directly compare microglial activation states between species, we integrated all human and mouse microglia snRNA-seq data using homologous genes and reciprocal principal component analysis (PCA; Fig. 6a). All major superclusters were preserved upon integration. Within the shared transcriptional cross-species manifold, human microglia showed increased representation of the 'cytokine production' supercluster, whereas mouse microglia exhibited higher proportions of 'inflammation' and 'neuroprotection' superclusters across conditions, suggesting partial overlap and species-enriched regions (Fig. 6b,c).

Supercluster-level correlations revealed conservation of 'surveillance', 'antigen presentation', 'phagocytosis' and 'IFN signature' programs between species (Fig. 6d). At the cluster level, human cytokine-producing clusters aligned weakly with murine counterparts, whereas mouse clusters from the 'neuroprotection' and 'phagocytosis' superclusters aligned strongly with human 'phagocytosis' clusters derived from neurodegenerative disease and corresponding mouse models (Fig. 6e). These relationships remained robust even after under gene resampling (Supplementary Fig. 2).

Focused analysis of individual superclusters revealed context-dependent conservation. Mouse phagocytosis-associated clusters from EAE and FAD mapped to human clusters in MS and AD, respectively, with greater concordance for EAE–MS than FAD–AD comparisons (Extended Data Fig. 9a,b). Antigen-presentation superclusters showed fewer shared clusters between mouse and human, with human Mg33 and Mg36 showing highest similarity to mouse Mg40 and Mg41 in EAE–MS comparisons (Extended Data Fig. 9c,d). Interferon signature clusters from $Rnaset2^{-/-}$ mice aligned closely with human AGS microglia (Extended Data Fig. 9e,f). Together, these data demonstrate broad conservation of functional microglial superclusters alongside species-specific and context-specific cluster implementations.

## CSF-1R and interferon signaling shape microglial superclusters

To identify molecular drivers of supercluster induction, we interrogated signaling pathways underlying microglial state transitions. In the cuprizone model, ligand–receptor analysis identified robust activation of colony-stimulating factor 1 (CSF-1)–colony-stimulating factor 1 receptor (CSF-1R) signaling between oligodendrocytes and microglia (Fig. 7a). Low-dose small-molecule CSF-1R inhibitor (sCSF-1R$_{inh}$)[33,34] inhibition preserved microglial viability[34] and morphology while inducing limited transcriptional changes, with modest upregulation of $Apoe$ reaching significance (Fig. 7b–d). During EAE, CSF-1R inhibition increased Mg40 associated with the 'antigen presentation' supercluster and reduced Mg14, Mg15 and Mg18 linked to 'neuroprotection' (Fig. 7e,f). In contrast, LPS-induced Mg34, Mg35, Mg37 and Mg38 ('inflammation') were abolished by CSF-1R inhibition, while Mg14 and Mg15 ('neuroprotection') were increased (Extended Data Fig. 10d,e).

**Fig. 6 | Conserved and divergent microglial states across species. a**, Schematic of the integration workflow: mouse and human microglia were mapped to one to many and integrated using reciprocal PCA (RPCA). **b**, Joint UMAP of 270,493 cells showing co-clustering between human (pink) and mouse (green) microglia. **c**, UMAP showing the supercluster assignment of microglia within each species. Colors represent the individual superclusters. **d**, Heat maps depicting similarity between human and mouse microglial superclusters based on average gene expression profiles. Pairwise similarity was quantified using Pearson correlation

coefficients. Statistical significance was assessed using two-sided Pearson correlation tests. $P$ values are unadjusted; *$P < 0.01$, **$P < 0.001$, ***$P < 0.0001$. **e**, Cluster-level Pearson correlations between species. Bars indicate the disease/ model composition contributing to each cluster with significance by two-sided Pearson test. MHC, major histocompatibility complex. *$P < 0.01$, **$P < 0.001$, ***$P < 0.0001$. Maps and charts in **c**–**e** created in BioRender; Dumas, A. https://biorender.com/jmh4pys (2026).

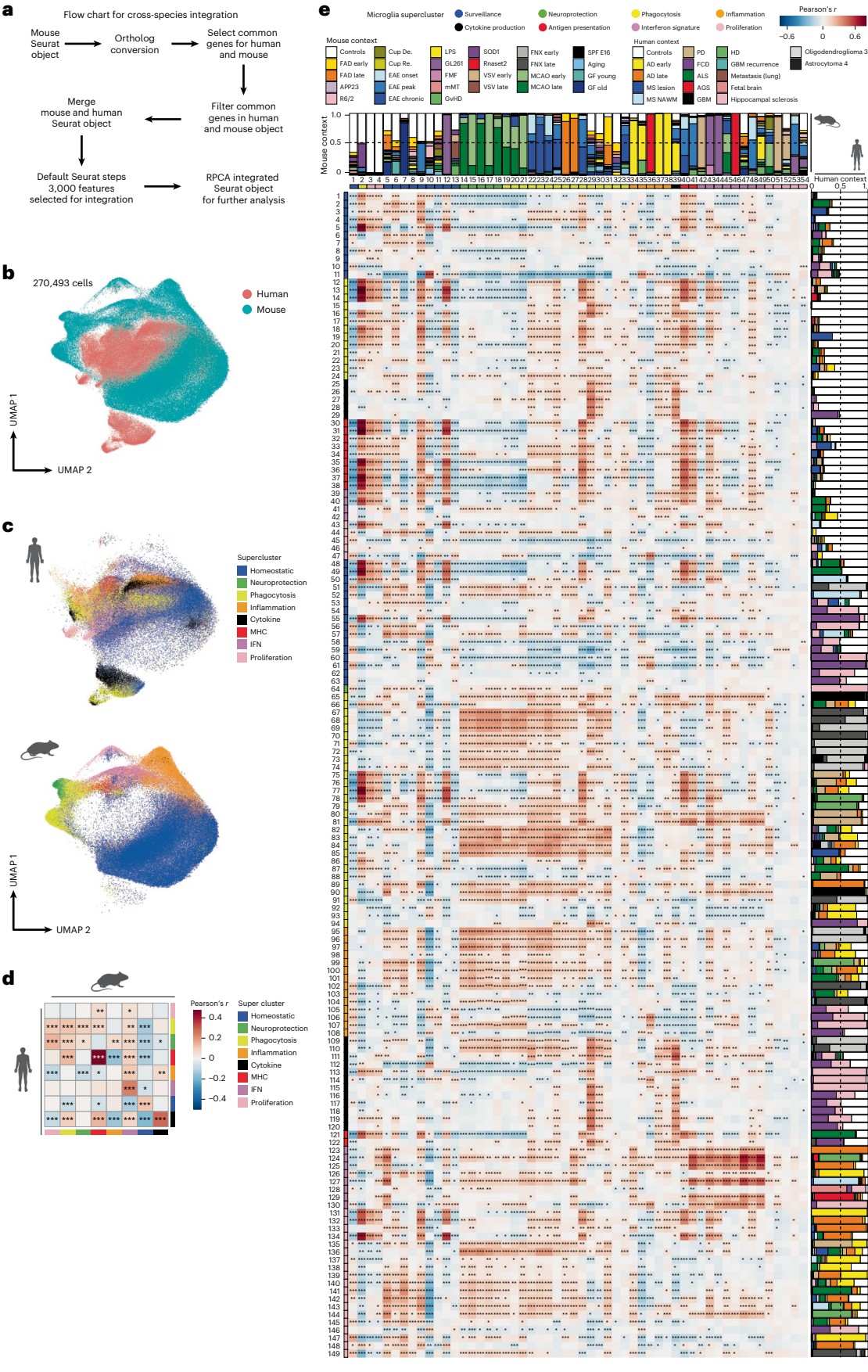

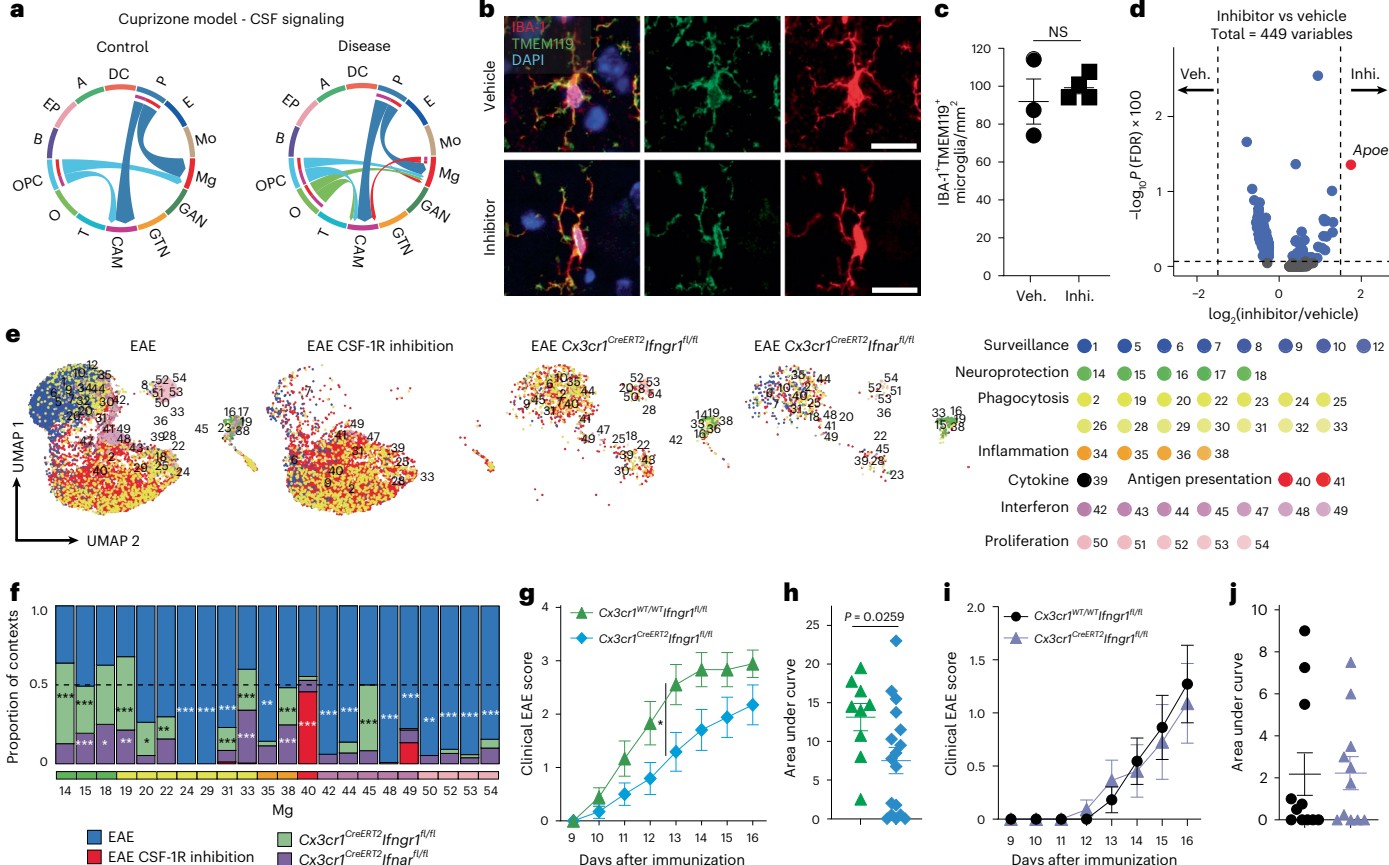

**Fig. 7 | CSF-1R and interferon signaling shape microglial state. a**, Inferred CSF signaling network between CNS cells during cuprizone-mediated toxic demyelination (disease) and respective control from the cuprizone model data in Fig. 5. Colors depict different cell types. GAN, GABAergic neuron; GTN, glutamatergic neuron; T, T cells; O, oligodendrocytes; B, B cells; EPs, epithelial cells; A, astrocytes; P, pericytes; E, endothelial cells; Mo, monocytes. **b**, Representative immunofluorescence image for TMEM119 (green), IBA1 (red) and DAPI (blue) in brains of the vehicle-treated or CSF-1R inhibitor-treated mice. Scale bars, 20 μm. **c**, Quantification of IBA1$^+$TMEM119$^+$ microglia from vehicle (Veh.; $n = 3$) or CSF-1R inhibitor (Inhi.; $n = 5$)-treated mice. Data represent the mean ± s.e.m. One symbol represents one animal. NS, not significant. **d**, Volcano plot showing differential gene expression between CSF-1R inhibitor-treated and vehicle-treated microglia. The log$_2$ fold change in expression ($x$ axis) is plotted against the −log$_{10}$-transformed Benjamini–Hochberg-adjusted $P$ value ($y$ axis). Differential expression was assessed using two-sided Wilcoxon rank-sum tests implemented in Seurat (FindMarkers), with false discovery rate (FDR) control by Benjamini–Hochberg correction. Dashed lines indicate fold change

and adjusted $P$-value thresholds. **e**, UMAP visualization of single microglial clusters colored by contexts. Mice underwent either vehicle or CSF-1R inhibitor treatment or for a genetic deletion of IFNAR or IFNGR. **f**, Overall proportions of significantly altered microglial clusters across different conditions from **e**. Dashed line represents proportion of 0.5. One-sided hypergeometric test with Benjamini–Hochberg correction; *$P < 0.05$, **$P < 0.01$, ***$P < 0.001$. **g**, Clinical EAE course in $Cx3cr1^{CreERT2}Ifngr1^{fl/fl}$ mice. Each data point represents the mean ± s.e.m. of 26 mice pooled from two independent experiments ($Cx3cr1^{WT/WT}Ifngr1^{fl/fl}$ = 9; $Cx3cr1^{CreERT2}Ifngr1^{fl/fl}$ = 17). Disease scores over time were analyzed using a repeated-measures mixed-effects analysis of variance (ANOVA) with genotype and time as factors, followed by two-sided Sidak's multiple-comparisons tests; *$P = 0.0357$. **h**, Area under the curve for clinical EAE course in $Cx3cr1^{CreERT2}Ifngr1^{fl/fl}$ mice in **g**. Groups were compared using a two-sided Mann–Whitney test; $P = 0.0259$. **i**, Clinical EAE course in $Cx3cr1^{CreERT2}Ifnar^{fl/fl}$ mice. Each data point represents the mean ± s.e.m. of 22 mice in total ($Cx3cr1^{WT/WT}Ifnar^{fl/fl}$ = 11; $Cx3cr1^{CreERT2}Ifnar^{fl/fl}$ = 11) pooled from two independent experiments. **j**, Area under the curve for clinical EAE course in $Cx3cr1^{CreERT2}Ifnar^{fl/fl}$ mice in **i**.

Given the conserved 'interferon signature' supercluster, we next assessed the roles of type I and II interferon signaling using inducible long-lived CNS myeloid cell-targeted deletion of IFNAR or IFNGR ($Cx3cr1^{CreERT2}Ifnar^{fl/fl}$, $Cx3cr1^{CreERT2}Ifngr1^{fl/fl}$). In EAE, IFNAR deletion reduced Mg42, Mg44, Mg48 ('interferon'), Mg24 and Mg29 ('phagocytosis'), while increasing Mg15, Mg18 ('neuroprotection'), Mg33 ('phagocytosis') and Mg38 ('inflammation'; Fig. 7e,f). IFNGR deletion reduced Mg24, Mg29, Mg42, Mg44 and Mg48, increased Mg14, Mg15, Mg18 and Mg45, and nearly abolished Mg40 ('antigen presentation') during EAE (Fig. 7e,f).

Functionally, $Cx3cr1^{CreERT2}Ifngr1^{fl/fl}$ mice exhibited attenuated EAE severity with reduced cumulative scores, whereas IFNAR deletion did not alter clinical outcome (Fig. 7g–j). Network analysis revealed diminished interactions among microglia superclusters in IFNGR-deficient mice (Extended Data Fig. 10e). In contrast, LPS-associated microglial states required IFNAR but not IFNGR signaling (Extended Data Fig. 10f).

Together, these data demonstrate that CSF-1R, IFNAR and IFNGR signaling pathways in CNS myeloid cells modulate disease-associated microglial superclusters.

## Discussion

Here we present an integrative, high-resolution spatial and transcriptomic atlas of CNS myeloid cells in human and mouse, capturing microglial states across diverse pathologies and experimental conditions. By combining large-scale single-cell transcriptomics, targeted and genome-wide subcellular spatial profiling and in vivo perturbation experiments, we delineate how microglial diversity is structured by local tissue environment, disease context and conserved signaling pathways. The study makes two principal contributions. First, it introduces a hierarchical framework that defines superclusters as broad transcriptional modules and Paris clusters as fine-grained subprograms, reconciling modular and high-resolution views of microglial

heterogeneity. Second, by applying this framework across human diseases and mouse models, it clarifies when cross-species comparisons are biologically meaningful and when they diverge.

Unlike neuronal taxonomies, for which stable relationships between molecular identity, anatomy and function have been successfully established, linking microglial states to CNS regions and disease contexts has remained challenging[24–26]. To address this, we generated harmonized human and mouse datasets using uniform sample selection, processing, sequencing, analysis and validation pipelines, rather than relying on meta-analyses of heterogeneous public datasets. This strategy minimized pre-analytical variability and enabled direct comparison across species, regions and pathological states.

Across >450 high-quality samples encompassing 32 pathological or physiological conditions and 11 CNS regions, we profiled ~1,046,000 cells from human and mouse. Graph-based clustering resolved CNS myeloid cells into three major clades, that is, microglia, CAMs and MDMs, reflecting their distinct developmental origins, anatomical locations and functions. In humans, this approach defined eight microglial superclusters and 192 transcriptionally distinct clusters organized within a hierarchical dendrogram that captures both broad modules and fine-grained subprograms. These results are consistent with prior observations of myeloid diversity while providing a unified structure that integrates cellular identity to conserved functional programs.

This taxonomy has direct functional implications. Based on gene expression and alignment with existing literature[27], microglial clusters grouped into eight superclusters: 'surveillance', 'inflammation', 'proliferation', 'antigen presentation', 'phagocytosis', 'cytokine production', 'neuroprotection' and 'interferon signature'. In the non-diseased CNS, microglia occupied relatively few clusters, primarily within surveillance-associated states, whereas disease conditions induced a marked expansion in both cluster number and supercluster diversity. CAMs, despite sharing prenatal origin and longevity with microglia, formed fewer functional superclusters, likely reflecting their distinct anatomical niches and the disease contexts analyzed. Bone marrow-derived MDMs, by contrast, emerged predominantly during leukocyte infiltration of the CNS and occupied multiple transcriptionally and functionally distinct superclusters.

Recent microglia research has been driven substantially by the availability of new single-cell technologies. A microglial transcriptional core signature was initially described in mice, consisting mainly of *Tmem119*, *Slc2a5*, *Hexb*, *Sall1*, *Siglech*, *Fcrls*, *Cx3cr1*, *Csf1r* and *Gpr34* (refs. 3,15,27,35–39). Under physiological contexts such as development and aging, microglial transcriptomes were subsequently described as discrete phenotypes such as axon tract-associated microglia, white matter-associated microglia and proliferative region-associated microglia. This was followed by a proliferation of disease-specific and context-specific microglial subsets, frequently defined by lab or study-specific acronyms[27]. Many of these states, whether associated with neurodegeneration, development, aging or injury, share substantial overlap with the canonical DAM signature, including genes such as *Trem2*, *Axl*, *Cst7*, *Lpl*, *Itgax*, *Apoe* and *Clec7a*. Formal assessments of similarity between these proposed states were often lacking, fostering the impression of discrete, condition-specific phenotypes. Application of latent factor approaches such as topic modeling or multi-omics factor analysis to compare the diverse microglial states uncovers broad transcriptional modules. However, these modules or factors typically encompass hundreds of genes and often compress disease-specific or region-specific variation into broadly defined programs such as DAM, interferon-responsive microglia or antigen presentation. By contrast, our hierarchical strategy explicitly separates scale: superclusters capture broad, conserved transcriptional modules, whereas Paris clusters resolve highly specific, context-enriched subprograms. This structure integrates modular and fine-grained views of microglial heterogeneity within a single framework. Our data support a model in which a limited set of conserved transcriptional programs is flexibly deployed across

perturbations. The supercluster–cluster hierarchy reconciles apparent complexity by jointly representing shared functional modules and their context-dependent specializations, without imposing artificial disease-specific identities onto conserved biological programs.

Past studies proposed regional molecular phenotypes of microglia, for example in the basal ganglia[40]. In the present study, we observed only subtle transcriptional differences across CNS regions in both adult humans and mice. Individual microglial superclusters were broadly distributed across regions, indicating that microglial states are largely region independent at the transcriptomic level. Nonetheless, regional microenvironmental cues likely shape finer-grained cluster heterogeneity, particularly under pathological conditions. Across all regions, non-homeostatic states consistently drove a pronounced expansion of microglial diversity, supporting the concept of context-dependent microglial reprogramming.

Spatial transcriptomic analyses further refined this view by revealing how microglial superclusters and clusters integrate into local tissue architecture. DAM formed specialized niches in proximity to neurons, oligodendrocytes, astrocytes and infiltrating immune cells. Antigen-presenting and cytokine-producing microglia localized to MS lesion borders and dysplastic regions in FCD, illustrating coordinated neuroimmune cross-talk. The transition of microglia from peripheral sentinels in healthy tissue to central hubs in pathological cellular networks underscores their active regulatory role during disease.

A generally conserved evolutionary pattern of microglial genes has previously been described across 450 million years of evolution in the healthy CNS[17]. Cross-species integration confirmed that several core microglial programs represented as superclusters such as phagocytosis, interferon signaling and major histocompatibility complex-mediated antigen presentation are evolutionarily conserved. Mouse models of inflammatory and demyelinating disease, such as EAE, closely paralleled human MS, while Rnaset2-driven interferonopathy recapitulated key features of human AGS. At the same time, species-specific differences, both at supercluster and at cluster levels, were evident, including a relative underrepresentation of cytokine-producing microglia in mice. These interpretations are also constrained by species-specific gene expression baselines and contextual differences. Together, these findings emphasize both the translational value and the limitations of mouse models, highlighting the need for informed model selection when interrogating human microglial biology or developing therapeutic strategies.

Functional perturbation experiments directly demonstrated how conserved signaling pathways shape microglial state trajectories. Type I and II interferons have previously been shown to govern myeloid activation programs, regulating effector functions such as phagocytosis, cytokine and chemokine release, as well as cell migration[41,42]. Although myeloid-specific interferon signaling can contribute to maintain CNS homeostasis, chronic unregulated activation, for example in genetic interferonopathies, is detrimental[43]. Modulation of CSF-1R, IFNAR and IFNGR signaling in long-lived CNS myeloid cells, including microglia, profoundly altered microglial supercluster composition in a context-dependent manner, influencing disease severity in models such as EAE and cuprizone demyelination. These interventions shifted microglia along distinct activation trajectories, producing divergent outcomes depending on disease context. Together, these data underscore the therapeutic potential of targeting CNS myeloid cells and microglial signaling pathways, while also illustrating the necessity of precise, context-aware intervention strategies. A limitation of *Cx3cr-1*[CreERT2] mice is that it does not confer strict microglia specificity, as recombination also occurs in CAMs. Therefore, the observed phenotypes should be interpreted with caution and are most appropriately attributed to broader myeloid-lineage targeting rather than exclusively microglia-restricted genetic manipulation.

In summary, this study provides a multilayered framework for understanding CNS myeloid cell biology, integrating transcriptomic,

spatial and functional dimensions across species. By resolving conserved and context-specific microglial programs within a unified hierarchy, we offer a foundation for rational model selection and targeted immunomodulation aimed at reshaping microglial states to improve outcomes in CNS disease.

## Online content

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

Chintan Chhatbar [1], Roman Sankowski [1,2], Michael Schulz[1], Takashi Shimizu[1], Marius Schwabenland[1], Ori Staszewski [1,37], Christian Scheiwe [3], Stefan Nessler[4], Katharina Borst[1], Anaelle Aurelie Dumas[1], Ella Trost[1], Daniel Berchtold [5], Wesley Brandão[6], Omar Mossad[1,7], Adrià Dalmau Gasull [1,7], Maximilian Frosch [1], Daniel Erny [1], Martin Diebold [1], Elena Guffart[1], Katharina Ternka [8], Mihaela Guranda [4], Janaki Manoja Vinnakota[7,9], Marina Friesen[7,10,11], Koliane Ouk[12], Inken Waltl[13], Michael LaMorte[14], Timothy R. Hammond[14], Giovanni Di Liberto[15,16], Ilena Vincenti[15], Mario Kreutzfeldt [15], Ibrahim T. Mughrabi[17], Yousef Al-Abed [17,18], Thomas Blank [1], Melanie Meyer-Luehmann [10,11], Yanick J. Crow [19,20], Nellwyn Hagen[14], Dimitry Ofengeim[14], Robert Zeiser [9,21,22], Matthias Kettwig [8], Jutta Gärtner[8,23], Andreas Meisel [5,24,25], Martin Schwemmle [26], Ulrich Kalinke [13,27], Jürgen Beck[3], Bertram Bengsch [21,22,28], Robert Thimme [28], Oleg Butovsky [6,29], Tamara Seredenina[30], Richard M. Ransohoff [31], Francisco J. Quintana [6,32], Katrin Kierdorf [1,33], Doron Merkler [15,34], Christine Stadelmann[4,23], Josef Priller [12,35,36] & Marco Prinz [1,22] ✉

[1]Institute of Neuropathology, Medical Faculty, University of Freiburg, Freiburg, Germany. [2]Single-Cell Omics Platform Freiburg, Faculty of Medicine, University of Freiburg, Freiburg, Germany. [3]Clinic for Neurosurgery, Faculty of Medicine, University of Freiburg, Freiburg, Germany. [4]Department of Neuropathology, University Medical Center Göttingen, Göttingen, Germany. [5]Department of Neurology with Experimental Neurology, Charité - Universitätsmedizin Berlin, Berlin, Germany. [6]Ann Romney Center for Neurologic Diseases, Brigham and Women's Hospital, Harvard Medical School, Boston, MA, USA. [7]Faculty of Biology, University of Freiburg, Freiburg, Germany. [8]Department of Pediatrics and Adolescent Medicine, Division of Pediatric Neurology, University Medical Center Göttingen, Georg August University, Göttingen, Germany. [9]Department of Medicine I - Medical centre - University of Freiburg, Faculty of Medicine, University of Freiburg, Freiburg, Germany. [10]Department of Neurology and Clinical Neuroscience, Medical Center – University of Freiburg, Freiburg, Germany. [11]Faculty of Medicine, University of Freiburg, Freiburg, Germany. [12]Neuropsychiatry and Laboratory of Molecular Psychiatry, Charité – Universitätsmedizin Berlin and DZNE, Berlin, Germany. [13]Institute for Experimental Infection Research, TWINCORE, Centre for Experimental and Clinical Infection Research, Hannover, Germany. [14]Sanofi Neurology Research, Cambridge, MA, USA. [15]Department of Pathology and Immunology, University of Geneva, Geneva, Switzerland. [16]Department of Clinical Neurosciences, Service of Neurology, Lausanne University Hospital and University of Lausanne, Lausanne, Switzerland. [17]Institute of Bioelectronic Medicine, Feinstein Institutes for Medical Research, Northwell Health, Manhasset, NY, USA. [18]Department of Molecular Medicine, Donald and Barbara Zucker School of Medicine at Hofstra/Northwell, Hempstead, NY, USA. [19]MRC Human Genetics Unit, Institute of Genetics and Cancer, University of Edinburgh, Edinburgh, UK. [20]Laboratory of Neurogenetics and Neuroinflammation, Institut Imagine, University of Paris, Paris, France. [21]German Cancer Consortium (DKTK), partner site Freiburg, German Cancer Research Center (DKFZ), Heidelberg, Germany. [22]Signalling Research Centres BIOSS and CIBSS, University of Freiburg, Freiburg, Germany. [23]German Center for Child and Adolescent Health (DZKJ), partner site Göttingen, Göttingen, Germany. [24]Center for Stroke Research Berlin, Charité - Universitätsmedizin Berlin, Berlin, Germany. [25]Neuroscience Clinical Research Center, Charité - Universitätsmedizin Berlin, Berlin, Germany. [26]Institute of Virology, Medical Faculty, University of Freiburg, Freiburg, Germany. [27]Centre for Systems Neuroscience (ZSN), Hannover, Germany and Cluster of Excellence RESIST (EXC 2155), Hannover Medical School, Hannover, Germany. [28]Faculty of Medicine, Clinic for Internal Medicine II, Gastroenterology, Hepatology, Endocrinology and Infectious Disease, University Medical Center Freiburg, Freiburg, Germany. [29]Gene Lay Institute of Immunology and Inflammation, Brigham and Women's Hospital, Mass General Hospital and Harvard Medical School, Boston, MA, USA. [30]AC Immune, Lausanne, Switzerland. [31]Third Rock Ventures, Boston, MA, USA. [32]Broad Institute of MIT and Harvard, Cambridge, MA, USA. [33]Centre for Integrative Biological Signalling Studies, University of Freiburg, Freiburg, Germany. [34]Division of Clinical Pathology, Geneva University Hospital, Geneva, Switzerland. [35]Department of Psychiatry and Psychotherapy, School of Medicine and Health, Technical University of Munich and DZPG, Munich, Germany. [36]University of Edinburgh and UK DRI, Edinburgh, UK. [37]Present address: Institute for Neuropathology, Saarland University Hospital and Medical Faculty of Saarland University, Homburg, Germany. ✉e-mail: marco.prinz@uniklinik-freiburg.de

## Methods

### Mice

Wild-type or transgenic female mice (*Mus musculus*) on a C57BL/6N background were used for all experiments. Animals were 6–12 weeks of age unless otherwise indicated. Exact animal numbers (*n*) are provided in the corresponding figure legends. *SOD1-tg* mice[44] (B6.Cg-Tg(*SOD1*G93A*)1Gur/J*; JAX, 004435) and *Ifngr1*[fl/fl] mice[45] (C57BL/6N-Ifngr1tm1.1Rds/J; JAX, 025394) were purchased from The Jackson Laboratory. *Ifnar1*[fl/fl] mice[46] were provided by U. Kalinke. In order to generate mice to delete *Ifnar1* or *Ifngr* in long-lived *Cx3cr1*-expressing CNS cells, *Cx3cr1*[CreERT2] mice[47,48] were bred to *Ifnar1*[fl/fl] or *Ifngr1*[fl/fl] mice, respectively. For induction of recombination, 6-week-old female animals were treated twice subcutaneously with 4 mg of tamoxifen (Sigma-Aldrich, T5648-1G) dissolved in 200 ml of corn oil (Sigma-Aldrich, C8267-500ML), 48 h apart, performed as before[15,49]. For all experiments, Cre-negative littermates were used as controls. Mice were housed under a 12-h light/12-h dark cycle and at temperatures of 18–23 °C with 40–60% humidity, with food and water provided ad libitum. Diseased mice received wet food placed on the cage ground. All animal experiments were approved and performed in accordance with national and institutional regulations (Regierungspräsidium Freiburg, approval nos. X17/01A, X-20/01A, G-19/084, G-18/044, G-19/124, G-20/49, G-22/094, G-20/131, G-17/063; LAVES approval nos. 17-2697, 16/2338; Landesamt für Gesundheit und Soziales, Berlin, registration nos. G0312/16 and G0167/20, LAGeSO approval no. G0031/21). Supplementary Table 1 provides an overview of all mouse lines used with time points and age of animals.

### Treatments

VSV stocks were prepared and titrated by standard plaque assay. Mice were infected intranasally under light anesthesia with the indicated viral dose as previously described[49]. For demyelination, mice were fed 0.25% (wt/wt) cuprizone (Sigma) mixed into ground breeder chow for 5 weeks. For remyelination, cuprizone was withdrawn and animals were maintained on a normal diet for 1 week[50,51]. Induction of EAE[15], GL261 tumor[52], FNX[53], MCAO[54] and GvHD[55] models was conducted using established protocols. LPS intraperitoneal injection was given at a dose of 1 mg per kg body weight. For inhibition of CSF-1R, mice received an intraperitoneal injection of 25 mg per kg body weight of vehicle or the small-molecule CSF-1R inhibitor (sCSF1R$_{inh}$)[34]. To examine the impact of the CSF-1R inhibitor on microglia under homeostatic conditions, inhibitor treatment was performed for 1 week. In the LPS model, treatment with the CSF-1R inhibitor was performed simultaneously with LPS injection. During cuprizone demyelination, CSF-1R inhibitor treatment was performed three times per week. For the EAE model, treatment with the CSF-1R inhibitor was performed at a dose of 2 mg per kg body weight for 1 week at every 12 h (14 doses, 1 week).

### Human samples

Experiments on human tissue were conducted in accordance with the Declaration of Helsinki. A total of 212 samples derived from 177 individual donors (male and female) were analyzed, spanning 10 anatomical brain regions; multiple regions were obtained from some donors. Postnatal donor ages ranged from 1 to 96 years, and fetal samples ranged from 7 to 20 post-conception weeks. Detailed donor information, including sex, age, diagnosis and tissue source, is provided in Supplementary Table 3. Tissue was obtained from the Miami Brain Endowment Bank, the National Institutes of Health (NIH) NeuroBioBank, the Netherlands Brain Bank, the Queen Square Brain Bank for Neurological Disorders and the University of Freiburg Medical Center. Fresh-frozen samples were collected under approval of the local Research Ethics Committee of the University of Freiburg Medical Center (protocol nos. 472/15, 253/17 and 10008/09) and the respective ethics committees of contributing repositories. Written informed consent for research use was obtained from participants or their legal guardians or next of kin. Participants did not receive financial compensation. Samples were shipped on dry ice and stored at −80 °C until use. Formalin-fixed paraffin-embedded (FFPE) tissue used for spatial transcriptomics was derived from the same cohort. Fetal data were obtained from ref. 6. No statistical methods were used to predetermine sample sizes, but our sample sizes are similar to those reported in previous publications[56].

### Randomization statement

Animals in inducible disease models (for example, EAE and treatment paradigms) were randomly assigned to experimental groups before disease induction. For genetic models and for experiments involving animals' floxed genes (for example, 5xFAD, APP23 and other transgenic lines, *Ifnar1*[fl/fl], *Ifngr1*[fl/fl] animals), group allocation was determined by genotype and, therefore, not randomized. Single-cell transcriptomic and spatial profiling experiments were designed as observational atlas studies; thus, cell collection itself was not subject to randomization. Samples from different conditions were processed in parallel and balanced across library preparation and sequencing runs to minimize batch effects. No additional randomization procedures were required. Data collection and analysis were not performed blind to the conditions of the experiments.

### Isolation and barcode labeling of nuclei from tissues

For nuclei isolation, a small piece of tissue (size of a rice grain) was put into a 1.5-ml microcentrifuge tube and homogenized in 500 µl of ice-cold nuclei EZ lysis buffer (Sigma-Aldrich, NUC101-1KT). After adding 500 µl lysis buffer, the sample was mixed and incubated on ice for 5 min. The sample was then filtered through a 70-µm filter (Miltenyi, B60160056) and centrifuged at 500*g* for 6 min at 4 °C. Supernatant was removed, 1 ml ice-cold EZ lysis buffer was added and, after mixing, the sample was incubated on ice for 5 min. After centrifugation, supernatant was removed and the pellet was incubated for 5 min with 0.5 ml nuclei resuspension buffer (NRS, consisting of 1× DPBS (Sigma-Aldrich, D8537), 1% BSA (Miltenyi, 130-091-376) and 0.2 U µl$^{-1}$ RNase inhibitor (New England Biolabs, M0314L)) without mixing. Upon addition of another 0.5 ml NRS, the sample was mixed by gentle pipetting. After centrifugation at 500*g* for 6 min at 4 °C, the washing step was repeated with 1 ml of NRS without incubation. After centrifugation at 500*g* for 6 min at 4 °C, the supernatant was removed. Nuclei were incubated for 10 min with staining mix containing DAPI (10 µg ml$^{-1}$), Anti-Olig2 Alexa-488 (1:100 dilution; Merck, MABN50A4) and anti-NeuN Alexa-647 (1:100 dilution; Novus biologicals, NBP1-92693AF647). When individual samples were multiplexed in a single reaction using 3′ CellPlex kit, 25 µl of cell multiplexing oligonucleotides (10x Genomics, PN: 1000261) were added to an individual sample in a total of 200 µl of NRS. After 10 min, 1 ml of NRS was added and the sample was mixed. After centrifugation at 500*g* for 6 min at 4 °C, the supernatant was removed and the pellet was resuspended in 300 µl NRS and filtered through a 40-µm cell strainer (Fisher Scientific, 14-100-150).

### Fluorescence-activated nuclear sorting

To remove debris and enrich for non-oligodendrocyte and non-neuronal nuclei, Olig2$^+$ and NeuN$^+$ nuclei were removed after gating for DAPI$^+$ nuclei fraction. This strategy of sorting allowed substantial enrichment of non-neuronal and non-oligodendrocyte nuclei. Sorted nuclei were collected in a 1.5-ml tube containing 15 µl of NRS buffer. For brain metastasis, all DAPI$^+$ nuclei were collected.

### Single-nucleus library preparation and sequencing

For library preparation, nuclei from multiple samples of the same experimental group were mixed and loaded onto a Chromium Single Cell Chip G (10x Genomics) to generate single-nucleus gel beads in emulsion, according to the manufacturer's instructions, using Chromium Next GEM Single Cell 3′ Reagents Kit v3.1 (10x Genomics). Library

preparation for gene expression and cell multiplexing oligonucleotide libraries as well as multiplexing for sequencing was performed according to applicable guidelines from 10x Genomics. The cDNA content and size of post-sample index PCR samples were analyzed using a 2100 BioAnalyzer (Agilent). Library quantification was done using NEBNext Library Quant Kit for Illumina (New England Biolabs, E7630L) following the manufacturer's instructions. Sequencing libraries were loaded on an Illumina NextSeq flow cell, with sequencing settings according to the recommendations of 10x Genomics. Sample demultiplexing was done using built-in BCL2FASTQ or DRAGEN v3.8.4 software.

### snRNA-seq data processing

To generate a count matrix, Cell Ranger software (v7.1.0, 10x Genomics) was implemented for gene alignment either to the mouse (GRCm38, Gencode v33) or to the human genome (GRCh38, Gencode v35) depending on species. The mouse and human genome assembly and annotation files were downloaded from Ensembl. A pre-mRNA reference genome was generated to include both introns and exons using the commands recommended by 10x's Cell Ranger pipeline. Cell Ranger v7.1.0 with the --include-intron parameter was used for the generation of the count matrix. Data for human and mouse were analyzed separately in Rstudio build 524, with R programming language version 4.4.1. Downstream analysis implemented the demuxafy, SeuratWrappers and Seurat v5 R-based package.

### Creation of Seurat objects and quality control

Doublet detection and labeling were performed on individual datasets with demuxafy[57], which integrates multiple single-cell demultiplexing and doublet detection methods. We used DoubletDetection, scDblFinder and Scds[58]—packages embedded within the Demuxafy pipeline—for doublet detection and consensus labeling of doublets. Additionally, DoubletFinder[59] was run externally in R on individual datasets after the creation of Seurat object. Final labeling as a doublet was based on consensus between Demuxafy and DoubletFinder.

Filtered feature-barcode matrices for each sample were read into R using Seurat. In cases where data were generated via 10x Genomics Cell Ranger multi, each sample's expression matrix was similarly loaded, and associated metadata, including doublet labeling via demuxafy, were incorporated. Data from all samples were merged into a single Seurat object and additional metadata were incorporated. Mitochondrial quality-control metrics were calculated with the 'PercentageFeatureSet' function. Cells were removed if they (i) had fewer than 500 detected genes, (ii) had >10% mitochondrial gene content, or (iii) were annotated as doublets. Quality control was also performed at several stages during the analysis after clustering to exclude low-quality and outlier populations and/or clusters. Additionally, cells that were assigned conflicting 'class' and 'subclass' after applying query-reference mapping via Azimuth[60], for example immune cells such as microglia or vascular leptomeningeal cells annotated with class 'GABAergic' or 'Glutamatergic', were excluded from analysis.

### Batch correction and assignment of cell-type identities

For assessment of broad cell types, we performed Louvain clustering. After filtering based on genes and mitochondrial genes, merged data were normalized ('NormalizeData'), and highly variable features (up to 10,000) were identified with the 'vst' method. We applied linear scaling ('ScaleData') to each cell and performed a PCA on the top variable features. To mitigate batch effects arising from factors such as sex or brain regions, we used the Harmony[61] integration algorithm with default parameters (function 'RunHarmony') using sex and CNS region as covariates to integrate principal components. We specifically did not integrate the data based on individual samples that represent diverse disease or disease states to avoid overcorrection and loss of biological heterogeneity due to excessive integration[15,62–64]. Batch-corrected embeddings were used to compute a $k$-nearest-neighbor graph

('FindNeighbors') and to generate two-dimensional embeddings via UMAP ('RunUMAP'). Cell-type assignments for both mouse and human datasets were guided by query-reference mapping via Azimuth[65] using human or mouse motor cortex as reference annotations[60], as well as other published reference datasets describing cell-lineage markers, tissue-resident immune cell markers, lineage-specific transcription factors and genes associated with function[15,24,26,62–64,66–71].

### Analysis of the immune compartment

After broad cell-type annotation, immune cells were reanalyzed. To remove any residual nonimmune contaminants or multiplet artifacts, immune cells were clustered via the Louvain method into over 100 meta cells[25]. Cells that coexpressed canonical markers from different cell classes (for example, microglial and neuronal genes) were flagged as doublets and excluded.

### Paris clustering

For higher-fidelity identification of immune cell states, we built an adjacency matrix (using the Seurat nearest-neighbor graph) and exported it in GraphML format. Paris (a Python-based clustering tool) was run externally on this graph. A Paris dendrogram was cut for cluster assignments, which were reimported into Seurat. We applied a custom hierarchical merging function to combine clusters lacking sufficient differentially expressed genes or with very small cell numbers, similarly to previously described studies[24–26] that provided final cluster assignments. The final order of clusters was based on hierarchical graph ordering.

### Identification of myeloid superclusters by module scoring

To categorize microglial superclusters, we defined gene modules based on previously reported murine microglial states and converted them to human orthologs using the babelgene R package (v22.9). UCell-based scoring (UCell R package) was applied to each cell, and the resulting scores were scaled to identify the dominant module per cell. Clusters were subsequently classified by their most enriched module, yielding 'superclusters'. The selection of genes for defining modules was based on published studies describing diverse microglial states[3,15,35]. The modules were defined as follows: 'Surveillance': *OLFML3, P2RY12, TGFBR1, TMEM119*; 'Neuroprotection': *BDNF, GDNF, IGF1, LIF, TGFB1*; 'Phagocytosis': *APOE, AXL, C3AR1, CD63, CD68, CD9, CLEC7A, LGALS3, MYO1E, SPP1, TREM2, TYROBP*; 'Inflammation': *AOAH, APOD, C5AR1, CCL8, FGR, MSR1, RNF169*; 'Cytokine production': *CCL2, CXCL10, IL10, IL1B, NFKBIA, TNF*; 'Antigen presentation': *B2M, HLA-A, HLA-DMA, HLA-DQA1, HLA-DQB1, HLA-DRB5, NLRC5*; 'Interferon signature': *CD69, CXCL10, IFI16, IFIT2, IFIT3, IFITM3, IRF7, ISG15 MNDA, MX1, USP18*; 'Proliferation': *BRIP1, RAD51B, TOP2A*. These supervised modules are conceptually analogous to factors that might be derived by unsupervised matrix factorization approaches such as nonnegative matrix factorization or topic modeling, but enable direct biological interpretation anchored in prior knowledge.

### Building myeloid cell hierarchy

We organized myeloid cell clusters into a four-level hierarchy consisting of Cell type, Condition, Supercluster and Cluster. Following Paris clustering, the level Cell type was assigned based on canonical marker gene expression. Because several human microglial superclusters were present across both homeostatic and disease contexts, and each supercluster typically contained clusters enriched in either state, we introduced the level Condition to classify clusters broadly as homeostatic or non-homeostatic. The level Supercluster was defined as described above, based on module scoring against literature-derived gene sets. The final level Cluster corresponded to the original Paris clusters. Hierarchical relationships among myeloid clusters were visualized by exporting metadata into igraph objects and rendering dendrograms with the R package ggraph, using layered metadata annotations.

## Cross-species integration of human and mouse microglia

For cross-species analysis, mouse genes were converted to human orthologs using the convert_orthologs function from the orthogene package (non121_strategy = drop_input_species, method = gprofiler). After ortholog conversion, only genes shared between the mouse and human datasets were retained and merged into a single Seurat object. The merged object was normalized with NormalizeData (scale. factor = 10,000), and 3,000 variable features were identified using FindVariableFeatures. Data were then scaled independently for each species with ScaleData, followed by PCA using RunPCA. Integration of the human and mouse layers was performed with IntegrateLayers (method = 'RPCAIntegration'). Cells were clustered with FindNeighbors (dims = 1:15) and visualized by UMAP embedding with RunUMAP (dims = 1:15). Microglia superclusters were assigned by mapping each cluster to the supercluster with the highest proportional representation across species. To quantify cross-species relationships, aggregate gene expression profiles were computed for each supercluster and cluster using Seurat's AggregateExpression function (scaled data). Pairwise Pearson correlations were then calculated between aggregated expression modules from human and mouse using cor.test (stats package). Both correlation coefficients and $P$ values were reported, with statistical significance summarized as $*P < 0.01$, $**P < 0.001$ and $***P < 0.0001$. Results were visualized as heat maps displaying correlation values annotated by significance level using the pheatmap package. To evaluate robustness of these correlations, we performed 1,000 bootstrap iterations in which 80% of orthologous genes were randomly resampled and the full human–mouse correlation matrix recomputed. Empirical $P$ values were derived as the fraction of bootstrap correlations whose absolute value met or exceeded the observed value. For Sankey plot analysis microglia annotated as phagocytosis, antigen presentation or interferon signature in human, mouse and merged superclusters were subset and used to create new Seurat objects for each disease–model pair. To account for differences in cell numbers, objects were randomly downsampled to equalize human and mouse contributions while preserving the original cluster proportions. Connections between mouse and human supersclusters and clusters (≥1% contribution) were visualized using the ggsankey package. Distribution patterns of human microglial clusters across contexts were further summarized as dot plots based on cell count.

## Tissue processing, imaging, segmentation and analysis for CosMx

The CosMx Human Universal Cell Characterization RNA Panel (1000-plex) including genes for cell typing, cell state and function, and cell–cell interaction studies plus an additional 30 genes for detection for GBM tumor cells was used for spatial transcriptomics (Supplementary Table 4). Four FFPE tissue matrix arrays containing CNS tissues from multiple conditions and individuals were prepared and 4-µm-thick sections were cut. Sample processing, staining, imaging and cell segmentation were performed as previously described[72]. Briefly, tissue sections were placed on VWR Superfrost Plus Micro Slides (48311-703) for optimal adherence. Slides were then dried at 37 °C overnight, followed by deparaffinization, antigen retrieval and proteinase-mediated permeabilization (https://nanostring.com/products/cosmx-spatial-molecular-imager/single-cell-imaging-overview/). RNA-ISH probes (1 nM) were applied for hybridization at 37 °C overnight. After stringent washing, a flow cell was assembled on top of the slide and cyclic RNA readout on CosMx was performed (16-digit encoding strategy). After all cycles were completed, additional visualization markers for morphology and cell segmentation were added including Histone H3, 18S rRNA, GFAP antibodies and DAPI. Twenty-four 0.985-mm × 0.657-mm fields of view (FOVs) were selected for data collection in each slice. The CosMx optical system has an epifluorescent configuration based on a customized water objective (×13, NA 0.82), and uses wide-field illumination, with a mix of lasers and light-emitting diodes (385 nm, 488 nm, 530 nm, 590 nm, 647 nm) that allow imaging of DAPI, Alexa Fluor 488, Atto-532, Dyomics Dy 605 and Alexa Fluor 647, as well as removal of photocleavable dye components. The camera was a FLIR BFS-U3_200S6M-C based on the IMX183 Sony industrial CMOS sensor (pixel size of 180 nm). A three-dimensional multichannel image stack (nine frames) was obtained at each FOV location, with a step size of 0.8 µm. Registration, feature extraction, localization, decoding of the presence of individual transcripts and machine learning-based cell segmentation (developed upon Cellpose[73]) were performed as previously described[72]. The final segmentation mapped each transcript in the registered images to the corresponding cell, as well as to subcellular compartments (nuclei, cytoplasm, membrane), where the transcript is located. Data analysis was carried out with Seurat package.

## CosMX data analysis

A data matrix containing the NanoString counts was used to create a Seurat object with 'CreateSeuratObject' function as the Nanostring assay. Data matrices for segmentation and centroids were imported using the 'ReadNanoString' function, which was used to generate a centroid object using 'CreateCentroids' function, a segmentation object using the 'CreateSegmentation' function and a molecule object using the 'CreateMolecules' function. The centroid, segmentation and molecule objects were used to create spatial coordinates using the 'CreateFOV' function. The spatial coordinates for intracellular transcripts were imported in the Seurat object, which then contained the count data as well as spatial coordinates for molecules. Data from all samples were merged into a single Seurat object and additional metadata were incorporated. Cells containing less than five features and ten counts were eliminated using the 'subset' function. Data analysis and cell-type annotations were performed similarly to the snRNA-seq analysis as described above. After cell-type assignment, identification and assignment of microglial clusters was performed.

## Sample and library preparation, sequencing and data preprocessing for Visium HD

FFPE tissue samples of large areas were prepared from the frozen samples, which were then utilized to prepare tissue arrays containing regions of interest from diverse CNS region and contexts. FFPE tissue arrays were evaluated for quality and prepared for library construction based on recommendations in the Visium HD FFPE Tissue Preparation Handbook Rev A (CG000684) from 10x Genomics. After imaging, library construction was performed based on recommendations in Visium HD Spatial Gene Expression Reagent Kits User Guide (CD000685) from 10x Genomics. Visium Human Transcriptome Probe Set v2.0 was utilized for hybridization. Library quality control and sequencing were performed based on the run parameters provided by 10x Genomics. Sample demultiplexing was conducted using built-in DRAGEN software. To generate count matrix and spatial data, the Space Ranger pipeline (v3.0.0) with the spaceranger count command was run. Human genome reference transcriptome and probe set details (GRCh38-2020-A) were downloaded from the 10x Genomics website and provided to spaceranger as a reference. To compare gene expression between cells in the snRNA-seq data and Visium HD data, we binned the 2-µm expression data and reconstructed single-nucleus gene expression data using a custom data processing pipeline as described below.

## Nuclei segmentation, spatial binning and expression reconstruction

High-resolution bright-field images of tissue arrays were acquired following manufacturer protocols (10x Genomics). Stitched TIFF files for each tissue array were loaded into Python (v3.8). Spatial barcoded gene expression matrices and corresponding spatial coordinates were obtained from Space Ranger outputs and read into AnnData (scanpy v1.9)[74] and pandas, respectively. Nuclei were detected with the StarDist2D 'versatile_he' pretrained model (stardist v0.8.0). Raw images

were percentile-normalized between the 5th and 90th percentiles, then passed to 'model.predict_instances_big' to generate instance coordinates. The resulting instance coordinates were converted to Shapely Polygon objects, then dilated by a 2-µm buffer to approximate true nuclear boundaries. All polygons were aggregated into a GeoDataFrame with unique IDs. Visium HD counts were merged with pixel-level coordinates by barcode using pandas merge. Coordinates were converted to Shapely Point and assembled into a separate GeoDataFrame. A spatial join assigned each transcriptomic spot to a nucleus polygon if its center fell within a polygon. Spots mapping to multiple overlapping nuclei were flagged and removed, retaining only barcodes within exactly one nucleus. Barcodes uniquely assigned to a single nucleus were extracted. Expression matrices were grouped by nucleus ID and summed per gene using a compressed sparse row matrix, yielding one gene-by-nucleus AnnData object. The nucleus-specific polygon area was stored in the GeoDataFrame. Final single-nucleus AnnData objects were saved in H5AD format for importing and downstream analyses in R/Seurat. Corresponding nucleus polygons were exported in GeoJSON format for spatial mapping.

### Analysis of Visium HD data

H5AD files were imported in R/Seurat and spatial coordinates for polygons were imported using 'st_read' with the 'sf' R package. Segmentation and centroids for each cell were created and incorporated in the Seurat object for each tissue array. Metadata for all samples were incorporated and a merge Seurat object containing all tissue arrays was created. The percentage of mitochondrial gene expression was calculated. Quality control included exclusion of cells expressing less than five features and more than 75% mitochondrial genes. Data normalization, scaling and PCA were performed as described in snRNA-seq analysis. Because Visium HD data contained diverse neuronal, glial, vascular and immune cells from various CNS regions, we performed preliminary clustering analysis with different PCA dimensions (1:10, 1:20, 1:30, 1:40) and at different resolutions. The final analysis was performed with 40 PCs and a clustering resolution of 2. This allowed for proper separation between diverse cell populations. After clustering, we performed broad cell-type labeling to group cells into immune cells, oligodendrocytes, excitatory and inhibitory neurons, structural and vascular cells, and a mixed population of cell types that did not properly separate, probably due to low cell counts of individual cell types. We individually reanalyzed immune cells, structural and vascular cells, and the mixed cell populations. Depending on context and region, immune cells partly clustered together with other cell types and, therefore, cell labels were manually corrected based on marker gene expression. After cell-type assignment, identification and assignment of microglial clusters was performed.

### Imputation and plotting of microglial clusters

We projected 10x snRNA-seq-defined human microglial clusters onto CosMX and Visium HD data by leveraging cluster-specific differentially expressed gene signatures. Briefly, shared genes between snRNA-seq and CosMX as well as Visium HD datasets were identified. Cluster-wise differentially expressed genes for each human microglial cluster were determined with the 'FindAllMarkers' function using the set of common genes for each individual dataset. Differentially expressed genes with an adjusted $P$ value < 0.05 were selected. The top 10 to 30 differentially expressed genes per cluster were used as an input for creating cluster-wise gene modules. Microglia within CosMX and Visium HD datasets were scored for average expression of each cluster's gene module using the 'AddModuleScore_UCell' function. Each microglial cell was then assigned the snRNA-seq cluster whose gene module yielded the maximum score. Due to restricted gene panels in the CosMX dataset, not all microglial clusters could be imputed. For projection of mouse microglial clusters in the data with pathway targeting, the same strategy was used. The assignment for superclusters was based on the imputed cluster of microglia corresponding to the supercluster

grouping of either human or mouse microglia. For overlay of amyloid beta and microglial clusters in AD Visium HD data, individual spatial plots were imported and aligned in Adobe Illustrator.

### Overall and differential gene expression analysis and visualization

Marker detection and differential gene expression for all clusters, cell types and other comparisons for metadata were done with Seurat's 'FindAllMarkers' and 'FindMarkers'. Genes encoded by mitochondrial chromosomes or ribosomal genes were excluded.

### Supercluster similarity analysis and statistical testing

To assess the transcriptional similarity between microglial superclusters, we computed the log-normalized average gene expression per supercluster using the AverageExpression() function (slot = 'data'). To ensure comparability across supercluster, the resulting expression matrix was $z$-score normalized across supercluster (that is, column-wise) for each gene, and genes with zero variance were excluded. We computed pairwise module similarity using Spearman's correlation on the scaled average expression matrix. To assess statistical significance, we calculated two-tailed $P$ values for each pairwise correlation using the cor.test() function in R, and constructed a symmetric $P$-value matrix. For visualization, both the correlation and $P$-value matrices were subset to a manually defined order of biological relevance: major histocompatibility complex, phagocytosis, proliferation, interferon, neuroprotection, surveillance, inflammation and cytokine supercluster. A corresponding annotation table and color palette were used to label and color-code superclusters in the heat map. The final heat map was generated using the pheatmap R package, displaying the Spearman correlation coefficients between modules. Statistical significance of correlations are: *$P$ < 0.05, **$P$ < 0.01 and ***$P$ < 0.001. Asterisks on the diagonal were omitted for visual clarity. The color scale ranges from −1 (blue) to +1 (red), representing negative to positive correlation, respectively.

For correlation and visualization of cluster-wise differentially expressed genes for myeloid cells, the top differentially expressed genes were provided as input for expression summarization, variance filtering and $z$-scoring as above. Pearson correlation coefficients were computed pairwise among clusters based on the $z$-scored expression profiles and visualized. For correlation and visualization of gene expression in the imputed microglial clusters in CosMX data, heat maps were plotted with 'DoHeatmap' function (disp.min = -1, disp.max = 1) from Seurat after calculation of aggregate gene expression.

### Plotting of CosMx and Visium HD spatial data

Individual FOVs were selected as a representative figure for the context. Spatial visualization of all cell types, microglia, imputed 10x microglial clusters and associated enriched genes was generated by 'ImageDimPlot' function in Seurat. UMAPs were plotted using the 'DimPlot' function. For overlay of Visium HD analyzed data on the original H&E image, a bespoke Python GUI pipeline built atop standard image and geospatial processing libraries was utilized. High-resolution H&E images were first loaded and contrast normalized to enhance morphological features. Polygonal spot geometries imported from GeoJSON files were read into a GeoDataFrame (geopandas), joined with per-spot metadata (CSV), and colored according to a user-supplied palette CSV mapping each cell-type label to an RGB hex code. Visualization routines rely on matplotlib for rendering, with AnchoredSizeBar for micrometer scale bars. Custom function was applied for selection of metadata to display and for image exporting and saving.

### Neighborhood analysis of CosMx data

A custom R function was implemented to quantify spatial proximity relationships between annotated cell types and microglial superclusters. In brief, cell identities were assigned (cell-type identity for

non-microglial cells and supercluster identity for microglia) and only those identities with at least three cells were retained to ensure robust network inference. Pairwise Euclidean distances between all cell centroids were computed using base R's 'dist' function. For each cell $i$, neighboring cells within a radius of 100 μm were identified.

### Edge and node summary statistics

For each cell identity pair (type A → type B), the average inter-cell distance was calculated as the mean of all cell-to-neighbor distances, normalized to micrometers by dividing by 10 if necessary. The interaction proportion was defined according to equation (1)

$$\text{Proportion}_{A \to B}$$
$$= \frac{\text{number of neighbors of type B around cell of type A}}{\text{number of neighbors total around cell of type A}} / (N_A \times N_B) \quad (1)$$

and expressed as a percentage. Edge-level values were aggregated across all source cells to yield a single average distance and proportion per cell-type pair.

### Graph assembly and visualization

The summarized edge table was converted into an undirected graph object using the igraph R package (v1.2.6). Cell-type nodes with zero edges were reintroduced to maintain a complete type palette. Node sizes were scaled to their total cell counts, and edge thicknesses were mapped to the interaction proportions. To distinguish, nodes for microglial superclusters were rendered squares. Graph layouts were computed with the force-directed Fruchterman–Reingold algorithm. Network visualizations were produced with ggraph (v2.0.5) and ggplot2 (v3.3.5).

### Image feature plots for CosMX data

Gene modules to highlight expression of selected pathway or function as below. Interferon signaling: *IFNGR1, IFNGR2, CXCL10, IDO1, ICAM1, CD274*. Cytokine response: *IL6, TNF, IL1B, CCL2, CCL3, CCL4*. Homeostatic signature: *CX3CR1, P2RY12*. Antigen presentation: *HLA-DPA1, HLA-DPB1, HLA-DQA1, HLA-DQB1, HLA-DRA, HLA-DRB1, HLA-DRB5, HLA-A, HLA-B, HLA-C, HLA-E, NLRC5, CIITA*. Phagocytic activity: *CD9, SPP1, TYROBP, CLEC7A, CD63, LGALS3, AXL, CD68*. All count data were extracted from the combined Seurat object using Seurat's FetchData function. For each gene module, expression values were summed across member genes only within microglia; non-microglial cells were assigned an expression of zero. Module-level scores were then normalized across all FOVs by computing the $z$-score of the cumulative module expression per cell. The resulting $z$-scores were constrained between −1 and +1.

### Spatial mapping and visualization

$z$-scored module scores were written back into the Seurat object metadata. Cumulative $z$-score values (range between −1 and +1) were assigned back into the full object, and data plotted as 'Imagefeatureplot'.

### Cell–cell interaction analysis

CellChat package[75] version 2.1.2 was used for inference of cell–cell interactions. To decipher interactions between individual cell types or microglial clusters, appropriate identities were set. Seurat objects were used for creating CellChat objects. Overall evaluations for cell–cell interaction, signaling pathways and contribution of individual cell identities were performed, and data for individual pathways, cellular interactions and pathways were plotted using the inbuilt plotting functions of the CellChat package.

### IMC

IMC was conducted as reported previously[76]. In short, antibodies were conjugated to lanthanide metals using the Maxpar X8 antibody labeling kit. Four-micron-thick FFPE sections were deparaffinized and cooked in EnVision FLEX Target Retrieval Solution High pH (DAKO, K8000) for 40 min. Sections were blocked using SuperBlock Blocking Buffer (Thermo Fisher, 37581). The slides were then incubated with the antibody mix (Supplementary Table 2) in 0.5% BSA, 1% Triton X-100 in TRIS at room temperature (RT) overnight. Iridium Cell-ID intercalator (Fluidigm, 201192A) was used to visualize DNA and applied for 30 min. The measurement was conducted using the Hyperion IMC system (Fluidigm). Data analysis was performed as described previously[76].

### Immunohistochemistry

Paraffin-embedded tissue sections (3-μm thick) were initially incubated at 80 °C for 1 h to facilitate deparaffinization, followed by immersion in xylene to complete the process. Antigen retrieval was carried out by incubating the sections for 40 min at 95 °C in EnVision FLEX Target Retrieval Solution, using either pH 6 buffer (S1699, Dako) for IBA1 or pH 9 buffer (S2367, Dako) for MAP2 staining. To block endogenous peroxidase activity, sections were treated with 3% hydrogen peroxide for 10 min. This was followed by blocking in PBS containing 5% BSA (8076.3, Roth) and permeabilization with 1% Triton X-100 (T8787, Sigma) for 1 h. Primary antibodies were applied overnight at 4 °C using the following dilutions: IBA1 (ab178846, Abcam) at a dilution of 1:1,000, beta-amyloid (M0872, Dako) at 1:100, and MAP2 (ab183830, Abcam) at 1:10,000. After three PBS washes, sections were incubated with biotinylated secondary antibodies (Southern Biotech) for 45 min at RT, either goat anti-mouse IgG at a dilution of 1:200 (1031-08) or goat anti-rabbit IgG at a dilution of 1:300 (4050-08). Following another three PBS washes, streptavidin-conjugated peroxidase (PK-6100, Vector Laboratories) was applied for 45 min at RT. After a final set of three washes, signal detection was performed using DAB: one drop of EnVision FLEX DAB Chromogen (DM827, Dako) per 1 ml of EnVision FLEX Substrate Buffer (DM823, Dako). Luxol fast blue combined with the periodic acid-Schiff procedure was performed as previously described[76]. For staining of frozen sections (8-μm thick), antigen retrieval was performed by incubating sections in pH 9 buffer for 30 min after which sections were blocked with PBS containing 5% BSA and permeabilized with 0.1% Triton X-100 in blocking solution. Anti-TMEM119 primary antibody was added overnight at a dilution of 1:500 (Abcam, ab209064) at 4 °C, followed by washing and incubation with Alexa Fluor 647 anti-mouse CD74 (1:100 dilution; 151004, BioLegend) and secondary antibody anti-rabbit Alexa Flour 488 (1:1,000 dilution; Thermo Fisher Scientific) for 2 h at RT. Coverslips were mounted with ProLong Diamond Antifade Mountant with DAPI (Thermo Fisher Scientific). Images were taken using a conventional fluorescence microscope Olympus BX-61 with a color camera (Olympus DP71).

### Immunofluorescence for microglial activation-associated genes

For immunofluorescence staining, after antigen retrieval (Pascal citrate, 10 mM, pH 6.0, 30 s, 125 °C, 21 psi) and blocking of nonspecific binding (peroxidase blocking solution; DAKO, S2023; 10% fetal calf serum in PBS), formalin-fixed sections were incubated with rabbit anti-FKBP5 (polyclonal, 1:1,000 dilution; Proteintech, 14155-1-AP), and bound antibodies were visualized with anti-rabbit tyramide signal amplification Plus Opal 570 (Akoya, OP-001003). All bound antibodies were denatured with denaturation buffer (Roche, 07570791001), and sections were incubated with rabbit anti-P2Y12R (polyclonal, 1:3,000 dilution; Sigma-Aldrich, HPA014518) followed by anti-rabbit tyramide signal amplification Plus Vivid 520 (Tocris, 7534). All bound antibodies were denatured again with denaturation buffer (Roche, 07570791001). After blocking of nonspecific binding (1% BSA in PBS), sections were incubated with goat anti-GPNMB (polyclonal, 1:100 dilution; RD Systems, AF2550), followed by rabbit anti-goat (polyclonal, 1:250 dilution; DAKO, E0466), and bound antibodies were visualized with donkey anti-rabbit-Al647 (Life Technologies, A31573). GPNMB was utilized as a marker for dysmorphic neurons in FCD. Nuclei were stained with DAPI (Invitrogen). Immunostained sections were scanned

using a Panoramic 250 FLASH II (3DHISTECH) Digital Slide Scanner with an objective magnification of ×20, and single-plane confocal images were acquired (Zeiss LSM800), sampling brain tissue at ×40 magnification. To quantify positive signals, an image analysis rule set based on the Definiens Cognition Network Language was applied as previously described[77]. For representative images, white balance was adjusted, and contrast was linearly enhanced using the tool levels, curves, brightness and contrast in Adobe Photoshop CC. Image processing was applied uniformly across all images within a given dataset.

### RNAscope-based in situ hybridization

RNAscope Multiplex Fluorescent Reagent Kit v2 (323100, Bio-Techne), RNAscope Probes Mm-*Hexb*-C3-*Mus musculus* hexosaminidase B (*Hexb*) mRNA (314231-C3, Bio-Techne) and Mm-*Msr1*-*Mus musculus* macrophage scavenger receptor 1 (*Msr1*) transcript variant 1 mRNA (824201, Bio-Techne) were used for in situ hybridization. Sample pretreatment for fixed-frozen tissues and RNAscope assay was performed following manufacturer's instructions with the following modifications. Post-fixation of samples was performed in 4% paraformaldehyde in PBS and dehydration in 50% ethanol for 1 min. Incubation of samples with RNAscope protease III was performed for 10 min. Visualization of C1 and C3 channel signals was performed using Opal dyes according to the manufacturer's instructions. Opal 570 dye was used for visualization of *Msr1* transcripts (C1 channel), and Opal 520 dye was used for visualization of *Hexb* transcripts (C3 channel). To assess the background signals for each channel, a negative control sample was used where hybridization of RNAscope 3-plex Negative Control Probe (320871, Bio-Techne) was performed along with test samples. Imaging was performed on a TCS SP8 X (Leica, LAS X 3.5.7.23225) microscope using a ×20 0.75 NA (HC PL APO CS2 ×20/0.75 IMM) objective. Serial confocal sections were obtained in the *z*-plane at a pixel size of 2 μm. Image processing and quantification was performed using Fiji software. Confocal stacks were imported, maximum intensity projection was performed, and channels were split. Brightness and contrast were adjusted, and a composite image for Hexb and Msr1 channels was generated. For quantification of *Msr1*⁺ microglia, the first microglia were identified as *Hexb*⁺ cells and after that only *Hexb*⁺ or *Hexb*⁺*Msr1*⁺ cells were counted using the Cell Counter plugin. Quantifications were imported in GraphPad Prism for statistical analysis and plotting. Representative composite or single-channel images were imported in Adobe illustrator.

### Final visualization

All UMAP plots were generated in Seurat. Dot plots for marker genes (DotPlot), violin plots (VlnPlot) and correlation heat maps (pheatmap) were used to visualize expression patterns across clusters or states. Dendrograms to depict hierarchical relationships among subpopulations were created by exporting cluster metadata into igraph objects and using ggraph with layered metadata annotation. Data for the Marimekko plots were prepared by merging the relevant sample metadata with our quantitative dataset using the dplyr package[78]. The merged table was then visualized in ggplot2 by customizing bar widths and fill positions (position = 'fill') to emulate the Marimekko layout.

### Data inclusion and exclusion criteria

No animals were excluded from the study after group allocation. For single-cell transcriptomic and spatial analyses, cells were filtered using standard quality-control criteria to remove low-quality or artifactual profiles, including cells with low transcript counts, high mitochondrial RNA content or predicted doublets, as specified below. These filtering steps are standard for single-cell analyses and were applied uniformly across all samples before downstream analyses.

### Statistical analysis and reproducibility

Data distribution was assumed to be normal, but this was not formally tested. Unless otherwise stated, comparisons between groups were performed using one-way ANOVA followed by Tukey's multiple-comparison test in GraphPad Prism (v5.04). A significance threshold of 0.05 was used.

To quantify inequality in the distribution of myeloid cell clusters across conditions, the Gini coefficient was calculated as a descriptive measure. Before calculation, cluster cell counts were normalized to the total number of cells per condition. Gini coefficients were computed using the ineq R package (v0.2-13).

To assess enrichment of cell types, superclusters or clusters within specific conditions, hypergeometric tests were performed using the phyper function in base R to estimate the probability of observing *n* or more cells by chance. Resulting *P* values were corrected for multiple testing using the Benjamini–Hochberg method.

### Microglial cluster count analysis

To quantify differences in microglial cluster diversity between conditions, we first summarized the number of distinct microglial clusters and the number of nuclei analyzed per brain region × condition. For each region, clusters were defined using the Paris clustering of microglia, and nuclei counts were obtained directly from the Seurat metadata. We then fit generalized linear models with a Poisson error distribution using the lme4 package (v1.1-37). Models included condition (homeostasis versus non-homeostasis) as the main predictor and log-transformed nuclei count as a covariate to control for sampling depth. Where donor information was available, patient ID was added as a random intercept; in sensitivity analyses, we fit alternative models with anatomical region as a random effect or without random effects. Overdispersion was assessed using Pearson residuals, and negative binomial models were fit using the MASS package if dispersion exceeded 1.5. All models consistently showed increased cluster numbers in non-homeostasis.

### Reporting summary

Further information on research design is available in the Nature Portfolio Reporting Summary linked to this article.

## Data availability

Human raw sequencing data files are restricted and can be accessed at the European Genome–phenome Archive under accession code EGAS50000001289. Mouse raw sequencing data are available in the NCBI Gene Expression Omnibus under accession code GSE304010. Processed Seurat objects and the code to reproduce figures are deposited in Zenodo (https://doi.org/10.5281/zenodo.16938034)[79]. Source data are provided with this paper.

## Code availability

Analysis code used to reproduce the figures is available in Zenodo (https://doi.org/10.5281/zenodo.16938034)[79].

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

## Acknowledgements

We thank J. Bodinek-Wersing, T. El Gaz, S. Wundt, A. Müller and A. Frömming for excellent technical support. We are very grateful to H. Kaessmann, B. Zaremba, T. Yamada and X. Yuan from ZMBH - Center for Molecular Biology, Heidelberg for their excellent and helpful advice for analyzing cross-species microglia data. We thank the Lighthouse Core Facility for its support. R.S. is supported by the IMMediate Advanced Clinician Scientist-Program, Department of Medicine II, Medical Center – University of Freiburg and Faculty of Medicine, University of Freiburg, funded by the BMBF, Federal Ministry of Education and Research – 01EO2103. Furthermore, R.S. is supported by the Fritz Thyssen Foundation. T.S. is supported by a JSPS Overseas Research Fellowship (grant no. 202560306). A.A.D. is supported by the Hans A. Krebs Medical Scientist Program, Faculty of Medicine, University of Freiburg. M. Schwabenland was supported by the Berta-Ottenstein-Programme for Clinician Scientists, Faculty of Medicine, University of Freiburg, and the IMM-PACT-Programme for Clinician Scientists, Department of Medicine II, Medical Center – University of Freiburg and Faculty of Medicine, University of Freiburg, funded by the Deutsche Forschungsgemeinschaft (DFG, German Research Foundation) – 413517907. M.P., J.P., A.M., D.E., K.K., T.B., M.M.-L. and R.Z. are supported by the DFG collaborative research center (CRC/TRR167 project ID259373024, 'NeuroMac'). J.P. is also supported by the DFG (CRC/TRR265 project ID402170461), the BMBF (DZPG, project 01EE2303B; DZNE, Internal Order 186000060) and a program award of the UK DRI. D.E. is further supported by the Else Kröner-Fresenius-Stiftung (CRC/TRR 359 project ID 491676693) and the BMBF (project ID 01ED2305A). M.P. is further supported by the Sobek Foundation, the Ernst-Jung Foundation, the Klaus Faber Foundation, the Novo Nordisk Foundation, the German Research Foundation (SFB 1160, SFB 1479 project ID 441891347, TRR 359 Project ID 491676693, Gottfried Wilhelm Leibniz Prize) and by the DFG under Germany's Excellence Strategy (CIBSS – EXC-2189 – project ID390939984). M. Schulz is supported by the DFG (533752938). W.B. is supported by National Multiple Sclerosis Society (FG-2108-38372), Department of Defense (W81XWH-22-1-0945), and NIH (R01NS088137). Y.J.C. acknowledges the European Research Council (786142 E-T1IFNs), a UK Medical Research Council Human Genetics

Unit core grant (MC_UU_00035/11), a state subsidy from the Agence Nationale de la Recherche (France) under the 'Investissements d'avenir' program bearing the reference ANR-10-IAHU-01. R.Z. is supported by SFB 1479 project ID 441891347 and EU Proposal no. ERC-2022-ADG Project: 101094168 — AlloCure (ERC Advanced grant). G.D.L. is supported by grants of the Baasch-Medicus Foundation, Swiss Neurological Society, Glauser Foundation, Novartis Foundation for Biological-Medical Research and University of Lausanne. S.N. was supported by the DFG, NE2447/1-1 (individual research grant); C. Stadelmann was supported by the DFG, STA1389/5-1 (individual research grant), STA1389/6-1, the DFG under Germany's Excellence Strategy (EXC2067/1-390729940), the Hertie Foundation, and the National MS Society (USA). C. Stadelmann and S.N. are supported by the DFG Priority Program 2395 'Local and peripheral drivers of microglia diversity and function' (project ID 500301720; NE2447/2-1/2; STA1389/7-1/2) and the DFG TRR274 'Checkpoints of CNS recovery' (project ID 408885537 B01/B02). T.B. was funded by the Alzheimer Forschung Initiative (AFI, no. 23012R). J.G. is supported by the German Research Foundation (DFG), grant no. TRR 274, by the DFG Transregional Collaborative Research Centres (CRC) TRR 274/1 and 2, and by the DFG under Germany's Excellence Strategy (grant no. EXC 2067/1-390729940). J.G. and C. Stadelmann are supported by the German Center for Child and Adolescent Health (DZKJ), funded by the Federal Ministry of Research, Technology and Space (BMFTR), funding code 01GL2402. M. Schwemmle is supported by the German Research Foundation (SFB 110, project C01). M.M.-L. is funded by the DFG German Research foundation ME 3542/2-4 project ID 323695292. D.M. is supported by the Swiss National Science Foundation (310030B_201271 and 310030_215050) and the ERC (865026). K.K. is supported by the DFG through TRR 359 (Project ID 491676693, SFB 1160 (Project ID 256073931), TRR167 (Project ID 259373024), CRC1479 (Project ID 441891347), FOR5775 (Project ID 533863915) and by the DFG under Germany's Excellence Strategy (CIBSS-EXC-2189, Project ID 390939984). K.K. is also supported by the Heisenberg program of the DFG (Project ID 544402801). Human tissues were obtained from the Miami Brain Endowment Bank (University of Miami), the NIH Neurobiobank at the University of Maryland, Mount Sinai, NIH Brain & Tissue Repository-California, Human Brain & Spinal Fluid Resource Center, VA West Los Angeles Medical Center, which is supported in part by the NIH and the US Department of Veterans Affairs, the Netherlands Brain Bank (Netherlands Institute for Neuroscience) and University Freiburg Medical Center.

## Author contributions

C.C., R.S., M. Schulz, E.T., M.L., M.F., G.D.L., M.D. and I.V. performed experiments. C.C., R.S., M. Schulz, T.S., M. Schwabenland, O.S., G.D.L. and M. Kreutzfeldt analyzed the data. C. Scheiwe performed tissue acquisition. C.C., S.N., K.B., A.A.D., D.B., W.B., O.M., A.D.G., M.F., D.E., E.G., K.T., J.B., M.G., J.M.V., M.F., K.O., J.P., I.W., M.L., T.R.H., I.T.M., Y.A.A., T.B., M.M.-L., N.H., D.O., R.Z., M. Kettwig, J.G., A.M., K.K., M. Schwemmle, U.K., B.B., R.T., T.S., O.B., C. Stadelmann and Y.J.C. contributed to in vivo studies and/or provided mice or reagents. C.C., D.M., R.M.R., F.J.Q., J.P. and M.P. supervised the study. C.C. and M.P. wrote the paper with input from all authors.

## Funding

## Competing interests

The authors declare no competing interests.

## Additional information

**Extended data** is available for this paper at https://doi.org/10.1038/s41590-026-02472-z.

**Correspondence and requests for materials** should be addressed to Marco Prinz.

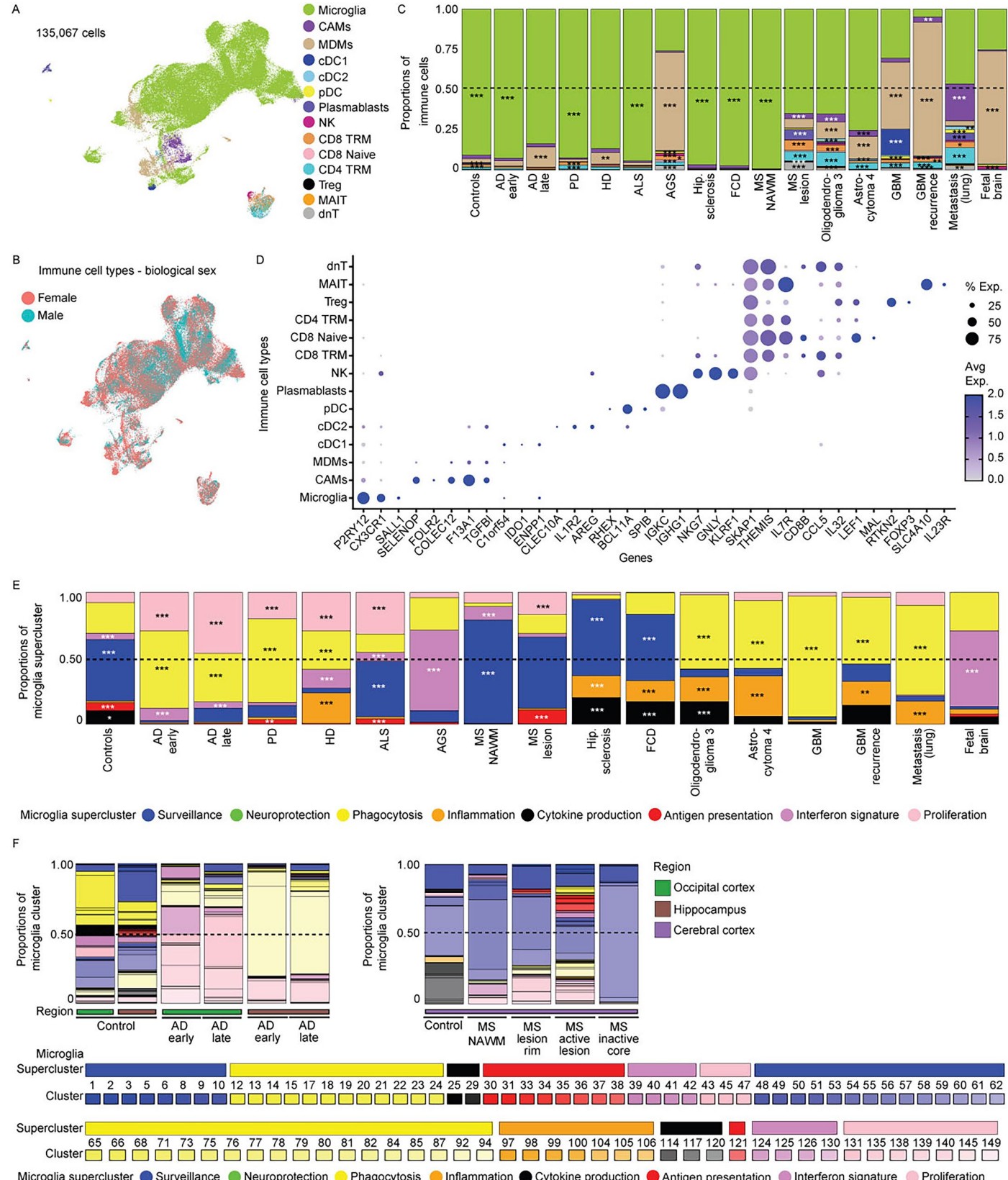

**Extended Data Fig. 1 | See next page for caption.**

**Extended Data Fig. 1 | Proportions of immune cell types across CNS disease in humans. a** UMAP visualization of single-cell transcriptomic states of CNS immune cells from contexts in Fig. 1 colored by cell types. CAMs: CNS-associated macrophages, MDMs: Monocyte-derived macrophages, cDC: conventional dendritic cells, pDC: plasmacytoid dendritic cells, NK: natural killer cells, TRM = T resident memory, Treg: regulatory T cells, MAIT: Mucosal-associated invariant T cells, dnT: double negative T cells. **b** UMAP visualization of immune cells based on biological sex from Extended Data Fig. 1A. **c** Stacked bar plot representing the relative contribution of each immune cell types to the mentioned contexts. Cell types are same as in Extended Data Fig. 2A. One-sided hypergeometric test with BH correction; * p < 0.05, ** p < 0.01, *** p < 0.001. Significance marks are shown for a subset due to space constraints. **d** Dot plot representing the average expression of selected marker genes associated with each immune cell types. Cell types are same as in Extended Data Fig. 1A. **e** Bar plots representing distribution of microglia superclusters in different contexts. One-sided hypergeometric test with BH correction; * p < 0.05, ** p < 0.01, *** p < 0.001. **f** Bar plots representing distribution of microglia clusters in human contexts across diverse stages. AD early: early onset Alzheimer's disease, AD late: late onset Alzheimer's disease, MS NAWM: multiple sclerosis normal appearing white matter, MS lesion rim: multiple sclerosis lesion border, MS active lesion: multiple sclerosis actively demyelinating lesion, MS inactive core: multiple sclerosis lesion core with no active demyelination.

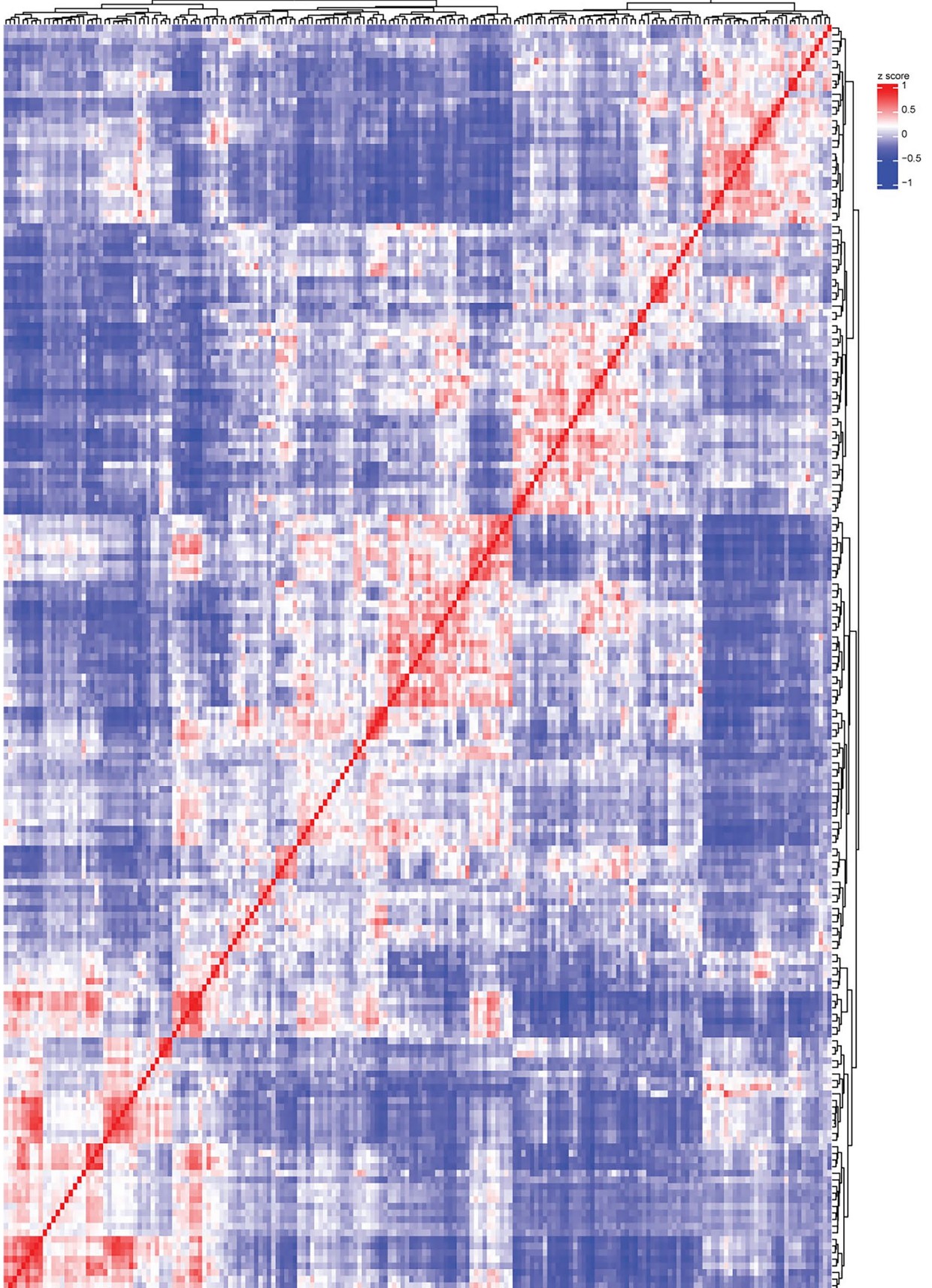

**Extended Data Fig. 2 | Correlation heat map of top differentially expressed genes across human myeloid clusters.** Heat map showing pairwise correlations of the top 10 differentially expressed (DE) genes for each human myeloid cell cluster identified in Fig. 1b. Correlation coefficients were calculated using normalized gene expression values across clusters. Warmer colors indicate stronger positive correlations, whereas cooler colors indicate weaker or negative correlations.

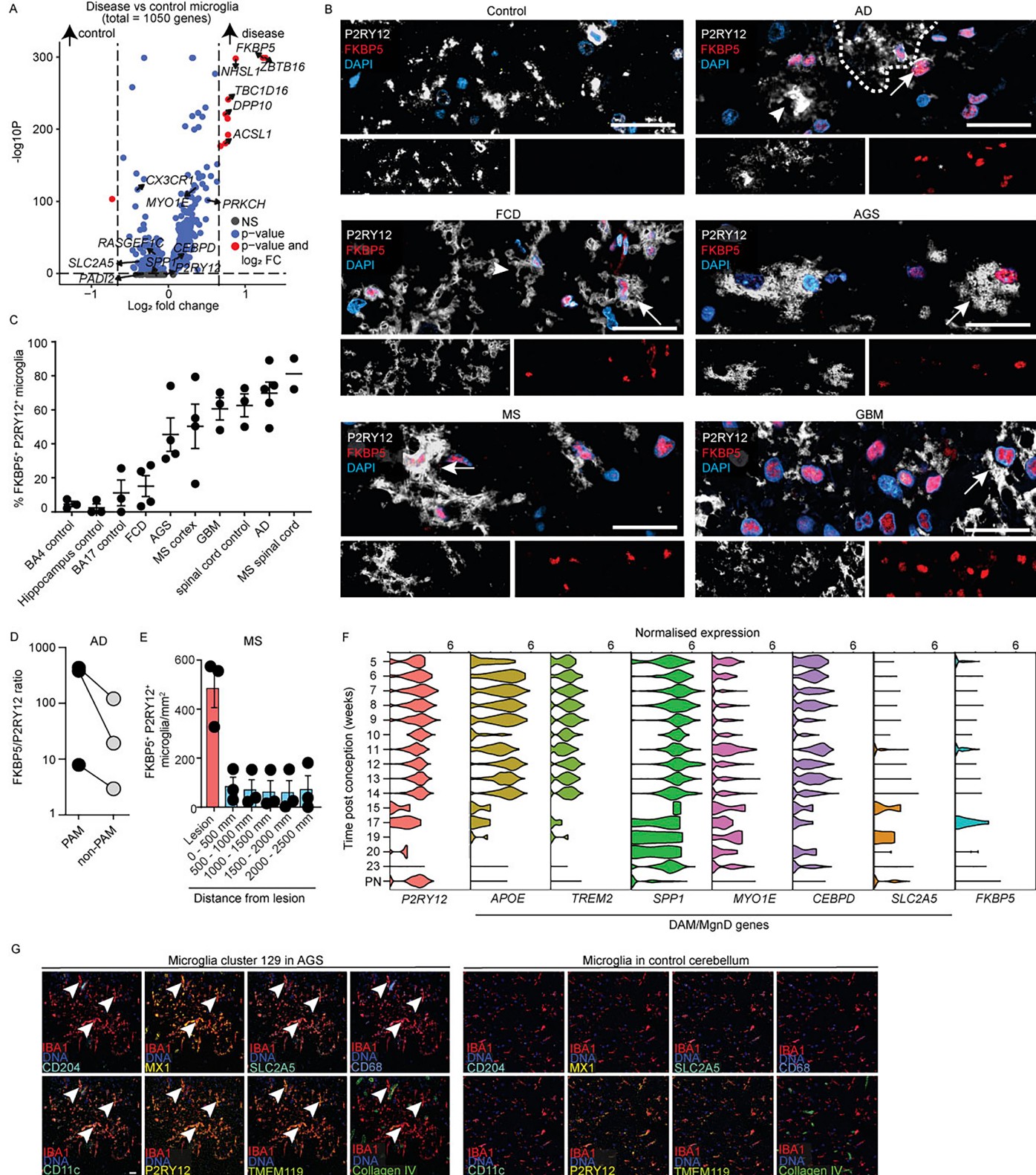

**Extended Data Fig. 3 | See next page for caption.**

**Extended Data Fig. 3 | Stress-related microglial genes are universally induced across human CNS pathologies. a** Volcano plot showing differential gene expression between disease and control human microglia across all conditions shown in Fig. 1. The $\log_2$ fold change in expression (x-axis) is plotted against the $-\log_{10}$-transformed Benjamini–Hochberg–adjusted P value (y-axis). Differential expression was assessed using two-sided Wilcoxon rank-sum tests implemented in Seurat (FindMarkers), with false discovery rate control using the Benjamini–Hochberg procedure. NS, not significant. **b** Representative immunofluorescence confocal images of FKBP5 and P2RY12 in different conditions. Ctrl = control, AD = Alzheimer's disease, FCD = Focal cortical dysplasia, AGS = Aicardi-Goutières syndrome, MS = multiple sclerosis, GBM = glioblastoma. * highlights the border of the amyloid plaque. Colors in each panel correspond to color legends within the image. Scale bars: 25 µm. Arrow heads highlight P2RY12+ microglia and arrows highlight P2RY12+FKBP5+ double positive microglia. **c** Quantification of P2RY12+FKBP5+ microglia across conditions. Each dot represents an individual patient (BA4 control, Hippocampus control, BA17 control, GBM, and spinal cord control n = 3; FCD, AGS, MS cortex, and AD, n = 4; MS spinal cord, n = 2). Data are shown as mean ± SEM. **d** Spatial distribution of P2RY12+FKBP5+ microglia in AD samples visualized as plaque-associated microglia (PAM) or non-plaque-associated microglia (Non-PAM). Each symbol represents one patient. **e** Quantification of P2RY12+FKBP5+ microglia densities in MS samples within demyelinated lesions and surrounding myelinated normal-appearing white matter, divided into zones at 500 µm increments from the lesion border. Each dot represents an individual patient (n = 3). Data are shown as mean ± SEM. Differences across zones were analyzed using one-way repeated-measures ANOVA with Geisser–Greenhouse correction and two-sided testing, followed by Dunnett's multiple-comparisons test comparing each zone to the lesion; *P < 0.05. **f** Violin plots showing normalised expression of homeostatic and activation associated genes across human embryonic development. **g** Spatial proteomics of selected microglia clusters by high-dimensional imaging mass cytometry (IMC). Left panel: representative IMC image showing IBA1 (red), CD204 (turquoise), MX1 (yellow), SLC2A5 (greenish cyan), CD68 (light blue), CD11c (mint), P2RY12 (lemon), TMEM119 (greenish-yellow) expression representing microglial cluster 129 in Aicardi-Goutières syndrome (AGS) specimen. Collagen IV (green) highlights blood vessels. DNA is shown in blue. Arrowheads indicate microglial cells. Right panel: representative IMC image showing IBA1 (red), P2RY12 (yellow), TMEM119 (greenish-yellow) expression in cells representing homeostatic microglia in cerebellum. Scale bars: 20 µm.

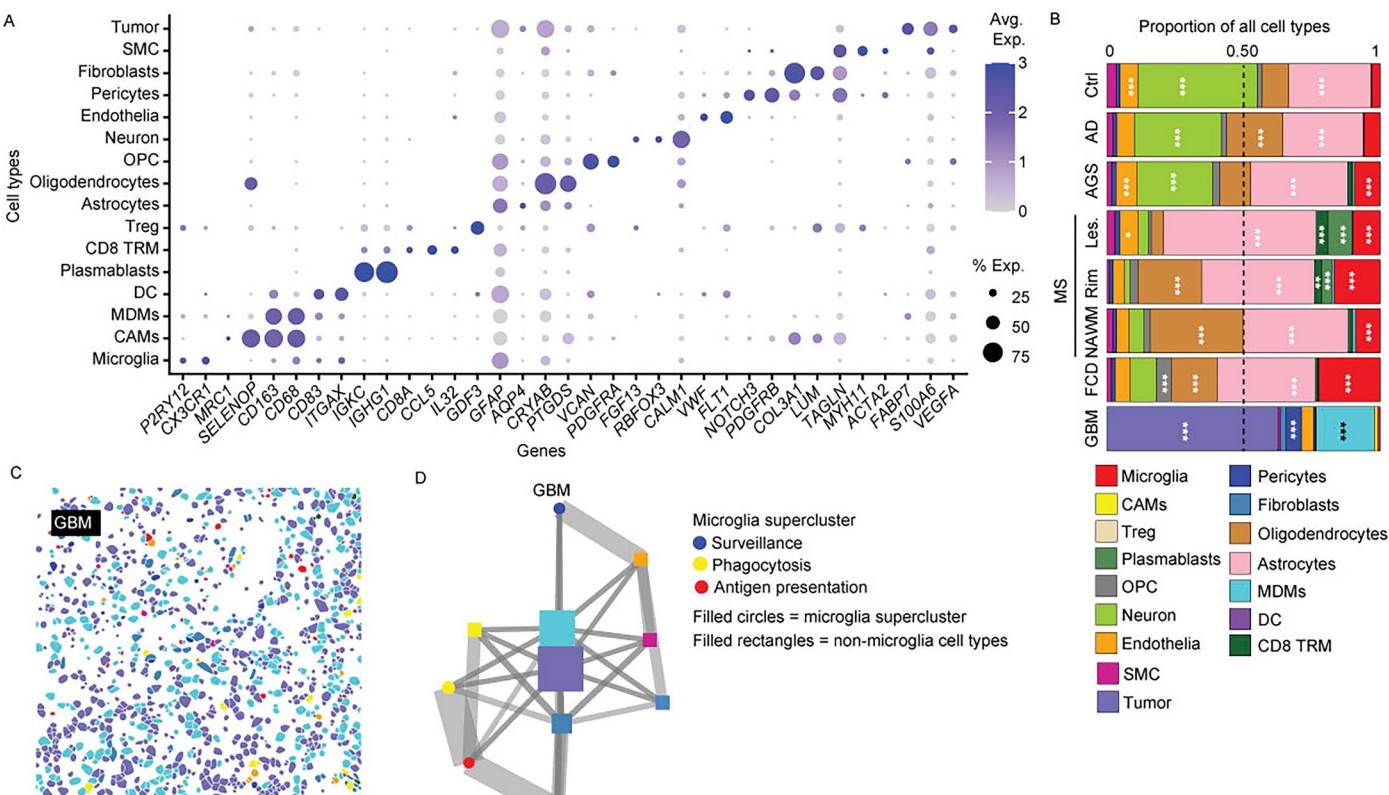

**Extended Data Fig. 4 | Marker gene expression and proportions of individual cell types in CosMxTM subcellular resolution spatial transcriptomics dataset.** **a** Dot plot representing the average expression (color intensity) and percent-expressing cells (bubble size) of selected marker genes associated with each cell types as in Fig. 2b. **b** Bar plot summarizing the proportions of each cell type across mentioned contexts. Ctrl: controls, AD: Alzheimer's disease, AGS: Aicardi-Goutières syndrome, MS: multiple sclerosis, Les.: MS lesion with complete demyelination, Rim: demyelination border region, NAWM: normal appearing white matter, FCD: Focal cortical dysplasia, GBM: glioblastoma. Colors represent individual cell types as depicted in legend. CAMs: CNS-associated macrophages, Treg: regulatory T cells, OPC: oligodendrocyte precursor cells, SMC: smooth

muscle cells. MDMs: Monocyte derived macrophages, DC: dendritic cells, CD8 TRM: resident memory CD8 T cells. One-sided hypergeometric test with BH correction; * p < 0.05, ** p < 0.01, *** p < 0.001. Significance marks are shown for a subset due to space constraints. **c** Representative spatial map of GBM displaying individual cells. Colors of cell types are same as in Fig. 2b. **d** Network diagram showing the interaction network of microglia supercluster (filled circles, coloured as in legend) with other cell types (filled rectangles, coloured as in b) in GBM from spatial data in C. The node size corresponds to the total number of cells of that type in the representative image, while the edge width reflects the proportion of neighbouring cells between connected nodes.

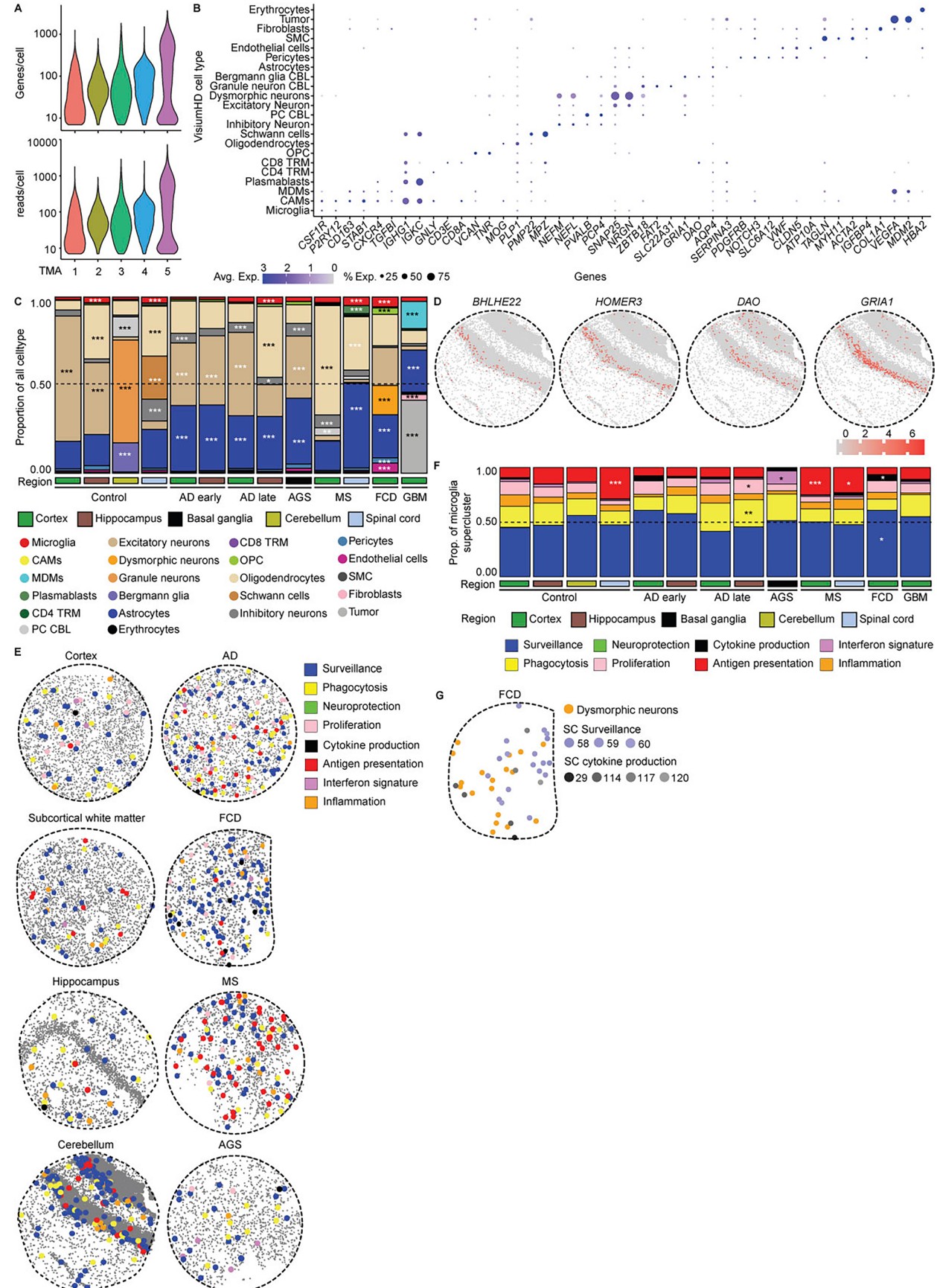

**Extended Data Fig. 5 | See next page for caption.**

**Extended Data Fig. 5 | Cell-specific marker expression and spatial organization of microglia superclusters visualized by genome-wide subcellular spatial transcriptomics. a** Violin plots showing numbers of genes detected per cell (upper graph) and reads per cell (lower graph) in Visium HD datasets. **b** Dot plot representing the average expression (color intensity) and percent expressing cells (bubble size) of selected marker genes associated with each cell types as in Fig. 4a. **c** Bar plots summarizing the relative proportions of all identified cell types across various CNS regions and disease contexts. AD: Alzheimer's disease, AGS: Aicardi-Goutières syndrome, MS: multiple sclerosis, FCD: Focal cortical dysplasia, GBM: glioblastoma. CAMs: CNS-associated macrophages, MDMs: monocyte-derived macrophages, TRM: resident memory T cells, OPC: oligodendrocyte precursor cells, PC CBL: Purkinje cells from cerebellum, SMC: smooth muscle cells. One-sided hypergeometric test with BH correction; * p < 0.05, ** p < 0.01, *** p < 0.001. Significance marks are shown for a subset due to space constraints. **d** Spatial feature plots showing the localization of selected marker genes (for example, *DAO*, *GRIA1*, *BHLHE22*, *HOMER3*) in a representative cerebellum sample, corresponding to Fig. 4c. **e** Spatial distribution of microglia superclusters colored by supercluster assignments, showing localized clustering patterns in the mentioned regions and contexts, corresponding to Fig. 4c,d. **f** Bar plots showing proportions of microglia superclusters across various CNS regions and disease contexts. AD: Alzheimer's disease, AGS: Aicardi-Goutières syndrome, MS: multiple sclerosis, FCD: Focal cortical dysplasia, GBM: glioblastoma. One-sided hypergeometric test with BH correction; * p < 0.05, ** p < 0.01, *** p < 0.001. Significance marks are shown for a subset due to space constraints. **g** Spatial plot showing distribution of surveillance and cytokine production superclusters microglia with respect to dysmorphic neurons in FCD, corresponding to Fig. 4d.

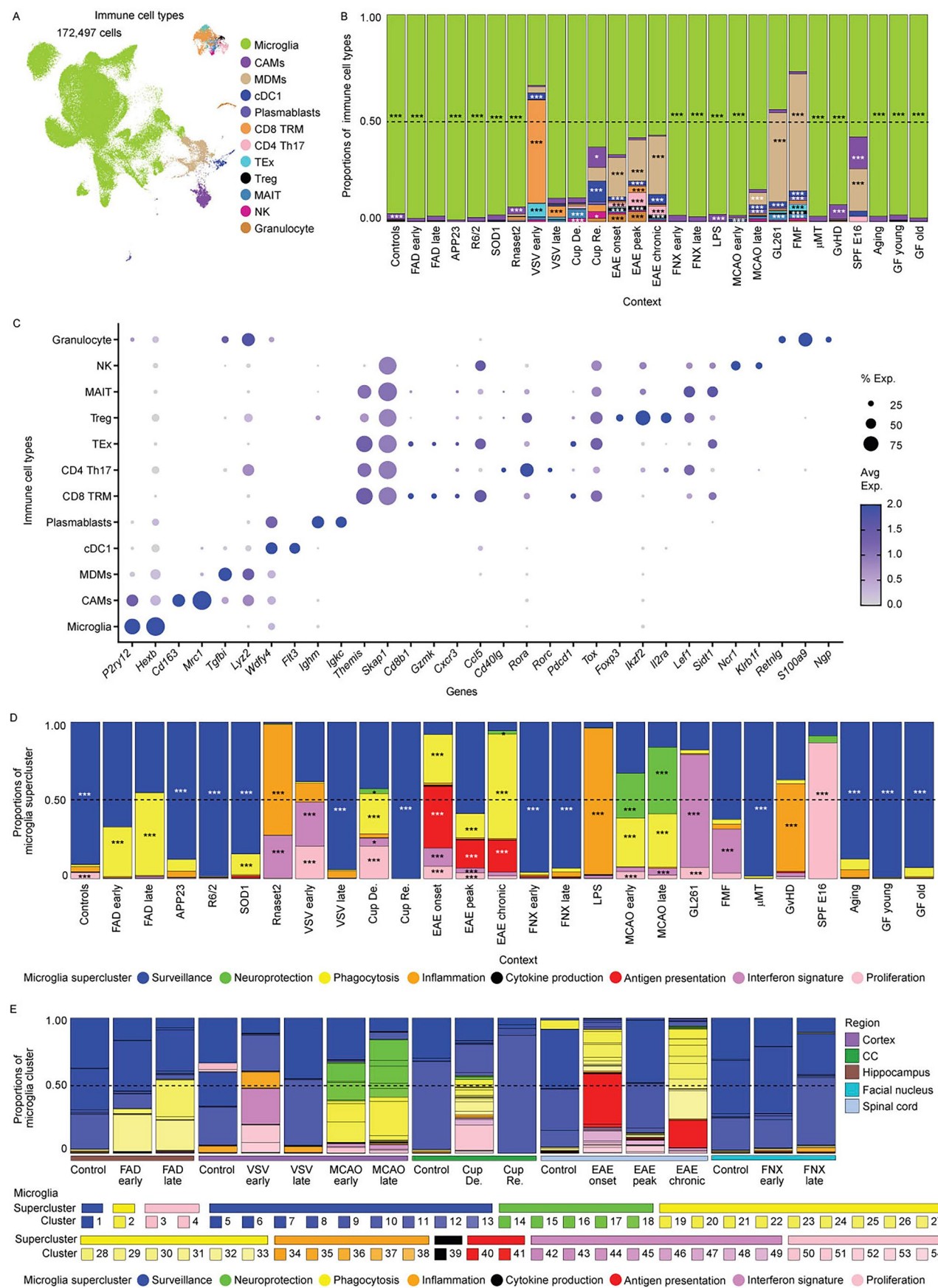

**Extended Data Fig. 6 | See next page for caption.**

**Extended Data Fig. 6 | Proportions of immune cell types in the CNS across mouse CNS disease models. a** UMAP visualization of 172,497 CNS immune cells from the mouse models of CNS diseases colored by cell types. CAMs: CNS-associated macrophages, MDMs: Monocytes-derived macrophages, cDC: conventional dendritic cells, TRM: T resident memory, TEx: T exhausted cells, Treg: regulatory T cells, MAIT: Mucosal-associated invariant T cells, NK: natural killer cells. **b** Stacked bar plot represents the relative contribution of each immune cell types to the mentioned contexts. Cell types are same as in Extended Data Fig. 6A. One-sided hypergeometric test with BH correction; * p < 0.05, ** p < 0.01, *** p < 0.001. Significance marks are shown for a subset due to space constraints. **c** Dot plot representing the average expression (color intensity) and percent-expressing cells (bubble size) of selected marker genes associated with each immune cell types. Cell types are same as in Extended Data Fig. 6A. **d** Bar plots representing distribution of microglia supercluster in different contexts. One-sided hypergeometric test with BH correction; * p < 0.05, ** p < 0.01, *** p < 0.001. **e** Bar plots representing distribution of microglia clusters in mouse models of CNS diseases across diverse stages. FAD: Familial Alzheimer's disease model, VSV: Vesicular stomatitis virus encephalitis model, MCAO: middle cerebral artery occlusion, Cup: Cuprizone model, EAE: Experimental autoimmune encephalomyelitis, FNX: Facial nerve axotomy.

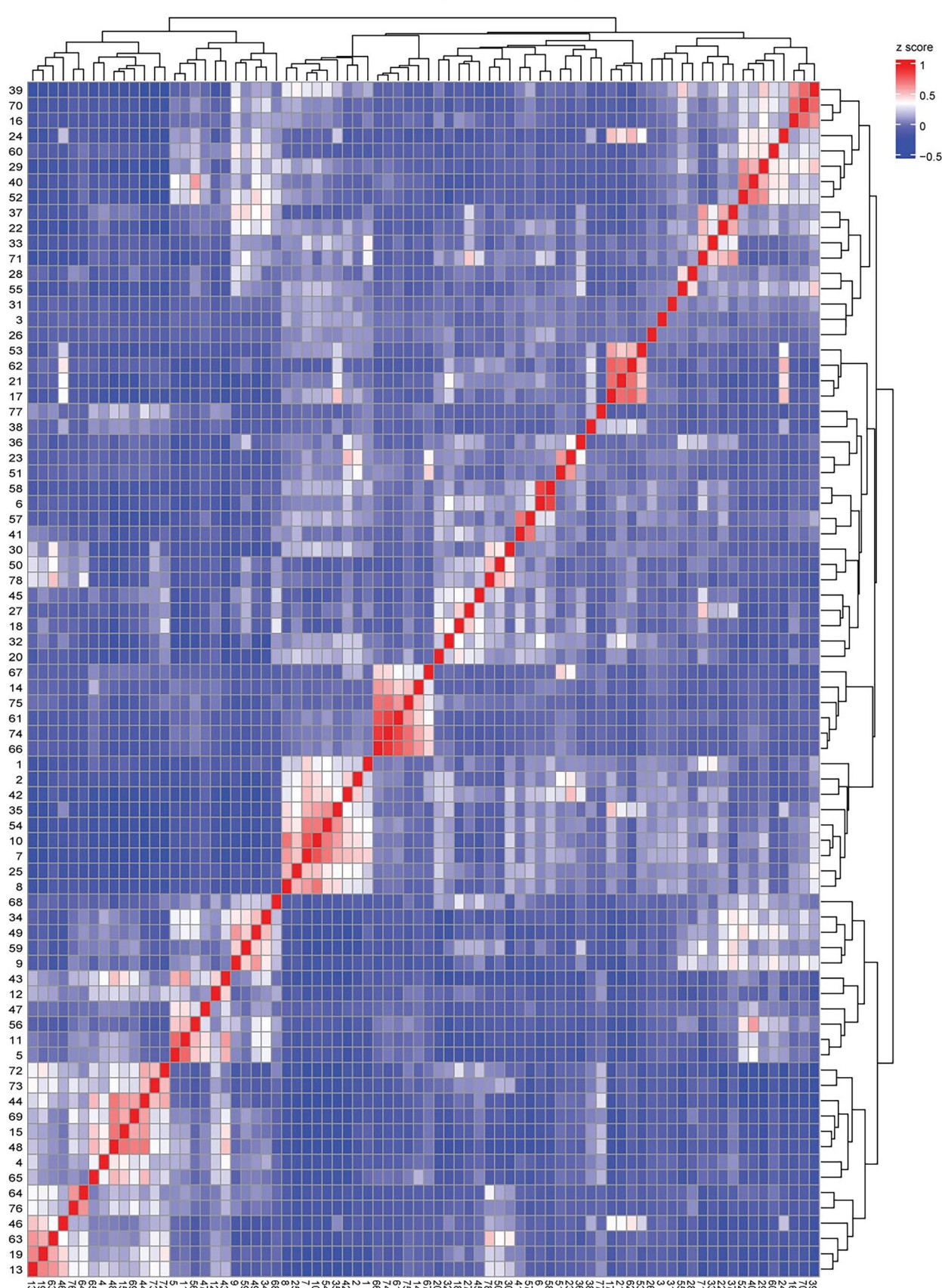

Cluster Correlation for mouse myeloid cells based on top 10 DE Genes

**Extended Data Fig. 7 | Correlation heat map of top differentially expressed genes across mouse myeloid clusters.** Heat map showing pairwise correlations of the top 10 differentially expressed (DE) genes for each mouse myeloid cell cluster identified in Fig. 5a. Correlation coefficients were calculated using normalized gene expression values across clusters. Warmer colors indicate stronger positive correlations, whereas cooler colors indicate weaker or negative correlations.

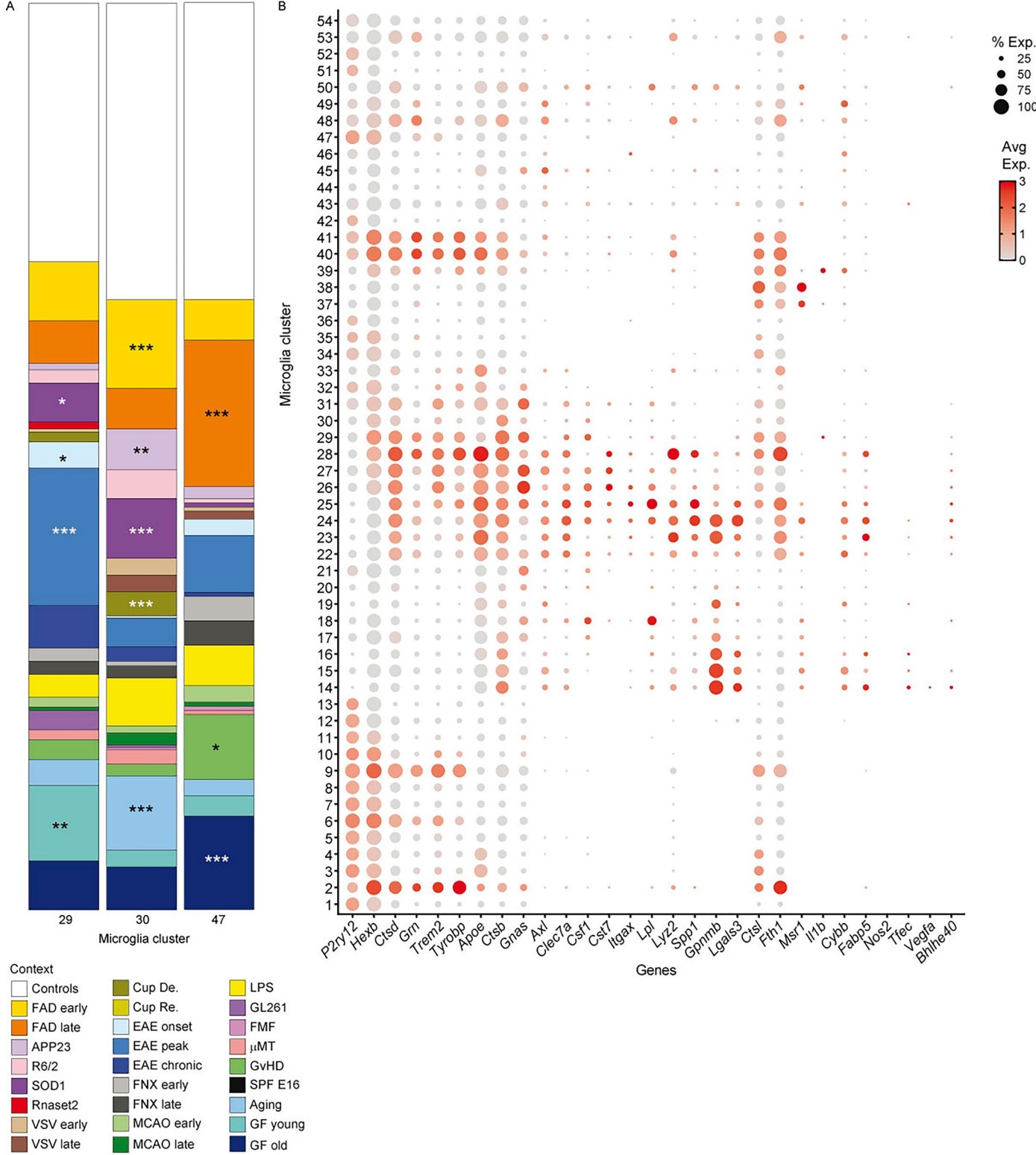

**Extended Data Fig. 8 | Distribution of commonly occurring mouse microglia clusters across contexts and shared gene expression. a** Bar plots depicting distribution of contexts across commonly occurring mouse microglia clusters. One-sided hypergeometric test with BH correction; * p < 0.05, ** p < 0.01, *** p < 0.001. **b** Dot plot representing the expression (colour intensity) and percent-expressing cells (bubble size) of previously described microglia states associated genes across microglia clusters in Fig. 5a.

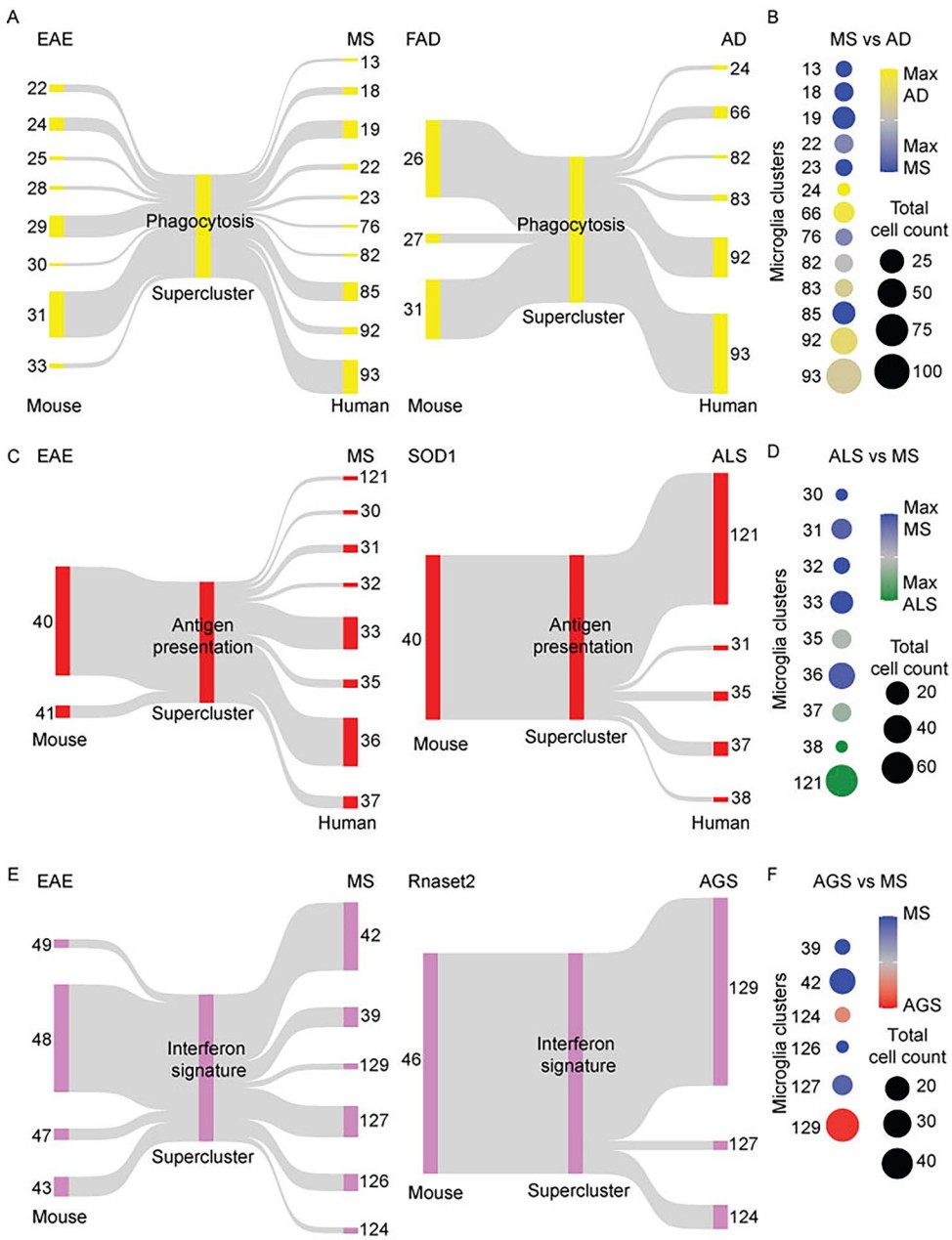

**Extended Data Fig. 9 | Cross-species Sankey maps of conserved microglial superclusters. a** Sankey alluvial diagrams illustrating individual mouse microglial clusters (left) mapped to the shared 'phagocytosis' supercluster (centre) and the corresponding human clusters (right). Left plots denote microglial clusters mapped between EAE versus MS. Right plot denotes microglial clusters mapped between FAD and AD. The width of each flow indicates the relative proportions of cells contributing to the supercluster from each cluster. **b** Dot plots depicting the number of cells (bubble size) and cluster overlap (color scale) between MS and AD derived 'phagocytosis' supercluster. The size of each bubble represents the number of cells in the cluster, while

the color scale indicates the degree of overlap between the clusters. **c** Sankey diagrams linking EAE and MS (left) or SOD1 and ALS (right) derived 'Antigen presentation' microglia supercluster. **d** Dot plot depicting the number of cells (bubble size) and cluster overlap (color scale) between ALS versus MS derived 'Antigen presentation' supercluster. **e** Sankey diagrams linking EAE and MS (left) or Rnaset2 and AGS (right) derived 'Interferon signature' microglia supercluster. **f** Dot plot depicting the number of cells (bubble size) and cluster overlap (color scale) between AGS versus MS derived 'Interferon signature' microglia supercluster.

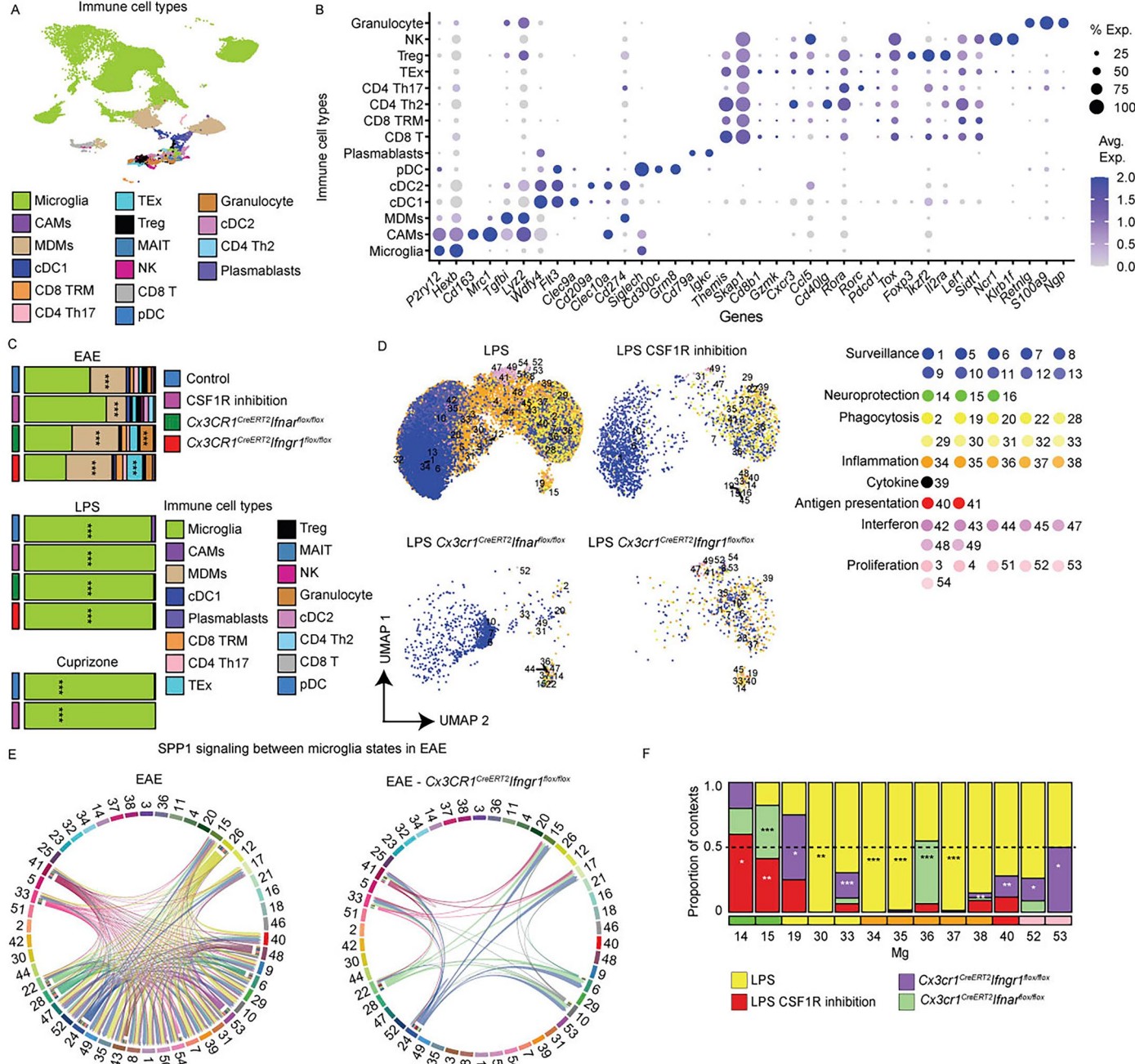

**Extended Data Fig. 10 | Context-linked dependency of microglial states on CSF-1R and IFNAR/IFNGR signalling. a** UMAP visualization of CNS immune cells from the EAE, LPS, and cuprizone demyelination with or without CSF1R, IFNAR, or IFNGR signaling inhibition colored by cell types. CAMs: CNS-associated macrophages, MDMs: monocytes-derived macrophages, cDC: conventional dendritic cells, TRM: T resident memory, TEx: T exhausted cells, Treg: regulatory T cells, MAIT: Mucosal-associated invariant T cells, NK: natural killer cells, pDC: plasmacytoid dendritic cells. **b** Dot plot representing the expression of selected marker genes associated with each immune cell type in Extended Data Figure 10A. **c** Stacked bar plot represents the relative contribution of each immune cell type to the mentioned contexts in Extended Data Figure 10A. One-sided hypergeometric test with BH correction; * p < 0.05, ** p < 0.01, *** p < 0.001. **d** UMAP visualization of single microglial clusters from LPS model. Mice underwent either CSF-1R inhibitor treatment or for a genetic deletion of IFNAR or IFNGR. Controls for each of treatment are merged and displayed together as LPS. **e** Overall proportions (%) of significantly altered microglial clusters across different conditions from (D). Dotted line represents proportion of 0.5. One-sided hypergeometric test with BH correction; * p < 0.05, ** p < 0.01, *** p < 0.001. **f** Inferred SPP1 signaling network between imputed microglial clusters during EAE in $Cx_3CR1^{CreERT2}Ifngr1^{fl/fl}$ animals from Fig. 7e.

# Reporting Summary

## Statistics

For all statistical analyses, confirm that the following items are present in the figure legend, table legend, main text, or Methods section.

| n/a | Confirmed | |
|---|---|---|
| ☐ | ☒ | The exact sample size (*n*) for each experimental group/condition, given as a discrete number and unit of measurement |
| ☐ | ☒ | A statement on whether measurements were taken from distinct samples or whether the same sample was measured repeatedly |
| ☐ | ☒ | The statistical test(s) used AND whether they are one- or two-sided<br>*Only common tests should be described solely by name; describe more complex techniques in the Methods section.* |
| ☒ | ☐ | A description of all covariates tested |
| ☐ | ☒ | A description of any assumptions or corrections, such as tests of normality and adjustment for multiple comparisons |
| ☐ | ☒ | A full description of the statistical parameters including central tendency (e.g. means) or other basic estimates (e.g. regression coefficient) AND variation (e.g. standard deviation) or associated estimates of uncertainty (e.g. confidence intervals) |
| ☐ | ☒ | For null hypothesis testing, the test statistic (e.g. *F*, *t*, *r*) with confidence intervals, effect sizes, degrees of freedom and *P* value noted<br>*Give P values as exact values whenever suitable.* |
| ☒ | ☐ | For Bayesian analysis, information on the choice of priors and Markov chain Monte Carlo settings |
| ☒ | ☐ | For hierarchical and complex designs, identification of the appropriate level for tests and full reporting of outcomes |
| ☒ | ☐ | Estimates of effect sizes (e.g. Cohen's *d*, Pearson's *r*), indicating how they were calculated |

*Our web collection on statistics for biologists contains articles on many of the points above.*

## Software and code

Policy information about availability of computer code

| Data collection | Sequencing and fastq file generation:<br>NextSeq 550 Control Software (NCS) v4.0.1.41 for the Illumina NextSeq 550 instrument<br>bcl2fastq v2.20 software for conversion of .bcl files into fastq files<br>NextSeq 1000/2000 Control Software (NCS) v1.2.0.36376 for the Illumina NextSeq 1000/2000 instrument<br>DRAGEN v3.8.4 software for sequencing and conversion of .bcl files into fastq files<br><br>Data processing:<br>Processed data such as counts matrix were generated with cellranger-7.1.0 with the GENCODE human genome release 35 or mouse genome release 33 as reference genomes.<br>For alignment of Visium HD data spaceranger-3.0.0. was used with Visium Human Transcriptome Probe Set v2.0 and GRCh38 human reference genome<br><br>Microscopy:<br>Fluorescence imaging: Olympus BX-61 with a color camera (Olympus DP71)<br>Pannoramic 250 FLASH II (3DHISTECH) Digital Slide Scanner<br>BZ-9000 (Keyence)<br><br>Confocal imaging:<br>TCS SP8 X (Leica, LAS X 3.5.7.23225) microscope using a 20x 0.75 NA (HC PL APO CS2 20x/0.75 IMM) objective<br>Zeiss LSM800 microscope |
|---|---|

Mass Cytometry:
Hyperion Imaging Mass Cytometry system (Fluidigm)

Data analysis

R environment:
R version 4.4.1 (2024-06-14)
Platform: x86_64-pc-linux-gnuRunning under: Ubuntu 22.04.5 LTSMatrix products: defaultBLAS:   /usr/lib/x86_64-linux-gnu/openblas-pthread/libblas.so.3 LAPACK: /usr/lib/x86_64-linux-gnu/openblas-pthread/libopenblasp-r0.3.20.so;  LAPACK version 3.10.0locale: [1] LC_CTYPE=en_US.UTF-8       LC_NUMERIC=C             LC_TIME=de_DE.UTF-8      LC_COLLATE=en_US.UTF-8      LC_MONETARY=de_DE.UTF-8 [6] LC_MESSAGES=en_US.UTF-8      LC_PAPER=de_DE.UTF-8       LC_NAME=C               LC_ADDRESS=C              LC_TELEPHONE=C          [11] LC_MEASUREMENT=de_DE.UTF-8 LC_IDENTIFICATION=C      time zone: Europe/Berlintzcode source: system (glibc)attached base packages:[1] grid     stats4   stats   graphics grDevices utils    datasets methods  base     other attached packages: [1] SoupX_1.6.2                magrittr_2.0.3            CellChat_2.1.2           DoubletFinder_2.0.4       reshape2_1.4.4         [6] arrow_18.1.0.1            scales_1.3.0             gridExtra_2.3            circlize_0.4.16          ineq_0.2-13           [11] igraph_2.0.3             UCell_2.8.0             ComplexHeatmap_2.21.1    data.table_1.16.4        Azimuth_0.5.0        [16] shinyBS_0.61.1          RColorBrewer_1.1-3       SeuratWrappers_0.3.5     SeuratData_0.2.2.9001    EnhancedVolcano_1.22.0 [21] ggrepel_0.9.6           monocle3_1.3.4          SingleCellExperiment_1.26.0  SummarizedExperiment_1.34.0 GenomicRanges_1.56.2 [26] GenomeInfoDb_1.40.1       IRanges_2.38.1          S4Vectors_0.42.1        MatrixGenerics_1.16.0     matrixStats_1.5.0   [31] Biobase_2.64.0          BiocGenerics_0.50.0      ggtree_3.12.0           assertthat_0.2.1        clustree_0.5.1     [36] ggraph_2.2.1            Matrix_1.6-5            viridis_0.6.5           viridisLite_0.4.2       lubridate_1.9.4    [41] forcats_1.0.0           stringr_1.5.1          purrr_1.0.2            readr_2.1.5            tibble_3.2.1       [46] tidyverse_2.0.0         harmony_1.2.3          Rcpp_1.0.14           patchwork_1.3.0        cowplot_1.1.3     [51] ggplot2_3.5.1           reticulate_1.40.0       pheatmap_1.0.12         tidyr_1.3.1           dplyr_1.1.4       [56] Seurat_5.2.1            SeuratObject_5.0.2      sp_2.2-0
loaded via a namespace (and not attached): [1] ica_1.0-3             plotly_4.10.4          Formula_1.2-5          zlibbioc_1.50.0      [5] tidyselect_1.2.1       bit_4.5.0.1           doParallel_1.0.17       clue_0.3-66           [9] lattice_0.22-5      rjson_0.2.23          blob_1.2.4           rngtools_1.5.2         [13] S4Arrays_1.4.1         parallel_4.4.1      seqLogo_1.70.0        png_0.1-8            [17] cli_3.6.3            ggplotify_0.1.2        registry_0.5-1    ProtGenerics_1.36.0      [21] goftest_1.2-3         gargle_1.5.2          BiocIO_1.14.0          BiocNeighbors_1.22.0 [25] ggnetwork_0.5.13        Signac_1.14.0         uwot_0.2.2            curl_6.2.0           [29] mime_0.12      tidytree_0.4.6        stringi_1.8.4         backports_1.5.0        [33] XML_3.99-0.18        httpuv_1.6.15     AnnotationDbi_1.66.0      rappdirs_0.3.3        [37] splines_4.4.1        RcppRoll_0.3.1        DT_0.33      sctransform_0.4.1       [41] DBI_1.2.3            terra_1.8-21          jquerylib_0.1.4        withr_3.0.2           [45] systemfonts_1.2.1       reformulas_0.4.0        lmtest_0.9-40         tidygraph_1.3.1         [49] rtracklayer_1.64.0 BiocManager_1.30.25      htmlwidgets_1.6.4      fs_1.6.5            [53] biomaRt_2.60.1         statnet.common_4.11.0 SparseArray_1.4.8       cellranger_1.1.0       [57] annotate_1.82.0        zoo_1.8-12           JASPAR2020_0.99.10 XVector_0.44.0         [61] network_1.19.0         TFBSTools_1.42.0        UCSC.utils_1.0.0        TFMPvalue_0.0.9 [65] timechange_0.3.0        foreach_1.5.2          caTools_1.18.3         rhdf5_2.48.0          [69] pwalign_1.0.0 R.oo_1.27.0          poweRlaw_1.0.0         RSpectra_0.16-2        [73] irlba_2.3.5.1         fastDummies_1.7.5     gridGraphics_0.5-1       lazyeval_0.2.2        [77] yaml_2.3.10           survival_3.7-0         scattermore_1.2   crayon_1.5.3          [81] RcppAnnoy_0.0.22        progressr_0.15.1        tweenr_2.0.3          later_1.4.1           [85] ggridges_0.5.6         codetools_0.2-19        GlobalOptions_0.1.2      KEGGREST_1.44.1        [89] Rtsne_0.17  shape_1.4.6.1          Rsamtools_2.20.0        filelock_1.0.3        [93] pkgconfig_2.0.3        xml2_1.3.6      spatstat.univar_3.1-1      ggpubr_0.6.0          [97] GenomicAlignments_1.40.0   aplot_0.2.4          spatstat.sparse_3.1-0 BSgenome_1.72.0        [101] ape_5.8-1            gridBase_0.4-7         xtable_1.8-4          car_3.1-3            [105] plyr_1.8.9          httr_1.4.7           rbibutils_2.3         tools_4.4.1          [109] globals_0.16.3        broom_1.0.7 nlme_3.1-165         dbplyr_2.5.0         [113] hdf5r_1.3.10         shinyjs_2.1.0         lme4_1.1-36 digest_0.6.37         [117] farver_2.1.2         tzdb_0.4.0          AnnotationFilter_1.28.0      yulab.utils_0.2.0       [121] DirichletMultinomial_1.46.0 glue_1.8.0          cachem_1.1.0          BiocFileCache_2.12.0    [125] polyclip_1.10-7 generics_0.1.3         Biostrings_2.72.1       ggalluvial_0.12.5        [129] googledrive_2.1.1        presto_1.0.0 parallelly_1.42.0      RcppHNSW_0.6.0        [133] carData_3.0-5         minqa_1.2.8          pbapply_1.7-2 httr2_1.1.0          [137] spam_2.11-1          graphlayouts_1.2.2       gtools_3.9.5          ggsignif_0.6.4        [141] shiny_1.10.0          GenomeInfoDbData_1.2.12    R.utils_2.12.3        rhdf5filters_1.16.0      [145] RCurl_1.98-1.16 memoise_2.0.1         R.methodsS3_1.8.2        googlesheets4_1.1.1      [149] svglite_2.1.3        future_1.34.0 RANN_2.6.2          Cairo_1.6-2          [153] spatstat.data_3.1-4      rstudioapi_0.17.1       cluster_2.1.6 spatstat.utils_3.1-2      [157] hms_1.1.3           fitdistrplus_1.2-2       munsell_0.5.1         colorspace_2.1-1       [161] FNN_1.1.4.1          rlang_1.1.5          dotCall64_1.2         shinydashboard_0.7.2    [165] ggforce_0.4.2 coda_0.19-4          sna_2.8             CNEr_1.40.0          [169] remotes_2.5.0        iterators_1.0.14 abind_1.4-8          EnsDb.Hsapiens.v86_2.99.0   [173] treeio_1.28.0         Rhdf5lib_1.26.0        bitops_1.0-9 Rdpack_2.6.2         [177] promises_1.3.2        RSQLite_2.3.9         DelayedArray_0.30.1      GO.db_3.19.1 [181] compiler_4.4.1        prettyunits_1.2.0       boot_1.3-30          listenv_0.9.1         [185] BSgenome.Hsapiens.UCSC.hg38_1.4.5 tensor_1.5          MASS_7.3-61          progress_1.2.3        [189] BiocParallel_1.38.0 spatstat.random_3.3-2      R6_2.5.1            fastmap_1.2.0        [193] fastmatch_1.1-6        rstatix_0.7.2 ensembldb_2.28.1       ROCR_1.0-11          [197] SeuratDisk_0.0.0.9021     rsvd_1.0.5           gtable_0.3.6 KernSmooth_2.23-24      [201] miniUI_0.1.1.1        deldir_2.0-4         htmltools_0.5.8.1       bit64_4.6.0-1 [205] spatstat.explore_3.3-4     lifecycle_1.0.4        nloptr_2.1.1         restfulr_0.0.15       [209] sass_0.4.9 vctrs_0.6.5          spatstat.geom_3.3-5       NMF_0.28            [213] ggfun_0.1.8          future.apply_1.11.3 bslib_0.9.0          pillar_1.10.1         [217] GenomicFeatures_1.56.0    magick_2.8.5          jsonlite_1.8.9 GetoptLong_1.0.5

Python environment:
Package,VersionBabel,2.15.0Brotli,1.1.0GDAL,3.9.0HeapDict,1.0.1Markdown,3.6MarkupSafe,2.1.5PIMS,0.6.1Pint,0.24PyOpenGL,3.1.7PyQt5,5.15.9PyQt5-sip,12.12.2PySocks,1.7.1PyWavelets,1.4.1PyYAML,6.0.1QtPy,2.4.1Rtree,1.2.0absl-py,2.1.0aiobotocore,2.5.4aiohttp,3.9.5aioitertools,0.11.0aiosignal,1.3.1alabaster,0.7.16anndata,0.10.7annotated-types,0.7.0app-model,0.2.7appdirs,1.4.4array-api-compat,1.7.1asciitree,0.3.3asttokens,2.4.1astunparse,1.6.3attrs,23.2.0bin2cell,0.1.1bokeh,3.4.1botocore,1.31.17branca,0.7.2build,1.2.1cached-property,1.5.2cachey,0.2.1certifi,2024.6.2charset-normalizer,3.3.2click,8.1.7click-plugins,1.1.1cligj,0.7.2cloudpickle,3.0.0colorama,0.4.6colorcet,3.1.0comm,0.2.2contourpy,1.2.1cramjam,2.8.3csbdeep,0.8.0cycler,0.12.1cyto

olz,0.12.3dask,2024.2.1dask-expr,1.1.2dask-image,2023.8.1datashader,0.16.2debugpy,1.8.1decorator,5.1.1distributed,2024.2.1docrep,0.3.2docstring-parser,0.16docutils,0.21.2exceptiongroup,1.2.0executing,2.0.1fasteners,0.17.3fastparquet,2024.5.0fcsparser,0.2.8filelock,3.15.3fiona,1.9.6flatbuffers,24.3.25flexcache,0.3flexparser,0.3.1folium,0.16.0fonttools,4.53.0freetype-py,2.4.0frozenlist,1.4.1fsspec,2023.6.0gast,0.5.4geopandas,0.14.4get-annotations,0.1.2google-pasta,0.2.0grpcio,1.62.2h5py,3.11.0hsluv,5.0.4idna,3.7igraph,0.11.5imagecodecs,2024.6.1imageio,2.34.1imagesize,1.4.1importlib-metadata,7.1.0in-n-out,0.2.1inflect,7.2.1ipykernel,6.29.4ipython,8.25.0jedi,0.19.1jinja2,3.1.4jmespath,1.0.1joblib,1.4.2jsonschema,4.22.0jsonschema-specifications,2023.12.1jupyter-client,8.6.2jupyter-core,5.7.2keras,3.3.3kiwisolver,1.4.5lamin-utils,0.13.2lazy-loader,0.4legacy-api-wrap,1.4leidenalg,0.10.2llvmlite,0.42.0locket,1.0.0loguru,0.7.2louvain,0.8.2lz4,4.3.3magicgui,0.8.3mapclassify,2.6.1markdown-it-py,3.0.0matplotlib,3.8.4matplotlib-inline,0.1.7matplotlib-scalebar,0.8.1mdurl,0.1.2ml-dtypes,0.3.2more-itertools,10.3.0mpmath,1.3.0msgpack,1.0.8multidict,6.0.5multipledispatch,1.0.0multiscale-spatial-image,0.11.2munkres,1.1.4namex,0.0.8napari,0.4.19.post1napari-console,0.0.9napari-matplotlib,2.0.1napari-plugin-engine,0.2.0napari-spatialdata,0.4.1napari-svg,0.1.10natsort,8.4.0nest-asyncio,1.6.0networkx,3.3npe2,0.7.5numba,0.59.1numcodecs,0.12.1numpy,1.26.4numpydoc,1.7.0nvidia-cublas-cu12,12.1.3.1nvidia-cuda-cupti-cu12,12.1.105nvidia-cuda-nvrtc-cu12,12.1.105nvidia-cuda-runtime-cu12,12.1.105nvidia-cudnn-cu12,8.9.2.26nvidia-cufft-cu12,11.0.2.54nvidia-curand-cu12,10.3.2.106nvidia-cusolver-cu12,11.4.5.107nvidia-cusparse-cu12,12.1.0.106nvidia-nccl-cu12,2.20.5nvidia-nvjitlink-cu12,12.5.40nvidia-nvtx-cu12,12.1.105ome-zarr,0.9.0omnipath,1.0.8opencv-python,4.10.0.84opt-einsum,3.3.0optree,0.11.0packaging,24.1pandas,2.2.2param,2.1.0parso,0.8.4partd,1.4.2patsy,0.5.6pexpect,4.9.0pillow,10.3.0pip,24.0platformdirs,4.2.2ply,3.11pooch,1.8.2prompt-toolkit,3.0.47protobuf,4.25.3psutil,5.9.8psygnal,0.11.1ptyprocess,0.7.0pure-eval,0.2.2pyarrow,16.1.0pyarrow-hotfix,0.6pyconify,0.1.6pyct,0.5.0pydantic,2.7.4pydantic-compat,0.1.2pydantic-core,2.18.4pygeos,0.14pygments,2.18.0pynndescent,0.5.12pyparsing,3.1.2pyproj,3.6.1pyproject-hooks,1.1.0python-dateutil,2.9.0pytz,2024.1pyzmq,26.0.3qtconsole,5.5.2readfcs,1.1.8referencing,0.35.1requests,2.32.3rich,13.7.1rpds-py,0.18.1s3fs,2023.6.0scanpy,1.10.1scikit-image,0.23.2scikit-learn,1.5.0scipy,1.13.1seaborn,0.13.2session-info,1.0.0setuptools,70.0.0shapely,2.0.4shellingham,1.5.4sip,6.7.12six,1.16.0slicerator,1.1.0snowballstemmer,2.2.0sortedcontainers,2.4.0spatial-image,0.3.0spatialdata,0.1.2spatialdata-io,0.1.2spatialdata-plot,0.2.2sphinx,7.3.7sphinxcontrib-applehelp,1.0.8sphinxcontrib-devhelp,1.0.6sphinxcontrib-htmlhelp,2.0.5sphinxcontrib-jsmath,1.0.1sphinxcontrib-qthelp,1.0.7sphinxcontrib-serializinghtml,1.1.10squidpy,1.2.2stack-data,0.6.3stardist,0.9.1statsmodels,0.14.2stdlib-list,0.10.0superqt,0.6.7sympy,1.12.1tabulate,0.9.0tblib,3.0.0tensorboard,2.16.2tensorboard-data-server,0.7.0tensorflow,2.16.1tensorflow-estimator,2.15.0termcolor,2.4.0texttable,1.7.0threadpoolctl,3.5.0tifffile,2024.5.22tinycss2,1.3.0toml,0.10.2tomli,2.0.1tomli-w,1.0.0toolz,0.12.1torch,2.3.1tornado,6.4.1tqdm,4.66.4traitlets,5.14.3triton,2.3.1typeguard,4.3.0typer,0.12.3typing-extensions,4.12.2tzdata,2024.1umap-learn,0.5.5urllib3,1.26.19validators,0.28.3vispy,0.14.3wcwidth,0.2.13webencodings,0.5.1werkzeug,3.0.3wheel,0.43.0wrapt,1.16.0xarray,2024.6.0xarray-dataclasses,1.8.0xarray-datatree,0.0.14xarray-schema,0.0.3xarray-spatial,0.4.0xyzservices,2024.6.0yarl,1.9.4zarr,2.18.2zict,3.0.0zipp,3.19.2

For manuscripts utilizing custom algorithms or software that are central to the research but not yet described in published literature, software must be made available to editors and reviewers. We strongly encourage code deposition in a community repository (e.g. GitHub). See the Nature Portfolio guidelines for submitting code & software for further information.

# Data

Policy information about availability of data

All manuscripts must include a data availability statement. This statement should provide the following information, where applicable:
- Accession codes, unique identifiers, or web links for publicly available datasets
- A description of any restrictions on data availability
- For clinical datasets or third party data, please ensure that the statement adheres to our policy

All sequencing data and processed seurat objects are publicly available. Human raw sequencing data are deposited at the European Genome–Phenome Archive (EGA; accession EGAS50000001289; https://ega-archive.org/studies/EGAS50000001289). Mouse raw sequencing data are available at the NCBI Gene Expression Omnibus (GEO; accession GSE304010; https://www.ncbi.nlm.nih.gov/geo/query/acc.cgi?&acc=GSE304010). Processed Seurat objects and the code to reproduce the figures are deposited at zenodo (DOI: 10.5281/zenodo.16938034; https://zenodo.org/uploads/16938034).

# Research involving human participants, their data, or biological material

Policy information about studies with human participants or human data. See also policy information about sex, gender (identity/presentation), and sexual orientation and race, ethnicity and racism.

| Reporting on sex and gender | Sex and gender were not disaggregated in the present study. Since the presented data is on cell identity and states it is expected to apply regardless of sex and therefore both sexes were included in the study. Individual datasets, except for lung brain metastasis, mostly consisted of cells derived from multiple participants and both sexes. We integrated the data for sex to minize the impact of sex on clustering. Details for integration are provided in methods section. |
|---|---|
| Reporting on race, ethnicity, or other socially relevant groupings | Limited information was available of the anonymized patients. |
| Population characteristics | The age range for samples was between the prenatal to 96 years. The tissue from control patients were at least 2 cm from the pathological focus and radiologically and histologically normal. |
| Recruitment | Participants were not prospectively recruited for this study. Human brain tissue samples were obtained either obtained during surgery or retrospectively from established brain biobanks, which collect post-mortem donations with prior consent from donors or their families. Case inclusion was therefore determined by tissue availability and existing clinical/pathological |

metadata rather than experimental recruitment criteria.
This approach introduces potential selection biases inherent to biobank-based studies. Donors may not be fully representative of the general population, as brain donation is influenced by factors such as disease severity, healthcare access, socioeconomic background, and willingness to participate in research. Additionally, variability in post-mortem interval, tissue preservation, and clinical documentation may introduce technical and metadata-related heterogeneity. These factors may limit generalizability but are unlikely to systematically bias within-cohort molecular comparisons, as all samples were processed using standardized protocols and analyzed under consistent experimental and computational workflows.

| | |
|---|---|
| Ethics oversight | The study protocol was approved by the Ethics committee of the University of Freiburg Medical Center and local ethics committees. |

Note that full information on the approval of the study protocol must also be provided in the manuscript.

# Field-specific reporting

Please select the one below that is the best fit for your research. If you are not sure, read the appropriate sections before making your selection.

☒ Life sciences ☐ Behavioural & social sciences ☐ Ecological, evolutionary & environmental sciences

For a reference copy of the document with all sections, see nature.com/documents/nr-reporting-summary-flat.pdf

# Life sciences study design

All studies must disclose on these points even when the disclosure is negative.

| | |
|---|---|
| Sample size | The samples were acquired prospectively. No calculation of the sample size was performed. The sample size is in line with previous peer-reviewed studies for individual diseases (e.g. Movahedi et al, Nat Neurosci 2021; Friebel et al, Cell 2021; Klemm et al Cell 2021; Sankowski et al, Nat Neurosci 2019, Masuda et al. 2019). |
| Data exclusions | To minimize technical artifacts cells with unusual gene expression profiles were excluded. We also employed cut offs for number of transcripts and genes detected per cell across all the single cell and spatial transcriptomics datasets which are similar to previously published studies and comprehensively described in methods section. |
| Replication | We have included more than 3 patients per disease when available. Similarly for mouse samples we have harvested tissues from at least 3 animals. Additionally, the snRNA-seq data were validated by several orthogonal methods for example at protein levels using mass cytometry and immunohistochemistry as well as RNAscope based on smFISH with multiple biological replicates. We also validated the presence of several microglial transcriptional states in situ in patient samples via Nanostring CosMx and and Visium HD based single cell spatial transcriptomics. |
| Randomization | Randomization was not performed as the study does not include an intervention. Samples were collected prospectively and stratified according to the radiological and histological appearance as normal or pathological. |
| Blinding | Microscopy analyses were conducted by blinded experimenters. scRNA-Seq and mass cytometry data were analyzed by algorithms in an unsupervised manner. Due to the unsupervised manner of the analysis, blinding is not expected to affect the results. |

# Reporting for specific materials, systems and methods

We require information from authors about some types of materials, experimental systems and methods used in many studies. Here, indicate whether each material, system or method listed is relevant to your study. If you are not sure if a list item applies to your research, read the appropriate section before selecting a response.

## Materials & experimental systems

| n/a | Involved in the study |
|---|---|
| ☐ | ☒ Antibodies |
| ☒ | ☐ Eukaryotic cell lines |
| ☒ | ☐ Palaeontology and archaeology |
| ☐ | ☒ Animals and other organisms |
| ☒ | ☐ Clinical data |
| ☒ | ☐ Dual use research of concern |
| ☒ | ☐ Plants |

## Methods

| n/a | Involved in the study |
|---|---|
| ☒ | ☐ ChIP-seq |
| ☐ | ☒ Flow cytometry |
| ☒ | ☐ MRI-based neuroimaging |

## Antibodies

| | |
|---|---|
| Antibodies used | Anti-Olig2 Antibody, clone 211F1.1, Alexa Fluor®488 Conjugate Merck: Cat no. MABN50A4<br>RBFOX3/NeuN Antibody, clone 1B7, Alexa Fluor® 647 Conjugate Novus biologicals: Cat no. NBP1-92693AF647<br>Recombinant Anti-TMEM119 antibody clone 28-3 Abcam: Cat no. ab209064 |

Alexa Fluor® 647 anti-mouse CD74 (CLIP) Antibody clone In1/CD74 Biolegend: Cat no. 151004

FKBP5 Polyclonal antibody Proteintech Cat. No. 14155-1-AP
Anti-P2RY12 antibody Sigma-Aldrich Cat. No. HPA014518
Human Osteoactivin/GPNMB Antibody RD Systems Cat. No. AF2550

Secondary antibodies
anti-rabbit Alexa Flour® 488, Thermofisher scientific Cat no: A-21206
Rabbit Anti-Goat Immunoglobulins/Biotin DAKO Cat No. E0466
donkey anti-rabbit-AF647 Life Technologies Cat. No. A31573

| | |
|---|---|
| Validation | Immunohistochemistry antibodies were validated using positive and negative control stainings following the manufacturer's instructions. The antibodies were validated at different dilutions and pretreatments by experienced staff at the histology lab of the department. FACS antibodies were validated according to the manufacturer's instructions using all relevant isotype controls and fluorescence minus one controls. |

# Animals and other research organisms

Policy information about studies involving animals; ARRIVE guidelines recommended for reporting animal research, and Sex and Gender in Research

| | |
|---|---|
| Laboratory animals | C57BL/6 WT mice or transgenic mice on C57BL/6 background were used for experiments. For experiments with transgenic animals, either WT littermates or cre negative litter mates were used. Depending on the model mice at different ages were used, the details for which are provided in the Supplemental table 1 of the manuscript. |
| Wild animals | Study did not invlove wild animals. |
| Reporting on sex | Only female mice were used in experiments. |
| Field-collected samples | Study did not involve samples collected from the field. |
| Ethics oversight | All animal experiments were approved and performed in accordance with national and institutional regulations (Regierungspräsidium Freiburg, approval numbers X17/01A, X-20/01A, G-19/084, G-18/044, G-19/124, G-20/49, G-22/094, G20/131, G-17/063; LAVES approval numbers 17-2697, 16/2338; Landesamt für Gesundheit und Soziales, Berlin, Registration numbers: G0312/16 and G0167/20, LAGeSO approval number G0031/21). |

Note that full information on the approval of the study protocol must also be provided in the manuscript.

# Plants

| | |
|---|---|
| Seed stocks | Not applicable |
| Novel plant genotypes | Not applicable |
| Authentication | Not applicable |

# Flow Cytometry

## Plots

Confirm that:

☐ The axis labels state the marker and fluorochrome used (e.g. CD4-FITC).

☐ The axis scales are clearly visible. Include numbers along axes only for bottom left plot of group (a 'group' is an analysis of identical markers).

☐ All plots are contour plots with outliers or pseudocolor plots.

☐ A numerical value for number of cells or percentage (with statistics) is provided.

## Methodology

| | |
|---|---|
| Sample preparation | Single-nucleus suspensions were prepared using the Frankenstein community protocol (https://www.protocols.io/view/frankenstein-protocol-for-nuclei-isolation-from-f-5jyl8nx98l2w/v2 accessed on July 1st 2022). |

| | |
|---|---|
| Instrument | Becton Dickinson FACSAria III, 17 color (Lasers: 375/405 nm, 488 nm, 561 nm, 633 nm) |
| Software | *Describe the software used to collect and analyze the flow cytometry data. For custom code that has been deposited into a community repository, provide accession details.* |
| Cell population abundance | *Describe the abundance of the relevant cell populations within post-sort fractions, providing details on the purity of the samples and how it was determined.* |
| Gating strategy | For single-nucleus mRNA sequencing DAPI+NeuN-Olig2- except for lung brain metastasis samples where all DAPI+ nuclei were sorted. The sorting data were not used for any inferences. |

☐ Tick this box to confirm that a figure exemplifying the gating strategy is provided in the Supplementary Information.

