## [Peer Review File · Nature Immunology]

Novel transcriptomic microglia taxonomy across mouse and human pathologies

Corresponding Author: Professor Marco Prinz

Version 0:

Reviewer comments:

Reviewer #1

(Remarks to the Author)

The authors have made a substantial effort to address the reviewers' comments, resulting in a markedly improved and much more rigorous manuscript. In response to reviewer 1's major concerns, they added extensive new analyses, clarified statistical treatments, incorporated appropriate controls, expanded disease-stage comparisons, and significantly strengthened the cross-species analyses with proper statistical validation. In particular, the reanalysis of human–mouse correspondence, the expanded spatial transcriptomic validation, and the improved presentation of figures and supplementary data greatly enhance the clarity, robustness, and value of this atlas for the field. Overall, this is a strong revision, and I commend the authors for the depth of work invested in addressing the critiques.

One issue remains insufficiently addressed (consider minor). Although I agree that no existing Cre-based approaches for microglia subclusters, the concern raised in Question 7 regarding the genetic strategy for microglia-specific manipulation is not fully resolved. While the authors rely on Cx3cr1-CreER, this line is known to have limitations, including recombination in non-microglial myeloid populations (such as CAMs). To further strengthen the conclusions and exclude potential confounds, it would be important to validate key findings using an independent microglia-restricted Cre line, such as TMEM119-CreER, or an equivalent approach. Even limited validation with an alternative microglial Cre line would substantially increase confidence that the observed effects are truly microglia-intrinsic.

Reviewer #2

(Remarks to the Author)

I thank the authors for their extensive revisions and efforts to improve the manuscript. Great also that the data and code are now shared. I am satisfied with the revisions with one exception: I appreciate the new Fig 6 with mouse/human integration but I'm still not convinced that the integration has worked. The figure legend to Fig 6B claims that "extensive co-clustering between human (pink) and mouse (green) microglia" but the actual figure seems to show little overlap. Same is true in Fig 6C where every area that is dense in human cells is sparse in mouse and vice versa. It's also evident that the control clusters in Fig. 6E do not match particularly well across species. I suggest removing the claim of extensive overlap.

If the authors clearly state the limitations, challenges and shortcomings of the integration, I think the analysis can stand. In particular, make sure to point out clearly that cross-species comparison of overall gene expression profiles is fraught with difficulty and therefore the observed correlation patterns may reflect more global species differences rather than anything related to microglia biology.

Reviewer #3

(Remarks to the Author)

This manuscript proposes a microglia resource for the community. While the reviewer recognises the value of such initiative and approach that is correctly done, the study presented here is somehow limited in term of novelty and biological insights. Indeed, there is no real novel discoveries presented here and it will have been interesting to see what can be learned from such massive integration of OMICS data. In addition, to compensate from the lack of novelty, it will have been interesting to compare to other initiatives previously published as well as to provide a website/portal to access such resource.

Decision Letter:

Our ref: NI-RS41978-T

13th Jan 2026

Dear Dr. Prinz,

Thank you for submitting your revised manuscript "A spatial map of microglial states across human and mouse CNS pathologies" (NI-RS41978-T). It has now been seen by:

reviewer 3 - your original reviewer 3 at Nature but who had declined to re-review at Nature. They agreed to take another look for us

reviewer 4 - your original reviewer 4 at Nature

reviewer 5 - your original Nature reviewer 1 declined to re-review for us, so this is a totally new reviewer who was asked to mediate on your response to reviewer 1

The reviewers find that the paper has improved in revision, and therefore we'll be happy in principle to publish it in Nature Immunology, pending minor revisions to satisfy the referees' final requests and to comply with our editorial and formatting guidelines. We do not expect additional cell specific knockout mice, but the other revisions suggested by our reviewers are expected. But first...

We will now perform detailed checks on your paper and will send you a checklist detailing our editorial and formatting requirements in about a week. Please do not upload the final materials and make any revisions until you receive this additional information from us.

If you had not uploaded a Word file for the current version of the manuscript, we will need one before beginning the editing process; please email that to immunology@us.nature.com at your earliest convenience.

Thank you again for your interest in Nature Immunology Please do not hesitate to contact me if you have any questions.

Sincerely,

Nick Bernard, PhD
Senior Editor
Nature Immunology

Reviewer #3 (was your reviewer 3 at Nature in the 1st round of review)

This manuscript proposes a microglia resource for the community. While the reviewer recognises the value of such initiative and approach that is correctly done, the study presented here is somehow limited in term of novelty and biological insights. Indeed, there is no real novel discoveries presented here and it will have been interesting to see what can be learned from such massive integration of OMICS data. In addition, to compensate from the lack of novelty, it will have been interesting to compare to other initiatives previously published as well as to provide a website/portal to access such resource.

Reviewer #4 (was your reviewer 4 at Nature)

I thank the authors for their extensive revisions and efforts to improve the manuscript. Great also that the data and code are now shared. I am satisfied with the revisions with one exception: I appreciate the new Fig 6 with mouse/human integration but I'm still not convinced that the integration has worked. The figure legend to Fig 6B claims that "extensive co-clustering between human (pink) and mouse (green) microglia" but the actual figure seems to show little overlap. Same is true in Fig 6C where every area that is dense in human cells is sparse in mouse and vice versa. It's also evident that the control clusters in Fig. 6E do not match particularly well across species. I suggest removing the claim of extensive overlap.

If the authors clearly state the limitations, challenges and shortcomings of the integration, I think the analysis can stand. In particular, make sure to point out clearly that cross-species comparison of overall gene expression profiles is fraught with difficulty and therefore the observed correlation patterns may reflect more global species differences rather than anything related to microglia biology.

Reviewer #5 (new reviewer - mediating for missing reviewer 1 from Nature)

The authors have made a substantial effort to address the reviewers' comments, resulting in a markedly improved and much more rigorous manuscript. In response to reviewer 1's major concerns, they added extensive new analyses, clarified statistical treatments, incorporated appropriate controls, expanded disease-stage comparisons, and significantly strengthened the cross-species analyses with proper statistical validation. In particular, the reanalysis of human–mouse correspondence, the expanded spatial transcriptomic validation, and the improved presentation of figures and supplementary data greatly enhance the clarity, robustness, and value of this atlas for the field. Overall, this is a strong revision, and I commend the authors for the depth of work invested in addressing the critiques.

One issue remains insufficiently addressed (consider minor). Although I agree that no existing Cre-based approaches for microglia subclusters, the concern raised in Question 7 regarding the genetic strategy for microglia-specific manipulation is not fully resolved. While the authors rely on Cx3cr1-CreER, this line is known to have limitations, including recombination in non-microglial myeloid populations (such as CAMs). To further strengthen the conclusions and exclude potential confounds, it would be important to validate key findings using an independent microglia-restricted Cre line, such as TMEM119-CreER, or an equivalent approach. Even limited validation with an alternative microglial Cre line would substantially increase confidence that the observed effects are truly microglia-intrinsic.

Manuscript 2023-11-21001B-Z

"A spatial map of microglia states across human and mouse pathologies"

Referee #1 (Remarks to the Author):

The manuscript has been improved with new data collected regarding diseases and brain regions. However, while the study is rich in descriptive data, it still lacks more specific functional assays that would help better characterize the role of these clusters in disease progression and establishment. These additional experiments would further strengthen the manuscript and make it more suitable for publication in Nature.

We thank the reviewer for their very positive statement as well as thoughtful and constructive critique. His/her feedback has helped us substantially improve the clarity, rigor, and accessibility of the manuscript. Below, we provide the details for the improvements we did in response to their comments.

Major Comments:

1. The authors reference data presented in tables as though statistical analyses have been performed (e.g., Ext. Data Figs. 2A-D, Supp. Files 1 and 2 on line 187). However, to support the claims made in the text, this data should be presented in plots, and the statistical significance should be clearly indicated.

We thank the reviewer for highlighting the need for clearer presentation of the statistical analyses. In the revised manuscript, this point has been fully addressed now. **New Extended Data Fig. 2** contains both descriptive visualizations (Panels A, B, and F) and panels derived from statistical analyses (Panels C-E). For Panel D and E, the underlying source data and statistical results were already provided in **Supplementary files 1 and 3**.

In the revision, we have:

- incorporated enrichment statistics and p -values (hypergeometric test with Benjamini-Hochberg correction) directly into **new Extended Data Fig. 2C**. Significance marks are shown for representative comparisons only due to space constraints. The complete statistical results, however, are available now in the **new Supplementary table 3**.
- updated the figure legend for the **new Extended Data Fig. 2C** to explicitly reference the relevant supplementary files for each panel.
- Additionally, we have extensively revised the statistical analyses throughout the manuscript. Each bar plot, dot plot, heat map are now supported by statistical analysis. The descriptive statistics are provided in supplementary tables.
- With these changes, all statistically supported claims in the text are now accompanied by their corresponding graphical representation in the main or extended figures and are directly linked to complete, tabulated statistical results. We thank the reviewer for bringing forward this important concern.

2. In Figure 1F, the authors should include results from control and non-disease groups to confirm that the observed clusters specifically emerge during disease.

We thank the reviewer for this helpful suggestion. This point has been fully addressed in the revision. Non-neurological control samples for MS have now been incorporated into the **new Figure 1F**, and the AGS panel has been moved to the **new Extended Data Fig. 4G**, where disease and matching controls are shown side-by-side. These additions demonstrate that the disease-associated microglial clusters observed in MS (microglia cluster 33) and AGS (microglia cluster 129) are virtually absent in non-diseased brains. The revised figures therefore confirm that these clusters emerge specifically in the disease context.

3. While I understand the challenges involved in collecting samples from different brain regions within the same disease context, the authors should clarify in the text that any observed differences may be influenced by the brain regions themselves.

We thank the reviewer for highlighting the complexity introduced by the interplay of disease context and anatomical region. We agree that certain patterns in Figures 2 and 3 could initially appear difficult to separate into purely disease-associated versus region-specific effects.

To address this, our analysis explicitly accounts for both variables:

- **Regional stratification.** In **Figure 2F**, microglia supercluster proportions are presented separately for each anatomical region within each disease context, with statistical testing performed independently for each region (one-sided hypergeometric test with Benjamini-Hochberg (BH) correction). This allows us to detect enrichments that are region-specific *versus* those that appear consistently across multiple regions.
- **Contextual comparisons.** For each pathology, matched control samples from the same region were included, enabling direct assessment of disease-associated shifts independent of anatomical distribution.
- **Spatial mapping.** **Figures 2D** and **3A-C** illustrate that some supercluster changes are highly localized to pathological foci within a region (e.g. MS rim vs. NAWM), while others appear in multiple regions (e.g. interferon-signature superclusters in AGS and MS lesions).
- **Supporting data table.** The underlying statistics for **Figure 2F** are provided in the **new Supplementary File 10** ("Statistics for human microglia superclusters across diverse regions"), which lists, for every [*disease x region*] combination: the proportion of each microglia supercluster, the corresponding proportion in matched controls, fold enrichment, one-sided hypergeometric test *p*-values, and Benjamini-Hochberg (BH)-adjusted *p*-values. This enables direct evaluation of whether enrichments are region-specific or shared across contexts.

While disease–region interactions are inherently intertwined in the human CNS, the dataset structure and statistical framework allow now that these effects to be distinguished where the sample distribution permits.

4. More consideration should be given to the disease stage for each sample when comparing diseases. It might be more comprehensive to focus on comparisons between the diseases and their respective controls.

We appreciate the reviewer's emphasis on the importance of disease stage in comparative analyses. We agree that disease stage is a critical factor that influences microglial activation. Accordingly, our dataset includes detailed annotation of sample disease stage, lesion type, and CNS region to contextualize microglial states. However, while pairwise comparisons between each disease and its matched controls are useful for identifying disease-associated changes, they are inherently limited when the goal is to construct a comprehensive and comparative atlas of microglial states across diseases and contexts.

In order to address the reviewer's suggestion, we compared different disease stages in humans from AD and MS with the respective controls. We show now that microglial diversity is increased with disease progression. In humans, AD samples showed distinct early *versus* late stage shifts relative to homeostatic occipital cortex and hippocampal controls, while MS samples exhibited progressive alterations across NAWM, lesion rim, active lesions, and inactive cores compared to control cerebral cortex (**new Ext. Data Fig. 2F**). We have described these results in manuscript (page 7, lines 229-233). Similarly in mouse, stage-specific analyses also revealed context-dependent remodeling, with early to late transitions evident across contexts (**new Ext. Data Fig. 7E**). We have described these results in manuscript (page 14-15, lines 428-432).

Therefore, while we do include disease-specific comparisons (e.g. Mg33 in MS vs control; Mg129 in AGS vs. control) to validate disease association (**Fig. 1F, Ext. Data Fig. 4G**) and clearly show microglia profiles in different disease stages in human in AD and MS, as well as in diverse mouse models of CNS diseases such as FAD, VSV, MCAO, Cuprizone, EAE and FNX, our primary analytical focus remains on an integrated, multi-disease perspective. This is essential to reveal the breadth and convergence of myeloid cell transcriptional programs across CNS pathologies, which would not be possible when focusing in individual diseases only. These additions ensure that both **disease-specific identity and stage-dependent progression** are appropriately accounted for in our analyses, while preserving the global comparative framework necessary to uncover the full spectrum of microglial states across CNS disorders.

5. The authors mention a 'robust conservation' of microglia clusters between mouse EAE and human MS, but the correlation is relatively low, with the highest correlation being 0.62. This is the case for the phagocytosis, antigen presentation, and interferon signature superclusters. The correlation might be meaningful, but the authors should report the p-values to ensure that these correlations are not due to chance.

We thank the reviewer for these very valid and important points. We agree that the term "robust conservation" requires clarification. For having best possible expertise on the species comparison for human *versus* mouse microglia we now teamed up with Henrik Kaessmann from the ZMBH Heidelberg, a world leader in the field of species gene comparison. As a consequence of this joint team effort, we have completely reanalyzed the cross-species dataset and provide now a completely **new Figure 6** with an updated RPCA-based integration and comprehensive correlation analyses including:

- **Complete correlation matrices.** We now present the full cross-species correlation landscape at both supercluster (**new Fig. 6D**) and cluster levels (**new Fig. 6E**), reporting Pearson's r values and corresponding two-sided p -values for all comparisons. Significant correlations ($p < 0.01$) are marked in the figures, with full statistics provided in the **new Supplementary Tables 27–29**.
- **Statistical validation.** To ensure correlations are not attributable to chance, we performed 1000 bootstrap comparisons with randomly sampled gene sets (80 % of genes). The empirical p -value heat map from this analysis is shown in the **new Extended Data Fig. 10**, with full statistics provided in the **new Supplementary Table 29** confirming the robustness of our findings.
- **Visualization and readability.** To make the large-scale matrices better interpretable, we overlaid bar plots showing disease/model composition within each cluster, enabling immediate assessment of both similarity and pathological context. Detailed Sankey plots for key superclusters are retained in the **new Extended Data Fig. 11**.

Together, these updates highlight conserved functional programs across mouse models and human disease, while also making explicit the substantial species-specific differences. This more

nuanced framing improves methodological transparency, strengthens statistical robustness, and aligns our interpretation more closely with the data shown.

6. After providing a detailed analysis of various microglia clusters, it is surprising that the authors chose to perform a global IFN receptor deletion in all microglia populations. It would be more informative to include experiments that examine the specific function of individual clusters rather than depleting a cytokine receptor that may affect all the clusters. In addition, the function of IFN receptor depletion in microglia during EAE has already been explored.

We thank the reviewer for this important point. While we agree that targeting individual microglial clusters would be an informative future direction, such experiments are currently not technically feasible. Our rationale and clarifications are as follows.

- **Technical feasibility.** No existing Cre-based or other targeted approaches permit reliable IFN receptor ablation in a single microglial subcluster.
- **Cluster-specific readouts from global deletion.** Although our deletion strategy was pan-Cx₃Cr1⁺ microglial, the effects were not uniform across all clusters. Both type I IFN receptor (IFNAR) and type II IFN γ receptor (IFN γ R) loss produced selective shifts, within only distinct microglial clusters showing quantitative or transcriptional changes during EAE. These are specifically highlighted in Figure 7F, which quantifies the UMAPs in Figure 7E. Other clusters remained unchanged.
- **Novelty relative to prior studies.** Earlier EAE work on IFN receptor deletion relied on bulk populations or did not use CNS myeloid cell specific knockouts, thereby also targeting blood-derived monocytes during disease induction. In fact, a former study used LysM-Cre (myeloid-wide) but not microglia-restricted targeting and found worsened EAE when IFNAR was deleted on all myeloid cells¹. True microglial receptor genetics in EAE have only been shown for IFN γ R, where microglia-restricted deletion increased disease severity by impairing Treg interactions². Notably, this work was performed in a mixed B6 x SJL F1 background (remitting relapsing (RR)-EAE model), which strongly differs from the standard B6 MOG₃₅₋₅₅ EAE paradigm used in our study and precludes direct comparison. Other reports focused on systemic IFN- γ administration followed by bulk RNA-seq³, which by definition cannot resolve the underlying microglial heterogeneity. Thus, the notion that “the function of IFN receptor depletion in microglia during EAE has already been explored” is, at least as we found it, not supported by the literature. In contrast, our microglia-focused knockout combined with single-cell profiling precisely identifies for the first time the microglial states engaged in IFN γ signaling, revealing heterogeneity inaccessible to bulk assays.

In light of these points, we truly believe that a global microglial IFN γ R knockout, coupled with single-cell resolution analysis, represents the most rigorous and interpretable strategy currently available to dissect IFN γ -dependent pathways in a microglia-focused context.

References:

1. Prinz, M. *et al.* Distinct and nonredundant in vivo functions of IFNAR on myeloid cells limit autoimmunity in the central nervous system. *Immunity* 28, 675-686, doi:10.1016/j.immuni.2008.03.011 (2008).
2. Haimon, Z. *et al.* Cognate microglia-T cell interactions shape the functional regulatory T cell pool in experimental autoimmune encephalomyelitis pathology. *Nat Immunol* 23, 1749-1762, doi:10.1038/s41590-022-01360-6 (2022).
3. Tichauer, J. E. *et al.* Interferon-gamma ameliorates experimental autoimmune encephalomyelitis by inducing homeostatic adaptation of microglia. *Front Immunol* 14, 1191838, doi:10.3389/fimmu.2023.1191838 (2023).

Referee #2

I am impressed with the extensive reanalysis and restructuring of the manuscript and how the authors addressed my previous concerns. I think the paper will be a great resource for the field and I have no further questions and support publication.

We thank the reviewer for her/his enthusiastic feedback!

Referee #3 (Remarks to the Author):

Authors attempted to map microglial heterogeneity across brain pathologies in man and mouse, combining multiple omic/dimension analysis. While a commendable exercise in principle with the provision of relevant human single cell datasets to the community, the enthusiasm of the reviewer is limited by the descriptive nature of the study which is inherent to such integration approach (Figure 1 and 2, 5, 6) and the limited biological novelty (with limited functional validation of the results presented) that support Nature publication level (Figure 3 and 7).

We sincerely thank the reviewer for her/his thoughtful summary for highlighting the key findings, broader implications and overall significance of the study. We especially appreciate her/his emphasis on the value of the human data.

In order to improve the biological relevance of our study we substantially expanded the number of CNS conditions in both species (now 32 conditions in total derived from 14 in humans and 18 conditions in mice) and considerably increased the number of sequenced CNS cells to 1.046,000 cells including 271,000 microglia cells.

This largely extended profiling approach further motivated us to aim for a completely new microglia taxonomy (as thankfully suggested by reviewer #2). Indeed, high resolution Paris clustering allowed us to establish a novel microglia classification ultimately leading the description of several superclusters of microglia, CNS-associated macrophages (CAMs) and monocyte-derived macrophages (MDMs). This innovative data-driven clustering of CNS phagocytes with resulting dendrograms permitted to spotlight their distinct functional aspects across several clinical conditions. The next level of new taxonomy tree consisted of the individual clusters. This new myeloid cataloguing is now displayed in the **new Fig. 1** (for human samples) and **new Fig. 5** (for mouse samples), respectively.

In addition, we significantly expanded our subcellular spatial analyses of the identified myeloid cell clusters and performed now both targeted (1000-plex gene panel by CosMx™) as well untargeted genome-wide spatial transcriptomics (Visium HD) to deeply characterize microglia cluster distributions and their individual interactomes (**new Figures 2-4**). This kind of high dimensional spatial analyses has not been performed on microglia *in situ* before.

Taken together, these completely new experiments allow ground breaking and fresh insights into the biology of CNS myeloid cells in both human and mouse at an unprecedented depth.

In addition, some of the data presented do not show clear statistical significance and lack power (Figure 4, D-G for ex). About Figure 7, the importance of CSF1R and IFNGR for microglial maintenance and activation has already been addressed by several studies.

We thank this reviewer for pointing this out. As described above we considerably increased the number of sequenced CNS cells to > 1.000,000 cells including several hundred thousands of myeloid cells. We truly believe that due this increase samples size our study is not underpowered anymore. In fact, new samples were added for FKBP5 protein analysis in microglia with inclusion of three new CNS regions e.g. motor cortex, hippocampus, and spinal cord. We also added more samples to increase the number of patients in each disease with at least 3 samples in each disease. Control spinal cord microglia showed high expression of FKBP5 which has been previously shown (Yadav et al, Neuron 2023). In MS, we could separately analyse four cortex and two spinal cord samples. These new data are now displayed in the **new Ext. Data Fig.4 B,C**.

We respectfully disagree with the reviewer's comment about Fig. 7. regarding the importance of CSFR1, IFNAR, IFNGR signalling for microglia states. First, our newly introduced microglia taxonomy with accompanying high dimensional subcellular spatial transcriptomics provided a unique basis to classify microglia/myeloid cells across perturbations that was not available before. Second, to the best of our knowledge no scRNA-sequencing data on *Cx3CR1^{CreERT2}Ifngr1^{fl/fl}* or *Cx3CR1^{CreERT2}Ifnar1^{fl/fl}* mice during EAE or LPS challenge in C57BL/6 mouse model are available yet (**new Fig 7E-J, new Extended Data Fig 12D,E**). Therefore, the impact of cell-autonomous CSF1R- and type I/II IFN signalling on distinct microglial states *in vivo* during precise disease conditions reflects a completely new set of data in the field.

Major comments:

The article focuses on microglia, and the use of single-cell nuclear transcriptome sequencing results in a substantial loss of microglia transcriptome information, which is and will be detrimental to subsequent analyses describing subpopulations of microglia. The authors need to address this potential major issue.

We thank the reviewer for raising this key point of our study. To unequivocally define the breadth of microglial and other CNS myeloid cell transcriptional states and compare them across several autoimmune, neoplastic, degenerative, dysplastic and inflammatory disorders, we were using a) consistent isolation protocols, b) identical pipelines for experimental measurements and downstream analysis and, when applicable, c) mice of the corresponding models with identical genetic background and sex. This enables the best comparison of myeloid states within diverse pathological conditions as well as between different species such as mouse and human.

We were completely aware of the fact that usage of ncRNA-seq compared to scRNA-seq will lead to some loss of cytoplasmatic/mitochondrial genes but for assessments through pathologies no other approaches are technically and logistically possible. Importantly, in our ncRNA-seq analyses we were able to detect many known "DAM" genes such as *Trem2*, *Apoe*, *Ctsb*, *Axl*, *Cst7*, *Spp1*, *Clec7a* and *Lpl* that have been described before using scRNA-seq approaches (Keren-Shaul et al. Cell 2017; Krasemann et al. Immunity 2017; Butovsky & Weiner, Nat Rev Neurosci 2018; Thrupp et al. Cell Rep 2020) (**new Ext. Data Fig. 9**).

We therefore truly believe that our extensive, profiling-based myeloid cell classification is biologically highly relevant and of importance for the field.

In the attempt to microglia heterogeneity, authors should have tried to acknowledge previous published work with correlation to previously reported populations. It would be informative to correlate signatures of the various populations associated with the microglia populations discovered in this work.

The reviewer provides an excellent point that we have carefully considered. We agree that previously reported microglia states have to be taken into account. Therefore, whenever possible,

we compared already published gene signatures, e.g. for “DAM” (*ApoE*, *Clec7a*, *Trem2*, *Tyrobp*, *Itgax*, *Spp1*, *Gpnmb*, *Lgals3* etc.) with our own established microglia superclusters and clusters. We will furthermore provide an openly available online platform for a full exploration of our snRNA-seq and spatial transcriptomics data upon acceptance of our manuscript.

We would like to further stress the point that the main achievement of our study is a novel myeloid cell taxonomy based on transcription-based hierarchical clustering. Furthermore, microglia clusters of this taxonomical tree were additionally characterized spatially resulting in a novel multidimensional map charting their distinct neighbourhood relationships during perturbations.

Origin of microglia: Without using a validated Fate-mapping system, how did the authors differentiate between Microglia and MDMs in results such as Figure2D, Figure4C, Figure5D, etc. It is difficult to differentiate between MGs and MDMs by relying only on markers such as IBA1, TMEM119.

Data shown previously shown in Fig. 4C were of human origin where no adequate fate mapping is possible. In contrast, data shown before in Fig. 5D (now depicted in **Fig. 5E**) were indeed generated from mouse experiments. *Tmem119* (that encodes for the protein TMEM119) is known as a prototypical microglia core gene as shown by us (Masuda et al. Nat Immunol 2020, Faust et al. Cell Rep 2023) and many others before. The presence of this gene/protein during disease conditions ensures their microglial cell identity. This view is widely accepted in the field (Paolicelli et al. Neuron 2022).

Figure2A, 5A: There are many cell subpopulations unique to diseases in these analyses. One will expect more common subpopulations, like "resting" microglia, rather than one disease microglia only containing unique subpopulations. This could be due to a problem with the integration method/approach? The authors wrote "we integrated the combined dataset for sex with Harmony using "RunHarmony" function(group.by.vars = "sex)". However, differences in diseases and samples might be more important than gender differences? This is important as if there is some data integration issues, the subsequent analysis and the entire article lose relevance. In this regard, the authors should clarify the rationale behind their decision to address batch effects between sexes but not among different samples. Furthermore, it is important to clearly indicate the sex information of the donors rather than relying on a "Male score"?

This is a very constructive suggestion that has informed our revisions. To directly address the point of microglia heterogeneity in patient specimens and to allow a better comparison of different cell states we systematically surveyed myeloid cells across several perturbations that allowed a novel myeloid cell taxonomy based on transcription-based hierarchical clustering (**new Figs 1 and 5, new Ext. Data Fig. 2,7, and 9**). This approach led to the distinction of eight different microglia superclusters that can be found to a variable degree in many samples. These data suggest that several myeloid states can exist concomitantly. Microglial clusters of this taxonomical tree were furthermore comprehensively spatially characterized resulting in a multidimensional map charting their distinct neighbourhood relationships during perturbations.

This reviewer is correct in stating that “*differences in diseases and samples might be more important than gender differences*”. To minimize the batch effects in gene expression that arise due to sex which could hamper the true biological signal from disease, we applied harmony to integrate the data for sex. Individual samples within our datasets represent diverse biological conditions and diseases. We only harmonized data by sex (not by sample) to avoid over-correction of data and keep genuine disease associated gene expression of microglia intact. Several studies highlighting such over-correction and loss of biological signal due to integration algorithms and

wrong integration approaches are already published. These studies compared several integration methods and even their impact on biological heterogeneity of data was also evaluated (Luecken et al. Nat Methods, 2022; Andreatta et al. Nat Commun, 2024; Eraslan et al. Science, 2022). We carefully considered these important studies before deciding the integration algorithm as well as the best approach for integration. We have also cited these studies in our updated methods section and provided better explanation for integration as well as other technical aspects of study. The "Male score" was the name of the metadata column which was used to fetch the sex information. We realise that this was confusing so we have updated the methods accordingly. In sum, we are confident that our used data analyses pipelines represent adequate and *state-of-the-art* bioinformatics tools that are best designed to detect biologically relevant processes in myeloid cells. We were able to gained already extensive experiences in these bioinformatics tools for single-cell approaches over the last years (Masuda et al. Nature, 2019; Sankowski et al. Nat Neurosci, 2019; Sankowski et al. Nat Med, 2024; Amann et al. Sci Immunol, 2024).

To justify the utilization of 70 principal components (PCs) for unsupervised clustering, the authors need to demonstrate the necessity of such a high number of PCs for their analysis, ensuring that it does not introduce unnecessary complexity into the clusters.

This is highly relevant concern, and we thank the reviewer for raising it. In the previous version of our manuscript, we analysed all the cell types together. To separate all the sequenced cell types (neurons, glia, immune cells, tumor cells) in the combined dataset we had utilized 70 PCs. However, in the current version of analysis, to eliminate the need of using 70 PCs and also to detect rare clusters of immune cells, we changed our strategy and analysed immune cells separately after initial identification of broad cell types based on low number of principal components (PC). We then applied machine learning and graph-based Paris clustering which represents state-of-the-art methodology for understanding the transcriptomic heterogeneity in atlas level datasets comprised of heterogeneous cell types as described before (Yao et al. Nature, 2023; Siletti et al. Science, 2023).

Given the substantial size of the dataset, it is crucial to assess the statistical robustness of the selected clusters when using Harmony as the model. The authors should provide evidence of the clusters' significance and statistically evaluate whether there is inherent heterogeneity within these clusters that remains unaccounted for or if there is an excessive clustering that lacks biological relevance.

We thank the reviewer for this thoughtful point and agree that statistical robustness is essential in our data analysis. In order to address these points, we completely changed our clustering approach. We now applied novel high resolution Paris clustering which assures maximum separation between clusters but still does not overcluster the cells. We have described the technical details of Paris clustering along with the proper citations which inspired our approach. Paris clustering has been recently utilized to provide atlas level datasets describing heterogeneity of cell types and states in the mouse and human brain under homeostatic conditions before (Yao et al. Nature, 2023; Siletti et al. Science, 2023).

There is an absence of a systematic comparison between microglia clusters in humans and mice. While the analysis of differentially expressed genes offers some insights, it falls short of providing a comprehensive understanding. Given the extensive identification of microglia clusters by the authors, it is essential to assess the similarity between these clusters in both human and mouse datasets.

We thank the reviewer for this insightful suggestion, and for the opportunity to address this point to achieve the level of practical application of our data for scientists which they envision. In that regard, we agree that a global, unbiased cluster-to-cluster comparison across species provides a more comprehensive view of microglial state conservation and model fidelity. For having best possible expertise on the species comparison for human *versus* mouse microglia we now teamed up with Henrik Kaessmann from the ZMBH in Heidelberg, a world leader in the field of species gene comparison.

In the revised manuscript, we have therefore fully reanalyzed the cross-species dataset and provide now a completely **new Figure 6** as follows:

- **Global correlation analysis.** We now present a complete cross-species correlation matrix at both the supercluster (**new Fig. 6D**) and cluster level (**new Fig. 6E**), based on Pearson correlations of mean expression profiles across all detected genes. This analysis captures the full similarity landscape without restriction to any subsets.
- **Statistical robustness.** For each correlation, we report the Pearson's r value and corresponding two-sided p -value, with significant correlations ($p < 0.01$) marked in the figures. Full statistics are provided in the **new Supplementary Tables 27–29**. To further validate robustness, we performed 1000 bootstrap comparisons using randomly sampled gene sets (80% of genes). The empirical p -value heat map from this analysis is now shown in the **new Extended Data Fig. 10**, with full statistics provided in the **new Supplementary Table 29** confirming the robustness of our findings.
- **Visualization and readability.** To aid interpretation of the large-scale correlation matrix, we overlaid bar plots showing the disease/model composition of each cluster. This enables readers to assess both the degree of cross-species similarity and the pathological context at a glance. While a global Sankey diagram would be impractical to compare hundreds of microglia clusters between human and mouse, we provide detailed Sankey plots for key microglia superclusters in the **new Extended Data Fig. 11**, allowing readers to explore these specific mappings.

Together, these updates improve methodological transparency, broaden the scope of the cross-species comparison, and provide readers with both a comprehensive overview and detailed supporting analyses to assess model fidelity and divergence.

On spatial data, authors have found 11 out of 20 microglia populations in the spatial omics data, but they don't show any clear evidence besides a visual plot where some cells only have one gene as a defining marker. A heatmap would help to convince the reader that the translation of the single-cell annotation can be mapped on the spatial data.

We fully agree that the initial manuscript felt short in providing sufficient evidence supporting the existence of the identified microglia clusters *in situ*. In the frame of the revision, we have therefore put special emphasize on the spatial characterization of the identified Mg clusters across pathologies and the description of their locally interacting cells. We first used high-resolution confocal microscopy combined with subcellular spatial transcriptomics using a predefined, myeloid cell-focussed 1000-plex gene panel (Nanostring CosMx) in human samples of several pathologies (AD, AGS, FCD, active MS and GBM) (**new Fig. 2 and 3**). This allowed us to determine the spatial localizations of the transcriptionally identified microglial clusters and their associated superclusters.

We next utilized a probe-based whole transcriptome level subcellular resolution spatial transcriptomics workflow (Visium HD) (**new Fig.4, new Ext. Data Fig. 6**). This unbiased approach was chosen to visualize the variety of microglial superclusters and clusters even better. Indeed, more microglia clusters were spatially detectable in diseased brains by this additional spatial transcriptomic approach. In summary, our extensive novel whole genome subcellular spatial

transcriptomics allowed unbiased spatial mapping uncovering the wealth of disease-associated microglia clusters that belong to functional superclusters.

The statement on cluster association of the different microglia subsets in fig 3E appears not exact. The authors claimed that Mg3 did not have any preferential neighbouring cell types in controls and AD and that they were preferentially located near oligodendrocytes in FCD, AGS, and MS. Still, the strength of neighbouring with Mg3 and SMC is as strong as Mg3 in FCD. Also, Mg3 is not present in AGS.

As described above the spatial transcriptomics analyses were considerably expanded and the resulting images extensively changed during the revision phase. As a consequence, the old Fig. 3 is not existing anymore and was replaced by the **new Figs. 2-4**.

The authors state in the text that certain populations of microglia are present in the different disease types they performed spatial transcriptomics on. Still, when we observe Figure 3E, there are most types of microglia present in every disease type. A barplot with the proportions should convince the reader of the statements claimed in the text.

Again, this is a very valid and important point raised by the reviewer. We now added quantifications of the spatially identified microglia clusters and their linked superclusters across brain regions and disease modalities in the respective figures (**new Figs. 2E,F, new Figs. 4C-E, new Ext. Data Fig. 6F**).

Using numbers to refer to MG subgroups greatly reduces the readability of the article.

In order to improve the readability of the manuscript we have extensively revised the text to increase the readability. Nevertheless, we could not completely omit using numbers to refer to microglia clusters.

Referee #4 (Remarks to the Author):

Chhatbar et al. present an atlas of myeloid cells in CNS across disease contexts, focusing on microglia. Overall, this is a valuable dataset, but I would have expected more powerful insights related to diseases. For example, could disease contexts be related via shared microglia states? it looks like tumor contexts were not shared with neurodegenerative disease contexts, and within tumors, GBM differed from astrocytoma/oligodendroglioma. Could such patterns of shared microglia states be explained by modular gene expression, e.g. GBM-associated microglia expressing modules A and B, while astrocytoma and oligodendroglioma expressed B and C? Such patterns would have greater explanatory power and could lead to insights into disease biology.

We thank the reviewer for her/his very positive statement as well as thoughtful and constructive critique. This feedback has helped us to substantially improve the clarity, rigor, and accessibility of the manuscript. Below, we provide the details for the steps we took in response.

Major comments:

1. Fig 6: integration is fraught with many pitfalls. I would like to see a simpler analysis, i.e. a correlation matrix between the clusters of each species, or equivalently a sankey plot. The authors do show correlations and sankey plots, but only within superclusters and within corresponding disease/models. E.g. when the authors say that “Comparison between EAE and MS revealed human clusters Mg 33 and 36 having the highest similarity to mouse clusters Mg 40, 41”, do they mean globally, or just within the “antigen presentation” supercluster? I assume the latter but that kind of analysis doesn’t show whether e.g. the clusters dominated by GL261 (a glioma model) in mouse are “the same” cell types as the clusters dominated by GBM in human. In other words, I would like to see if mouse cluster 12 is the best match to human cluster 61, and so on for each disease/model. That kind of unbiased analysis could show which mouse models are accurate (in terms of myeloid cell states) and which ones are not.

We thank the reviewer for this insightful suggestion, and for the opportunity to address this point to achieve the level of practical application of our data for scientists which she/he envisions. In that regard, we agree that a global, unbiased cluster-to-cluster comparison across species provides a more comprehensive view of microglial state conservation and model fidelity. For having best possible expertise on the species comparison for human *versus* mouse microglia we now teamed up with Henrik Kaessmann from the ZMBH in Heidelberg, a world leader in the field of species gene comparison.

In the revised manuscript, we have therefore fully reanalyzed the cross-species dataset and provide now a completely **new Figure 6** as follows:

- **Global correlation analysis.** We now present a complete cross-species correlation matrix at both the supercluster (**new Fig. 6D**) and cluster level (**new Fig. 6E**), based on Pearson correlations of mean expression profiles across all detected genes. This analysis captures the full similarity landscape without restriction to any subsets.
- **Statistical robustness.** For each correlation, we report the Pearson’s r value and corresponding two-sided p -value, with significant correlations ($p < 0.01$) marked in the figures. Full statistics are provided in the **new Supplementary Tables 27–29**. To further validate robustness, we performed 1000 bootstrap comparisons using randomly sampled gene sets (80% of genes). The empirical p -value heat map from this analysis is now shown in the **new Extended Data Fig. 10**, with full statistics provided in the **new Supplementary Table 29** confirming the robustness of our findings.
- **Visualization and readability.** To aid interpretation of the large-scale correlation matrix, we overlaid bar plots showing the disease/model composition of each cluster. This enables readers to assess both the degree of cross-species similarity and the pathological context at a glance. While a global Sankey diagram would be impractical at this scale, we retained detailed Sankey plots for key microglia superclusters in the **new Extended Data Fig. 11**, allowing readers to explore these specific mappings.

Together, these updates improve methodological transparency, broaden the scope of the cross-species comparison, and provide readers with both a comprehensive overview and detailed supporting analyses to assess model fidelity and divergence.

2. The CSF1R experiment (Fig 7) should be described better. First of all, it’s well known that CSF1R knockout mice nearly completely lack microglia, and that treatment of adult mice with potent and selective CSF1R inhibitor (PLX3397) leads to rapid loss of microglia within days, with about 99% loss in three weeks. The authors use an unnamed CSF1R inhibitor and cites ref. 24 in support of the claim that it can “efficiently block CSF1R downstream signalling *in vivo* in microglia without inducing cell depletion”. But ref. 24, as far as I can tell, did not examine CSF1R downstream signalling. Perhaps

https://doi.org/10.1038/s41419-020-03084-7 is a better citation? It also names the inhibitor sCSF1Rinh and gives the structure and IC50 values. The authors should this name unless there is a more official one. The way I read that paper, though, it doesn't claim that sCSF1Rinh efficiently blocks downstream signalling without inducing cell depletion. Rather, the claim is that depending on dose, it either causes cell death (high dose) or modulates microglia homeostasis (low dose). Presumably other CSF1R inhibitors behave similarly. Hence, a better way to rationalize the experiment in Fig 6 might be along the lines of "At high doses, CSF1R inhibitors are known to rapidly deplete microglia, whereas at lower doses, they modulate microglia homeostasis. Based on the latter observations, CSF1R inhibitors are being clinically developed for the treatment of..." etc etc.

We thank the reviewer for this important clarification. We now cite Hagan *et al.* (2020, Cell Death & Disease, DOI:10.1038/s41419-020-03084-7), which characterizes sCSF1Rinh, including its dose-dependent effects. Indeed, high-dose CSF1R inhibition is well established to cause near-complete microglial depletion, while lower exposures modulate proliferation and activation without global loss. In our study, we applied a low-dose regimen of sCSF1Rinh, and consistent with this, we observed preserved microglial numbers and morphology with almost no transcriptomic changes (only *ApoE* mRNA was changed). Exactly the same low-dose approach of sCSF1Rinh was already successfully used before for the inhibition of microglial downstream signalling in a different context in a Nature Neuroscience study by the Kerschensteiner group⁴. This experimental design allowed us to probe the functional consequences of partial CSF1R blockade *in vivo*, revealing no overt phenotype in EAE but exaggerated microglial reaction after LPS challenge, consistent with a shift in the CSF1R/CSF2R signaling balance.

Reference:

4. Locatelli, G. *et al.* Mononuclear phagocytes locally specify and adapt their phenotype in a multiple sclerosis model. *Nat Neurosci* 21, 1196-1208, doi:10.1038/s41593-018-0212-3 (2018).

3. The use of hierarchical clustering can conceal an underlying, simpler modular structure of the microglial phenotypic landscape. An analysis that would reveal modules, and show how they compose to yield the great variety of cell types reported here, would go a long way towards explaining and interpreting the findings, including interspecies differences. For example, maybe there are five independently activated modules, and they result in 2⁵ microglial cell states? A good approach is topic modelling, but there are many others. I wanted to check this myself, but since the data was not shared, nor a browser, I couldn't.

We thank the reviewer for raising this important conceptual point. We fully agree that microglial heterogeneity can also be viewed through the lens of modular transcriptional programs, and that methods such as topic modeling, Multi-Omics Factor Analysis (MOFA), or Non-negative Matrix Factorization (NMF) are valuable for capturing broad latent factors. In practice, however, these approaches typically assign hundreds of genes per factor, which often collapses disease-specific variation into generic Disease associated microglia (DAM) / Interferon-Responsive Microglia (IRM) /Antigen presentation (AP) programs. Our hierarchical framework was therefore designed to preserve both scales of organization: broad superclusters that capture overarching modules, while the Paris clusters break these modules into highly specific subprograms defined by a smaller set of genes, thereby enhancing resolution. This hierarchical clustering approach was successfully used by other labs recently to analyze the phenotypic landscapes of neurons in the human and mouse CNS^{5,6}.

In the revised discussion, we now explicitly acknowledge the reviewer's suggestion (page 22, lines 616-623) and explain how our approach integrates modular and fine-grained perspectives:

“While latent factor approaches such as topic modeling or Multi-Omics Factor Analysis (MOFA) are well suited to recover broad transcriptional modules, these typically involve hundreds of genes per factor and often collapse disease-specific variation into generic Disease associated microglia (DAM)/Interferon-Responsive Microglia (IRM)/Antigen presentation (AP) programs. By contrast, our hierarchical strategy defines superclusters as broad modules and Paris clusters as highly specific subprograms, thereby integrating modular and fine-grained perspectives of microglial heterogeneity.”

We believe this addition makes clear that our analysis does not preclude modular structures, but rather incorporates them at the level of superclusters while retaining disease-relevant specificity through finer cluster resolution.

References:

5. Yao, Z. *et al.* A high-resolution transcriptomic and spatial atlas of cell types in the whole mouse brain. *Nature* 624, 317-332, doi:10.1038/s41586-023-06812-z (2023).
6. Siletti, K. *et al.* Transcriptomic diversity of cell types across the adult human brain. *Science* 382, eadd7046, doi:10.1126/science.add7046 (2023).

4. Computer code for this project can be found upon request. is not acceptable. The code should be made available e.g. on github. The raw data for this project is available under GSE____.mis not acceptable. The raw data should be made available. For this kind of atlas project, there should also be an interactive browser, e.g. by uploading the data to cellxgene.

We thank the reviewer for highlighting the importance of data and code sharing. In the revised manuscript we have ensured that:

- **Codes.** All analysis codes to reproduce the figures is now deposited on GitHub (private repository; public release upon acceptance).
- **Human raw sequencing data** are deposited at the European Genome–Phenome Archive (EGA, accession: EGAS50000001289).
- **Mouse raw sequencing data** are deposited at NCBI GEO (accession: GSE304010).
- Processed final Seurat objects for all datasets and the code to reproduce the figures are available at zenodo (**10.5281/zenodo.16938034**).
- **Interactive browser** Unfortunately CZI does not allow new data uploads for cellxgene any more. We are working on providing a new data visualization platform via single cell portal from Broad institute.

These resources ensure full reproducibility and facilitate exploration of the atlas by the research community.

Minor Comment 1. Fig1A begins with all cells but then talks only about microglia. Numbers of cells is given only for microglia, but how many were CAMs or monocytes/MDMs?

We thank the reviewer for highlighting this point. Counts for all immune cell populations (including CAMs and monocytes/MDMs) were already provided in **Supplementary File 2**. To aid readability, we now also explicitly state the numbers for CAMs and MDMs in the **new Figure 1A**.

Minor Comment 2. Ext Fig 2D: I realize DCs are not the focus of the study, but probably CLEC9 would be a better marker for cDC1s.

We thank the reviewer for this suggestion. However, we did not detect high *CLEC9A* mRNA in our cDC1 populations. We have therefore chosen alternative but still reliable cDC1 markers such as *C1orf54*, *IDO1* and *ENPP1* (**Rebuttal figure 1, Supp. Table 2**). For cDC2 we plot expression of *IL1R2*, *AREG*, *RHEX* and for pDC we plot *BCL11A* and *SPIB*. We believe that one of the possible reason for low expression of *CLEC9A* could be the very low number of DCs present in our dataset compared to microglia, CNS associated macrophages (CAMs) and monocyte-derived macrophages (MDMs).

Rebuttal figure 1: Dot plot depicting expression of various myeloid and dendritic cell markers in myeloid and DCs in human data.

Minor Comment 3. It’s a matter of style of course, but I think the text would benefit from using fewer abbreviations. For example, the sentence “We named microglia SC based on gene expression...” is confusing (not only because of the missing plural “s”) and forces the reader to track back in the text to find the term “SC” defined, whereas if you spell it out “We named microglia superclusters based on gene expression...” is unambiguous and can be understood without referencing other parts of the text.

We thank the reviewer for the suggestion and spell out the term supercluster.

Minor Comment 4. *The use of singular or plural is sometimes confusing: “The myeloid cell SC was further deconvoluted...” for example, I take to mean “The myeloid cell superclusters were further deconvoluted...”. Or was there only one myeloid supercluster (that makes no sense to me)? If so then please point it out in the figure, and explain why it’s named “myeloid cell” and how that differs from all the other myeloid cells in the study.*

We thank the reviewer for pointing out the ambiguity. To clarify, our human dataset contains a total of 27 myeloid superclusters (SCs): 14 of microglia, 5 of CAMs, and 8 of MDMs. These SCs were further deconvoluted into 192 subclusters (149 microglia, 10 CAMs, and 33 MDMs). We have revised the text to explicitly state this, ensuring it is clear that multiple distinct superclusters exist rather than a single “myeloid SC.”

Minor Comment 5. *10 clusters of CAMs and 33 clusters of MDM based “ is not confusing, but still why are CAMs plural while MDM is singular?*

We thank the reviewer for spotting this inconsistency. We have standardized the terminology and now refer to “10 clusters of CAMs and 33 clusters of MDMs.”

Minor Comment 6. The “3D” plots in Fig 3C are not really three-dimensional, as they have been printed on a two-dimensional surface (the PDF). Therefore, the y dimension can’t really be interpreted. Anyway, the plots are also confusing as they show z scores “based on individual microglial cells belonging to the shown cluster (Mg)” but for two of the contexts (MS NAVM and MS core), no Mg clusters are listed in the heading. What are those scores based on? I would suggest plotting these as 2D (x vs y) plots and to use color to depict the Z score without faux-3D rendering.

We thank the reviewer for this constructive suggestion. Our intent with Fig. 3C was to map per-cell z-scored module or gene expression values directly onto spatial coordinates, using both height and color to encode the same statistic. This approach (i) places all cells on a common fixed scale (mean = 0, SD = 1), enabling direct cross-sample comparisons, (ii) highlights local heterogeneity and microniches, and (iii) exposes outliers and gradients that may be less visible in averaged or interpolated maps. The use of cell centroids avoids area-based bias and ensures that each point corresponds to one microglial nucleus. This approach has already been successfully used in the Giotto R package, which is popular and one of the most cited R package for analysis of spatial transcriptomics data with the original paper (Dries *et al.* 2021)⁷ being cited more than 800 times until 15th Sep. 2025.

We agree that the “3D” aspect is inevitably limited when rendered on a static PDF. To address this, we have (i) clarified in the figure legend that height and color are redundant encodings of the same z-score statistic, (ii) corrected the headings for MS NAWM and MS core to specify the microglial clusters (**new Figure 3C**), and (iii) provided alternative 2D projections (x–y with color only) for the reviewer below (**Rebuttal Figure 2**). We believe, however, that our adapted Figure 3 D best reflects our findings.

Rebuttal figure 2: Image feature plots depicting z-scored expression of selected microglia supercluster (row wise) based on individual microglial cells belonging to the shown cluster (Mg) from Figure 3A.

References:

7. Dries, R. *et al.* Giotto: a toolbox for integrative analysis and visualization of spatial expression data. *Genome Biol* 22, 78, doi:10.1186/s13059-021-02286-2 (2021)